# KAP1 negatively regulates RNA polymerase II elongation kinetics to activate signal-induced transcription

Usman Hyder [1], Ashwini Challa[1], Micah Thornton [2], Tulip Nandu[2], W. Lee Kraus [2] & Iván D'Orso [1] ✉

Signal-induced transcriptional programs regulate critical biological processes through the precise spatiotemporal activation of Immediate Early Genes (IEGs); however, the mechanisms of transcription induction remain poorly understood. By combining an acute depletion system with several genomics approaches to interrogate synchronized, temporal transcription, we reveal that KAP1/TRIM28 is a first responder that fulfills the temporal and heightened transcriptional demand of IEGs. Acute KAP1 loss triggers an increase in RNA polymerase II elongation kinetics during early stimulation time points. This elongation defect derails the normal progression through the transcriptional cycle during late stimulation time points, ultimately leading to decreased recruitment of the transcription apparatus for re-initiation thereby dampening IEGs transcriptional output. Collectively, KAP1 plays a counterintuitive role by negatively regulating transcription elongation to support full activation across multiple transcription cycles of genes critical for cell physiology and organismal functions.

Transcription consists of successive and interconnected steps regulated by multiple factors[1–4]. Once regulatory elements are made accessible by chromatin remodelers and transcription factors, the pre-initiation-complex (PIC) and associated factors (e.g., Mediator and CDK7) help recruit RNA Polymerase II (Pol II) to gene promoters to facilitate transcription initiation[5–8]. Shortly after promoter escape, Pol II undergoes pausing by two major players, Negative Elongation Factor (NELF), which binds to the Pol II funnel and restrains entry of new nucleotides, and DRB Sensitivity Inducing Factor (DSIF, composed of SPT4/SPT5), which together with NELF, stabilizes paused Pol II[9–11]. Over the last few decades, Pol II pausing has been thought to give a time window for the recruitment of RNA processing, elongation, and termination factors for accurate transcription and RNA processing[1,11–14], and thus is a feature of active transcription.

Pause release is facilitated by the positive transcription elongation factor (P-TEFb) kinase (composed of CDK9/CycT1), which is recruited by several complexes in different contexts[15,16] and phosphorylates both pausing factors (DSIF and NELF) and the Pol II carboxy-terminal domain (CTD) at the Ser2 position[17–19]. After pause release, transcription proceeds with many factors binding Pol II to regulate elongation rate, including SPT5 (a subunit of DSIF), SPT6, and PAF1C[20–22]. Dependency on the different transcription activating steps (initiation, pause release, and elongation rate) varies widely across different gene subsets, and relies on the stimulus and cell type for signal-inducible genes, increasing the diversity of gene regulatory mechanisms to establish accurate cell fate choices[23].

Upon reaching the 3' end of the gene, Pol II decelerates so that two events can occur for transcription termination and recycling of Pol II molecules for subsequent rounds of transcription. First, the pre-mRNA undergoes cleavage at the polyadenylation site (pA) to release the mature mRNA; second, the termination machinery including the 5'–3' Xrn2 exonuclease degrades the remaining nascent RNA chain to allow

[1]Department of Microbiology, The University of Texas Southwestern Medical Center, Dallas, TX 75390, USA. [2]Laboratory of Signaling and Gene Regulation, Cecil H. and Ida Green Center for Reproductive Biology Sciences, The University of Texas Southwestern Medical Center, Dallas, TX 75390, USA. ✉e-mail: Ivan.Dorso@utsouthwestern.edu

for proper Pol II eviction from chromatin[24–29]. In line with the interconnectedness of each transcription step, many studies have reported mechanisms where the 5′ and 3′ ends of genes are in spatial proximity, and mutations in the pA site have been shown to dampen re-initiation[30,31].

While the factors that control transcription of constitutive genes during homeostatic conditions are well established, less is known about the mechanisms that regulate signal-induced transcription to establish accurate cell fate choices in response to environmental cues. Immediate Early Genes (IEGs), or Primary Response Genes, are a set of genes that must be activated with precise kinetics and magnitudes in response to a variety of environmental stimulation signals[32,33]. IEGs encode transcription factors (TFs) that establish transcriptional programs important for multiple cell fate responses including neuronal plasticity and memory retrieval, differentiation, inflammatory and immune responses, as well as cell transformation, among others[34–37]. Given the urgent transcriptional demand for IEGs, it is likely that they require unique regulatory strategies. Therefore, defining the "specialized" factors that facilitate IEGs transcription activation and the underlying mechanisms is an unmet biomedical need.

In this study, we reveal that the KAP1 protein (also known as TRIM28 or TIF1β) is one of those specialized factors that promote signal-induced, rather than homeostatic, transcription. Work over the last three decades has proposed that KAP1 loss has detrimental effects on cellular functions in diverse contexts, including embryonic development, cancer cell growth, immune cell development, and viral gene expression[38–42]. While many of these cellular phenotypes have been attributed to KAP1's well-established role as a transcriptional repressor of endogenous retroviruses (ERVs) and critical genes in progenitor cells, KAP1 also has other functions such as transcription activation of human and viral genes and regulation of the DNA damage response[43–49].

Given that KAP1 regulates many critical processes[50], precisely defining KAP1's roles in signal-induced transcription requires advanced genetic approaches to directly define the underlying molecular mechanisms and help mitigate confounding indirect effects accrued upon long-term factor elimination. To circumvent this caveat, we devised a dTAG-inducible acute KAP1 depletion system to monitor synchronized, signal-induced transcription during a stimulation time course using several genomic approaches. Here, we reveal a previously unnoticed, counterintuitive role for KAP1 in the transcription cycle. KAP1 negatively regulates Pol II elongation kinetics at the earliest time points of cell stimulation to ultimately facilitate signal-induced transcription activation of IEGs. We propose that by tuning the rate at which Pol II elongates upon cell stimulation, KAP1 maintains the proper progression throughout the transcription cycle by synchronizing the connection of elongation to downstream steps (termination, re-initiation, and subsequent elongation), thereby fulfilling the urgent transcriptional demand of IEGs to mount accurate cell fate responses.

## Results

### Acute KAP1 depletion leads to virtually no transcriptional changes in homeostatic conditions

We utilized the dTAG system[51] to tag endogenous KAP1 in HCT116 cells with an FKBP12$^{F36V}$ degron and a 2xHA-containing tag in the C-terminus (~12 kDa tag). Using this approach, once cells are treated with a dTAG ligand (dTAG-13, hereafter referred to as dTAG), the Cereblon E3 ubiquitin ligase recruits the tagged protein to the proteasome for degradation (Fig. 1a). Monitoring of protein degradation kinetics in this system revealed that expression of KAP1 is reduced by ~88% by 8 h of dTAG treatment, and thus this time point was used for all subsequent experiments (Fig. 1b; Supplementary Fig. 1a). Previous reports have suggested that chronic KAP1 depletion leads to defects in cell growth and major transcriptional changes in basal conditions[40,45]. Examining cell confluence demonstrated that the dTAG treatment

does not alter cell proliferation over the course of ~6 days (Supplementary Fig. 1b), despite consistent KAP1 depletion throughout the entire time course (Supplementary Fig. 1c), suggesting that the effects on cell growth in other systems[45] may be a long-term consequence of KAP1 loss.

To test if acute KAP1 depletion led to major nascent transcription changes, we used Transient Transcriptome-Sequencing (TT-Seq)[52], a 4sU-based approach to isolate nascent RNA followed by next-generation sequencing. Intriguingly, very few genes were upregulated ($n = 12$) or downregulated ($n = 17$) at the 2-fold level after acute KAP1 depletion, with ~300 genes differentially expressed when using a 1.5-fold threshold (Supplementary Fig. 1d). Notably, the 1.5-fold upregulated genes showed signatures of signal-inducible TF response networks including Activator Protein 1 (AP-1) (Supplementary Fig. 1e). Because these genes are signal-inducible, we cross-referenced our TT-seq data with gene expression datasets in response to environmental cues, including the serum response network in HCT116 cells[53], and found that a small subset of serum-inducible genes in HCT116 was modestly upregulated upon acute KAP1 depletion in the HCT116 KAP1$^{dTAG}$ cell line ($n = 7$, accounting for ~24% of IEGs identified in their study). These IEGs are either lowly or not expressed in basal conditions but can be induced up to ~100–1000-fold upon cell stimulation to produce proteins required for several cell fate choices, suggesting that modest gene upregulation (<1.5-fold) upon acute KAP1 depletion may have minimal, if any, consequences on protein expression and cellular phenotypes.

Given the prominent role of growth-factor signaling to cancer development, we decided to examine if and how KAP1 modulates the transcriptional response to cell stimulation (when IEGs are truly expressed), using the serum response network as a surrogate, which has been implicated in multiple tumorigenic phenotypes.

### KAP1 is required for transcription activation of IEGs upon serum stimulation

To evaluate if KAP1 regulates signal-induced transcription (specifically the serum response network), cells were initially serum starved, treated with dTAG (8 h) to deplete KAP1 (or DMSO vehicle control), and later replenished with serum-containing media to induce IEGs expression for various treatment time points (Fig. 1c). Acute KAP1 depletion modestly decreased the expression of candidate IEGs (e.g., FOS, ATF3, NR4A1, and EGR1) (ranging from ~8–35% depending on the gene and time point), with no major changes to a control gene (U6), suggesting that KAP1 is required for full transcription activation of this subset of IEGs upon serum stimulation (Fig. 1d; Supplementary Fig. 1f).

To assess this function globally, serum-starved cells were pre-treated with dTAG or vehicle DMSO control and later challenged with serum (30 min) to induce IEGs or remained untreated. Cells were collected for RNA-Seq, which can easily capture the acute and high magnitude of inducible transcription activation. Differential expression (DE) analysis identified serum-inducible genes (hereafter referred to as IEGs) that were clustered according to their level of induction: 2-fold IEGs ($n = 236$), 4-fold IEGs ($n = 69$), 8-fold IEGs ($n = 34$), and 16-fold IEGs ($n = 16$), as well as a set of control uninduced, but expressed, genes labeled as non-DE genes ($n = 10,361$) (Fig. 1e, f). In line with the RT-qPCR data (Fig. 1d; Supplementary Fig. 1f), acute KAP1 depletion led to an overall decreased expression pattern for the four IEGs clusters, ranging from 8 to 17% median decrease, with the strongest effects at the most highly inducible genes (16-fold IEGs) (Fig. 1g). In addition, randomly selected genes from the non-DE cluster remained unperturbed upon acute KAP1 depletion, suggesting that KAP1-dependency is IEG-specific (Fig. 1g). Importantly, the changes observed at the RNA level are consistent with alterations at the protein level, as KAP1 loss upon a serum time course leads to decreased protein expression of two IEGs: c-Fos (with maximum difference with and without dTAG at 30–120 min post-serum stimulation) and ATF3 (with maximum

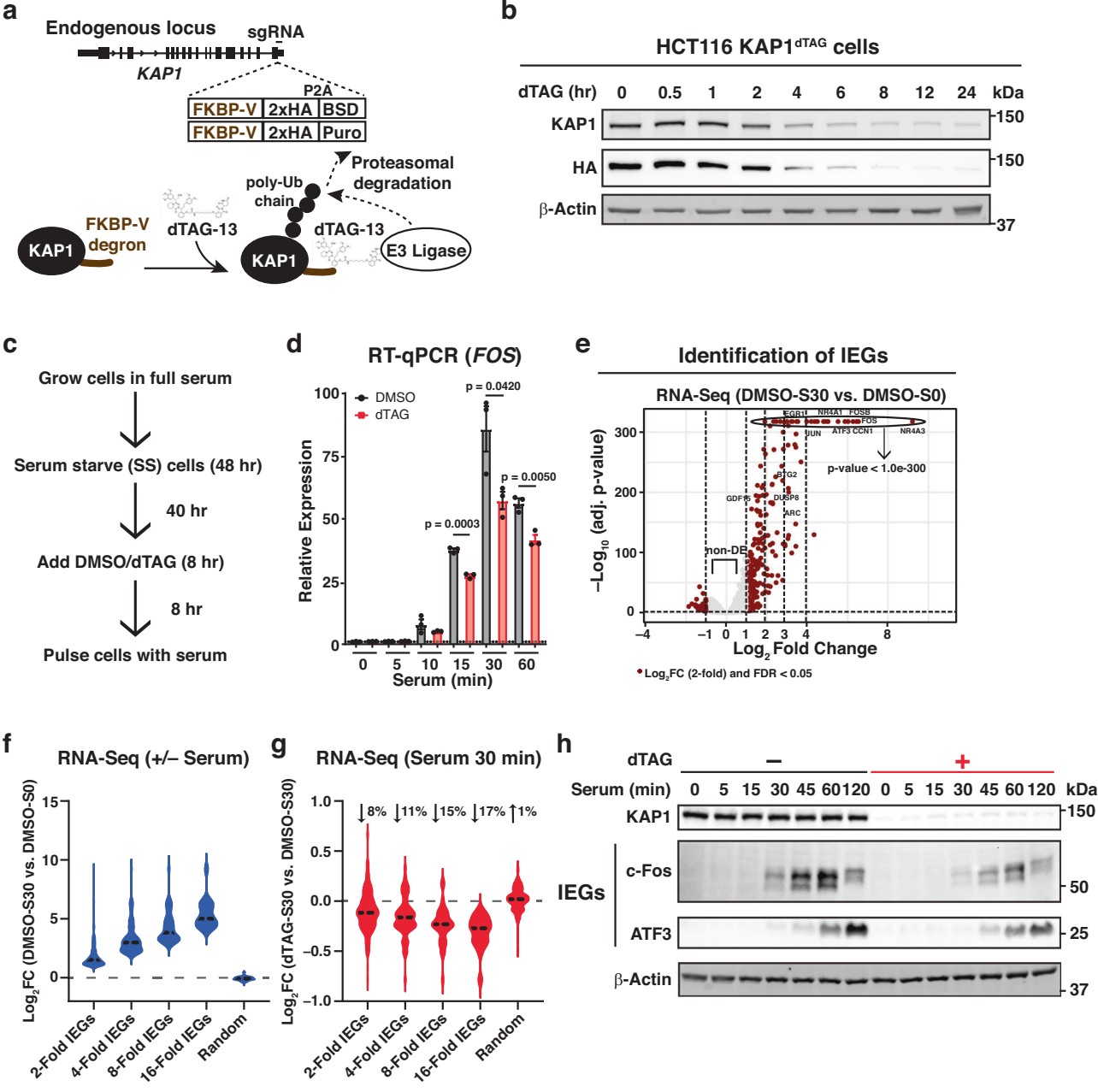

**Fig. 1 | KAP1 is a positive regulator of serum-induced transcription. a** Scheme showing dTAG strategy targeting the C-terminus of *KAP1* with the dTAG cassette. FKBP-V: degron; 2xHA: two tandem HA tags; P2A: 2A peptide sequence; Puro: Puromycin resistance cassette; BSD: Blasticidin resistance cassette. **b** Western blot showing dTAG-mediated KAP1 degradation kinetics in HCT116 KAP1$^{dTAG}$ cells. Blots are representative of two independent experiments. **c** Scheme showing the experimental design alongside the cell treatments. **d** RT-qPCR assay highlighting gene expression of one representative IEG (*FOS*) in DMSO- and dTAG-treated cells during a serum stimulation time course. Data represent mean ± SEM ($n = 3$ biological replicates, two-sided Student's *t*-test comparing DMSO to dTAG at each of the indicated time points). *P*-values are indicated. Black: DMSO, red: dTAG. **e** RNA-Seq volcano plot identifying IEGs as upregulated genes upon 30 min of serum stimulation ($n = 3$ biological replicates, two-sided Likelihood Ratio Test using the Benjamini–Hochberg (BH) multiple comparisons adjustment, false discovery rate [FDR] <0.05). Dots marked red were identified as differentially expressed. **f** Violin plot showing upregulation of IEGs organized by induction of expression: 2-fold IEGs ($n = 236$), 4-fold IEGs ($n = 69$), 8-fold IEGs ($n = 34$), and 16-fold IEGs ($n = 16$). Random

($n = 236$) denotes random genes selected from non-DE genes ($n = 10,361$). Data represents the Log$_2$FC value computed from RNA-Seq analysis between the DMSO-S0 and DMSO-S30 condition (generated from 3 biological replicates). The *n* number represents the number of genes in each cluster plotted. The median is denoted as a dashed black line. **g** Violin plot showing downregulation of IEGs upon acute KAP1 depletion organized by induction of expression: 2-fold IEGs ($n = 236$), 4-fold IEGs ($n = 69$), 8-fold IEGs ($n = 34$), and 16-fold IEGs ($n = 16$). Random ($n = 236$) denotes random genes selected from non-DE genes ($n = 10,361$). Data represents the Log$_2$FC value computed from RNA-Seq analysis between the DMSO-S30 and dTAG-S30 conditions (generated from 3 biological replicates). The *n* number represents the number of genes in each cluster plotted. Median % changes in expression are listed above the violin in each IEGs cluster and Random non-DE genes. The median is also denoted as a dashed black line. **h** Western blot showing protein expression of two representative IEGs (c-Fos and ATF3) during a serum time course ± dTAG treatment. Blots are representative of three independent experiments. Source data are provided as a Source Data file.

difference with and without dTAG at 120 min post-serum stimulation) (Fig. 1h) in line with the kinetics identified in the RT-qPCR data (Fig. 1d; Supplementary Fig. 1f). As expected, minimal protein expression of the two IEGs (c-Fos and ATF3) was detected in the 0 min serum stimulation time point with and without dTAG (Fig. 1h).

Because serum-induced IEGs expression relies both on transcription and upstream cell signaling events, bona fide cell signaling markers were interrogated to address the likelihood of an indirect signaling defect explaining the transcriptional phenotype. The master regulator Elk-1 is a critical component of the ternary complex that binds the serum response element and mediates gene activity through site-specific phosphorylation by signaling kinases in response to serum and growth factors[53–55]. To evaluate the possible contribution of signaling events to the long-term transcriptional phenotypes (decreased IEGs transcription upon KAP1 depletion), Elk-1 phosphorylation (pElk-1) was monitored throughout a serum stimulation time course in cells with and without KAP1. Remarkably, acute KAP1 depletion had no major effects on bulk levels of total Elk-1 and serum-induced site-specific pElk-1 (Phos-S383) during the 5–120 min serum stimulation time course (Supplementary Fig. 1g). Importantly, expression of KAP1 and c-Fos proteins were included to demonstrate KAP1 depletion and the reduced c-Fos expression upon KAP1 loss (Supplementary Fig. 1g). While these results indicate that the transcriptional defects may not be explained by differences in major cell signaling events such as pElk-1, they do not demonstrate if KAP1 regulates the occupancy of Elk-1 on chromatin. However, attempts to monitor total Elk-1 and pElk-1 by ChIP-Seq with commercial, previously validated antibodies[53], were unsuccessful in any condition.

Overall, these data indicate that KAP1 is required for full activation of IEGs upon serum stimulation, and that acute KAP1 loss does not alter key cell signaling events of the serum response network that initially turn on and later sustain IEGs transcription during serum stimulation, together suggesting that KAP1 is a regulator of IEGs transcription. Below, we investigate the functional consequences of IEGs transcriptional regulation upon acute KAP1 depletion prior to and after serum stimulation.

## KAP1 is redistributed from the promoters to the gene bodies of IEGs upon stimulation

To assess if KAP1 is recruited to IEGs to control their transcription, we performed Chromatin-ImmunoPrecipitation (ChIP)-Seq using an HA antibody directed against the C-terminal dTAG epitope (Fig. 1a) before (0 min) and after two serum stimulation time points (30 and 180 min), in addition to a dTAG sample treated with 30 min of serum to ensure KAP1 ChIP-Seq signal specificity. Metagene profile at 16-fold IEGs (Fig. 2a) and the *FOS* browser track (Fig. 2b) revealed that KAP1 occupies IEGs promoter-proximal (PP) regions in basal conditions, but also localizes to IEGs gene bodies (GB) and 3′ ends upon 30 min of serum treatment, suggesting that KAP1 travels with Pol II upon stimulation, perhaps like elongation, RNA processing, and termination factors[25,56]. Extension of the metagene profile 50-kb away from the TSS and pA sites revealed signal above background noise, highlighting the specificity of the KAP1 ChIP data (Supplementary Fig. 2a).

Quantitation of signal at PP regions (−100/+500-bp from the Transcription Start Site (TSS)), GB regions (+500-bp from TSS to pA site), and 3′ ends (pA to +3000-bp downstream) revealed that KAP1 occupancy modestly decreases at PP regions upon serum stimulation (median decrease of 27% for 4-fold IEGs and 14% for 16-fold IEGs), but largely increases in GB regions (11% for 4-fold IEGs and 84% for 16-fold IEGs) and 3′ ends (10% for 4-fold IEGs and 88% for 16-fold IEGs) (Fig. 2c, d; Supplementary Fig. 2b, median values reported), signifying that KAP1 is redistributed from the PP to GB regions of IEGs upon cell stimulation with serum. Importantly, low KAP1 occupancy was

detected in the dTAG and serum-treated cells, validating ChIP specificity (Fig. 2a–d; Supplementary Fig. 2a, b).

While the highly inducible, 16-fold IEGs cluster has the highest induction of KAP1 protein occupancy in the GB regions and 3′ ends of IEGs, signal quantitation at 4-fold IEGs also showed similar KAP1 redistribution (Fig. 2d; Supplementary Fig. 2b), which is in line with the RNA-Seq data (Fig. 1), strongly indicating that KAP1 promotes IEGs transcription broadly. Additionally, signal quantitation in GB regions revealed that while 180 min of serum stimulation began to redistribute KAP1 away from GB regions of 16-fold IEGs, 4-fold IEGs still had increased KAP1 density at their GB regions (Fig. 2d), potentially due to this lowly expressed IEGs cluster having longer kinetics of activation at the 30 min serum stimulation time point. Strikingly, this mode of reversible factor occupancy distribution (promoter to gene body and back to the promoter) during a cell stimulation time course differs from that of classical TFs, which get recruited to gene promoters, suggesting that KAP1 may be required to positively regulate transcription at post-initiation step(s) in the gene bodies of IEGs.

To assess the specificity of induced KAP1 redistribution from the PP to the GB regions of IEGs upon serum stimulation, peak calling was performed to define genome-wide KAP1 occupancy shifts upon serum stimulation. Here, $n = 12,665$ peaks ("All KAP1 peaks") were called in the no serum condition (time 0) followed by metagene profile analysis (Fig. 2e). Strikingly, mean KAP1 occupancy at the gene promoters of KAP1 peaks was decreased upon serum stimulation, suggesting that serum induces a potential redistribution of KAP1 from the PP regions of non-IEGs to IEGs, perhaps to selectively regulate IEGs transcription.

Browser track of KAP1 ChIP-Seq signal at a non-IEG (*SFPQ*) (Fig. 1f) and quantitation of KAP1 density at PP regions of the top-ranked 200 peaks (Supplementary Fig. 2c) underscore these results, which additionally showed that KAP1 occupancy begins to return to baseline by 180 min serum stimulation, the point at which much of serum-induced transcription has ended.

Overall, chromatin profiling of KAP1 at IEGs revealed three salient features: (1) KAP1 occupancy at PP regions during serum starvation, (2) specific shifts of KAP1 occupancy from the PP to the GB regions upon serum stimulation, and (3) reversibility of KAP1 occupancy from the GB regions back to the PP regions upon completion of the serum response cycle. Taken together, these observations provide evidence of signal-regulated transcription control by KAP1.

## Acute KAP1 depletion leads to disparate Pol II occupancy alterations during early and late serum stimulation

Given that KAP1 is recruited to IEGs, is required for their full activation, and directly interacts with Pol II[45], we probed how acute KAP1 depletion affects Pol II occupancy using ChIP-Seq with an antibody targeting the RPB3 subunit. Given that signal-induced transcription is dynamic, Pol II ChIP-Seq was performed before (0 min) and after two serum stimulation time points (15 and 30 min) with dTAG treatment or vehicle DMSO control. Metagene analysis at the 16-fold IEGs cluster at 0 min showed no noticeable changes to the Pol II signal when comparing DMSO to dTAG, signifying that acute KAP1 depletion does not dramatically alter Pol II levels at IEGs in the basal condition, prior to serum stimulation (Supplementary Fig. 3a).

Unexpectedly, KAP1 depletion leads to disparate Pol II occupancy phenotypes at IEGs at the two evaluated serum stimulation time points (15 and 30 min) (Fig. 3a, b). At 30 min stimulation, Pol II levels across IEGs modestly decreased upon KAP1 depletion in PP regions (7% for 4-fold IEGs and 14% for 16-fold IEGs), GB regions (7% for 4-fold IEGs and 18% for 16-fold IEGs), and 3′ ends (1% for 4-fold IEGs and 8% for 16-fold IEGs) (Fig. 3b–e, superimposed metagene with HA ChIP-Seq in Supplementary Fig. 3b), in line with decreased IEGs expression by RNA-Seq at this time point (Fig. 1).

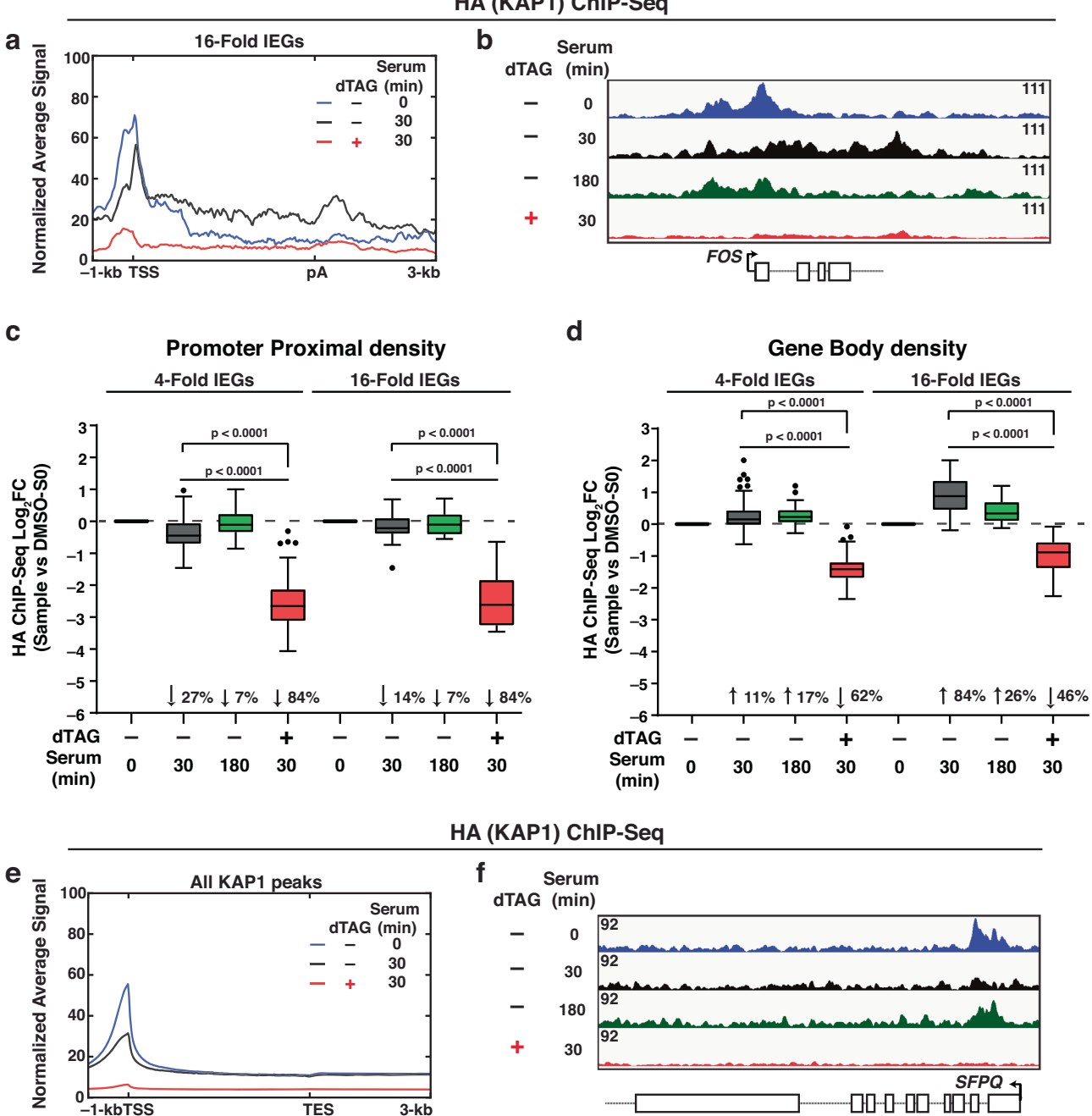

**Fig. 2 | KAP1 localizes to the gene bodies and 3' ends of IEGs upon serum stimulation. a** HA (KAP1) ChIP-Seq metagene analysis at 16-fold IEGs in the three indicated conditions. See legend for sample identification. **b** HA (KAP1) ChIP-Seq browser track of *FOS* in the four indicated conditions. See legend for sample identification. **c**, **d** HA ChIP-Seq quantitation of KAP1 density at 4-fold IEGs (*n* = 69) and 16-fold IEGs (*n* = 16) at **c** PP regions and **d** GB regions. Data represents the Log₂FC value for the respective sample (see X-axis) normalized to serum 0 min in DMSO-treated cells using ChIP-Seq signal from 2 biological replicates. The *n* number represents the number of genes in each cluster plotted. The Tukey plots indicate the median (black center line), the first and third quartiles (edges of the box), and the 1.5× interquartile range below and above the box as whiskers. Dots are presented as genes with normalized signals beyond these defined ranges. Statistics were calculated between the dTAG plus serum treatment condition and the respective condition shown on the Tukey plot. Two-sided Wilcoxon signed-rank test. *P*-values are indicated. **e** HA (KAP1) ChIP-Seq metagene analysis of all genome-wide KAP1 peaks (*n* = 12,665) in the three indicated conditions. See legend for sample identification. **f** HA (KAP1) ChIP-Seq browser track at the *SFPQ* locus in the four indicated conditions. See legend for sample identification. Source data are provided as a Source Data file.

However, surprisingly, at the 15 min mark, Pol II density primarily increased in GB regions (14% for 4-fold IEGs and 18% for 16-fold IEGs) and 3' ends (11% for 4-fold IEGs and 33% for 16-fold IEGs) (Fig. 3a, d, e) with no major changes observed at PP regions (Fig. 3a, c). A zoomed-in metaprofile of the GB region is provided to underscore these results (Fig. 3a).

Further, genome browser tracks of Pol II and HA (KAP1) ChIP-Seq data of two candidate IEGs (*NR4A1* and *ATF3*) illustrate examples of the opposing Pol II occupancy phenotype at the metagene level (Fig. 3f, g). To test the specificity of these findings, we assessed Pol II occupancy dynamics at non-DE genes and found no changes with or without dTAG in either the 0, 15, or 30 min time points, highlighting

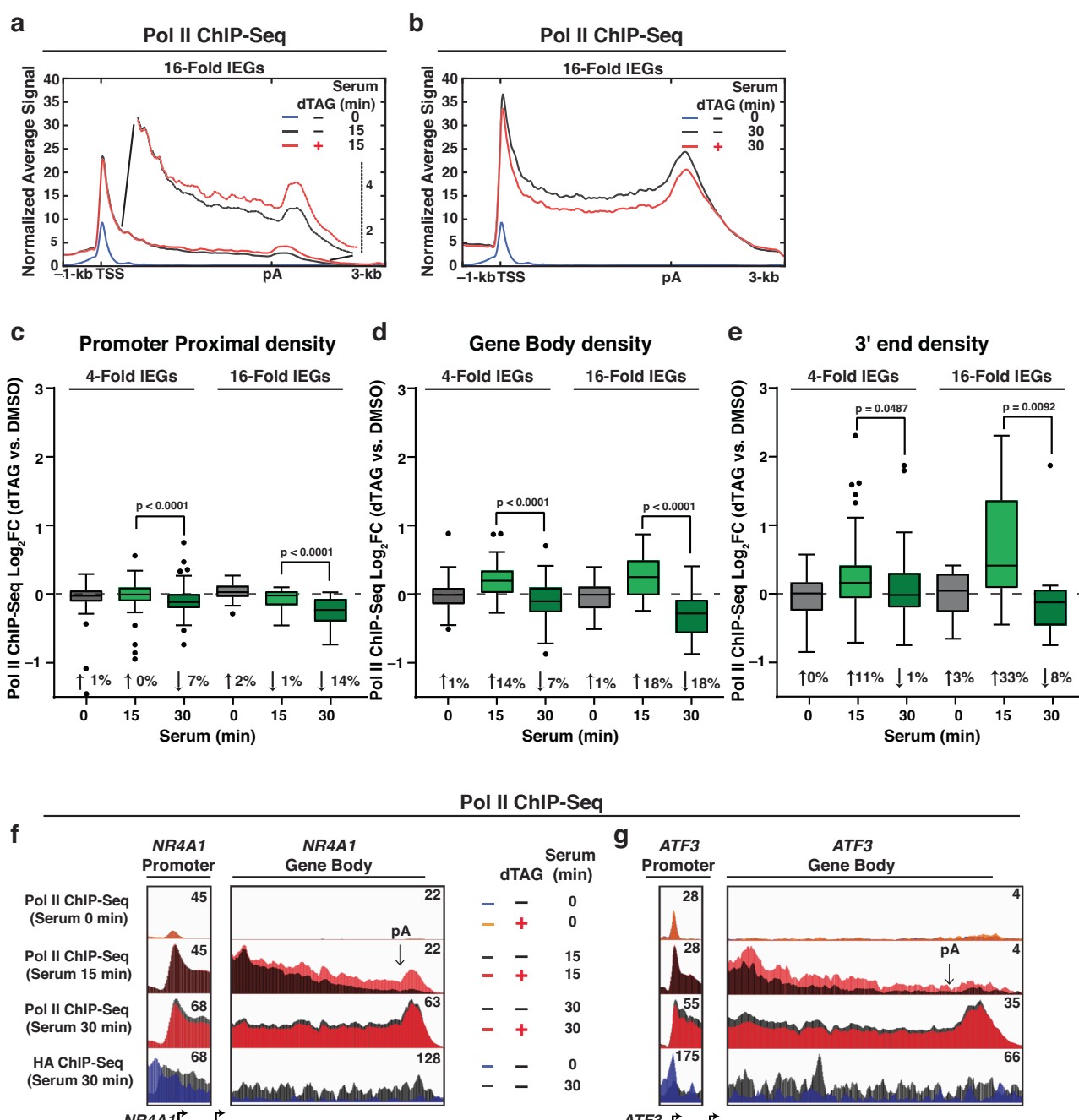

**Fig. 3 | KAP1 regulates Pol II occupancy during serum stimulation. a, b** Pol II ChIP-Seq metagene at 16-fold IEGs showing occupancy of Pol II at: **a** 15 min (with zoomed-in profile to show differences) and **b** 30 min serum stimulation time points. See legend for sample identification. **c-e** Pol II ChIP-Seq quantitation of Pol II density at 4-fold IEGs (*n* = 69) and 16-fold IEGs (*n* = 16) at: **c** PP regions, **d** GB regions, and **e** 3′ ends. Data represents the Log₂FC value for dTAG versus DMSO at the respective serum time point (see X-axis) using ChIP-Seq signal from 2 biological replicates. The *n* number represents the number of genes in each cluster plotted. The Tukey plots indicate the median (black center line), the first and third quartiles

(edges of the box), and the 1.5× interquartile range below and above the box as whiskers. Dots are presented as genes with normalized signals beyond these defined ranges. Statistics were calculated between the Log₂FC values between the early (15 min) and late (30 min) time points shown on the Tukey plot. Two-sided Wilcoxon signed-rank test. *P*-values are indicated in the figure. **f, g** Pol II and HA ChIP-Seq browser track in multiple conditions. See legend for sample identification at the: **f** *NR4A1* locus and **g** *ATF3* locus. The arrow indicates the polyadenylation (pA) site. Source data are provided as a Source Data file.

---

again that changes observed at IEGs are specific and centered on signal-induced, rather than basal, transcription (Supplementary Fig. 3c–f).

Notably, the strongest increases in Pol II signal upon KAP1 depletion in the 15 min serum stimulation time point were found at the 3′ end, after the pA site (Fig. 3a, e, f, g), leading to a Pol II pileup. However, at this time point, the increases in density can be seen ~30%

in the GB regions (Fig. 3a), suggesting that transcription alterations upon acute KAP1 depletion potentially start during early elongation, rather than at the termination stage, and lead to an accumulation of Pol II at the 3′ end.

Collectively, increased Pol II occupancy upon KAP1 depletion at an early stimulation time point (15 min) led to decreased Pol II occupancy at a late stimulation time point (30 min) ultimately dampening IEGs

expression, consistent with other metrics such as transcriptome and protein expression analyses (Fig. 1).

## Acute KAP1 depletion increases Pol II elongation kinetics during early serum stimulation

The divergent fluctuations in Pol II occupancy at IEGs upon KAP1 depletion (increased levels at 15 min but decreased levels at 30 min) suggested a kinetic difference at the early stimulation time point (15 min) that negatively affected transcription at the late stimulation time point (30 min) (Fig. 3). Given that ChIP-Seq detects a combination of different kinds of Pol II molecules (e.g., paused, active, backtracked)[57], it is difficult to assess if the increases in Pol II density at the GB regions and 3' ends in the absence of KAP1 are indicative of a faster moving, actively transcribing Pol II versus a slower moving Pol II that is being retained on chromatin for longer times.

To determine which model is likely accurate, temporal Precision Run-On (PRO)-Seq was performed during early serum stimulation time points (5 and 10 min) to identify the earliest time point that captures nascent transcripts and where active Pol II reaches the ends of most IEGs. Importantly, PRO-Seq signal at GB regions can be used to monitor nascent transcription[58], and thus provides a proxy to quantitate newly synthesized RNAs with and without KAP1 at early serum stimulation time points.

Active Pol II was detected at the GB regions of IEGs in control (DMSO-treated) cells at 5 min serum stimulation (Fig. 4a) and reached their 3' ends at 10 min serum stimulation (Fig. 4b). Notably, active Pol II had reached further into the GB regions of IEGs at 5 min serum stimulation upon KAP1 depletion (Fig. 4a, see zoomed-in metaprofile of the GB region), and its density increased towards the end of the GB regions and 3' ends at 10 min serum stimulation (Fig. 4b). Both findings are consistent with the increased Pol II occupancy upon KAP1 depletion at the early (15 min) serum stimulation time point detected by ChIP-Seq (Fig. 3a). Quantitation of signal in PP regions, GB regions, and 3' ends underscore these results (Supplementary Fig. 4a–c), with signal in GB regions revealing that nascent transcription at 4-fold IEGs and 16-fold IEGs increased upon KAP1 depletion across both 5 and 10 min serum stimulation time points, which was almost ~2-fold upregulated for 16-fold IEGs at the 5 min time point (Fig. 4c). These data together highlight that the increased Pol II density observed by ChIP-Seq at 15 min of serum stimulation upon KAP1 depletion (Fig. 3) is indicative of actively transcribing Pol II (Fig. 4c).

Genome browser tracks of several IEGs [FOSB (Fig. 4d), FOS (Supplementary Fig. 4d), NR4A1 (Supplementary Fig. 4e)] and a control non-IEG (FUS) (Supplementary Fig. 4f) demonstrate specific increases in active Pol II at IEGs at the earliest time points of serum stimulation upon KAP1 depletion, consistent with the metagene analysis. Additionally, no major alterations to active Pol II were detected at the GB regions of IEGs prior to serum stimulation (0 min) in DMSO and dTAG-treated cells, consistent with lack of active transcription (Supplementary Fig. 4g) and lack of Pol II occupancy changes in ChIP-Seq assays (Supplementary Fig. 3a). Moreover, no major alterations in PRO-Seq signal were detected at non-DE genes at any time point with dTAG or vehicle DMSO control treatment (Supplementary Fig. 4h–j), reinforcing the specificity of KAP1 depletion solely affecting serum-induced transcription of IEGs.

## A method to estimate elongation rates solidifies the notion of KAP1 as a negative regulator of Pol II elongation kinetics

The fact that Pol II transcribes further into IEGs upon KAP1 depletion is indicative of an increase in elongation rate. Many methods have been developed to measure elongation rates. Most notably, the use of transcription elongation inhibitors such as 5,6-dichloro-1-β-d-ribofuranosylbenzimidazole (DRB) to force Pol II pausing coupled with its washout to release paused Pol II to monitor elongating Pol II as it travels from the beginning to the ends of genes through, for example,

DRB-coupled Bru-seq (DRB-Seq)[59]. However, the quick rate of Pol II transcription after release alongside the need for intermediate time points at which Pol II has not reached the ends of genes, limits the DRB-Seq approach to long genes (>25-kb)[60]. The average gene length in the 4-fold IEGs cluster is ~17-kb, with many IEGs such as FOS (Supplementary Fig. 4d), being under 10-kb, thus posing a problem in using this method to measure Pol II elongation rate during serum stimulation.

An alternative to measuring the elongation rate for signal-inducible genes is the "leading edge" approach, which tracks the wavefront of active Pol II as it travels through a gene upon its temporal induction[23]. While the Pol II wavefront is further into IEGs when KAP1 is depleted, an indication of increased elongation rate (Fig. 4a), this approach is still dependent on gene length given that it requires the definition of the wavefront, which for some IEGs (e.g., FOS) have already reached their termination sites in both DMSO- and dTAG-treated cells by 5 min of serum stimulation.

Given the above constraints with the DRB-Seq and leading-edge approaches, we devised a method called the Rate of Change in Coverage (ROCC), which relies on using nascent RNA signal derived from PRO-Seq rather than the transcribing Pol II or the nascent RNA wavefront. The ROCC is estimated at each bin (50-bp) by examining the coverage at that position at each time point and regressing the coverage on time directly. The high-resolution estimates (one per each bin in the gene) are then combined to provide a gene-wide estimate, which, while not an exact approximation of the elongation rate (bp/min), serves as a "Proxy Rate" calculation that infers the elongation rate.

To provide benchmark validation that the ROCC method is consistent with elongation rates previously calculated using the leading edge method, a published nascent transcription dataset for estrogen-inducible genes ($n = 76$) was examined to estimate elongation rates according to the leading edge approach[23] and the newly devised ROCC method. A comparison of the two datasets showed that there is a statistically significant correlation between the average ROCCs and the original rates that were computed according to the Hidden Markov Model used in the leading edge approach (Supplementary Fig. 5a). While the two methods are indeed correlated (Pearson's $\rho = 0.40$, $p = 0.0002169$), it is noteworthy that the leading edge and ROCC approaches examine distinctive characteristics of the same dataset. The leading edge calls the most likely position of the front of the transcribing wave, while ROCC utilizes a nascent transcription signal to estimate a rate specifically for each binned position. Thus, we emphasize that ROCC is a "Proxy Rate" that infers the Pol II elongation rate, allowing us to bypass the gene length constraints of serum-inducible IEGs.

To calculate the ROCC for the serum dataset, the normalized binned signal at each time point was quantified in the GB region plus 3' end of each IEG (500-bp after the TSS to 3000-bp after the pA site), and a generalized linear model was applied to define the slope at which signal intensity increases at each bin before (0 min) and after two serum stimulation time points (5 and 10 min). This slope, calculated for both DMSO- and dTAG-treated cells, is estimated as the ROCC value for each 50-bp bin, which is then averaged across the entire gene to generate a single average ROCC value (the Proxy Rate). Utilization of this method revealed that KAP1 depletion led to an enhanced Proxy Rate at 4-fold IEGs and 16-fold IEGs across the 10 min serum stimulation time course, suggesting that indeed KAP1 depletion leads to increased Pol II elongation kinetics (Fig. 4e).

Given that IEGs are transcriptionally inert prior to serum stimulation, many bins in this time point are quantified as no signal, making many rate calculations uninterpretable. Due to this, for the Proxy Rate calculation for both the benchmarking (Supplementary Fig. 5a) and serum dataset (Fig. 4e), the y-intercept was fixed to 0 and the signal in the 0 time point was positioned at 0.5 min to allow precise

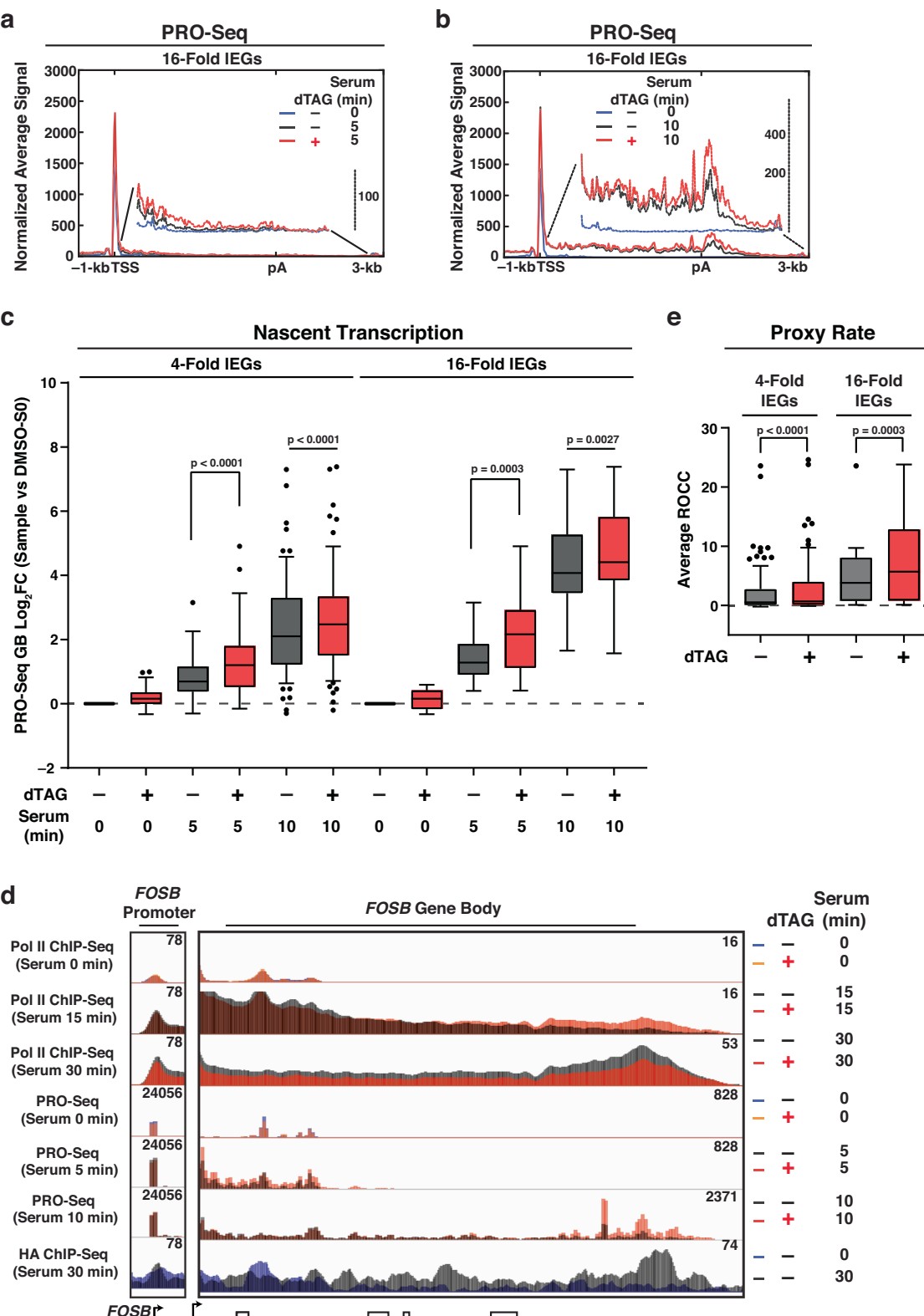

comparability between DMSO and dTAG conditions and to avoid negligible changes in the serum 0 time point subjugating large effects on the final rate calculation. However, to validate that fixing the y-intercept did not significantly modify the results, the ROCC was also estimated between the serum 0 and 5 min time points and serum 0 and 10 min time points individually to create a line with two points and a slope, with the slope being marked as the ROCC. This "single time point" approach allows the serum 0 min time point to exert maximal

influence on final ROCC values. Notably, the Proxy Rate estimates between the 0 and the individual (5 and 10 min) time points also revealed marked increases upon KAP1 depletion, highlighting that KAP1 indeed is a negative regulator of elongation kinetics during early serum stimulation time points (Supplementary Fig. 5b, c).

Individual IEGs have different kinetics of activation (Fig. 4d, Supplementary Fig. 4d, e). For example, *FOS* mRNA can be detected by 10 min of serum stimulation (Fig. 1d), while *NR4A1* is not detected

**Fig. 4 | Acute KAP1 depletion leads to increased elongation kinetics at early serum stimulation. a**, **b** PRO-Seq metagene profile at 16-fold IEGs showing active Pol II density at the: **a** 5 min (with zoomed-in profile to show differences) and **b** 10 min serum stimulation time points. See legend for sample identification. **c** PRO-Seq quantitation of nascent transcription (GB density) for 4-fold IEGs (*n* = 69) and 16-fold IEGs (*n* = 16). Data represents the Log$_2$FC value for the respective sample (see X-axis) normalized to serum 0 min in DMSO-treated cells using PRO-Seq signal from 2 biological replicates. The *n* number represents the number of genes in each cluster plotted. The Tukey plots indicate the median (black center line), the first and third quartiles (edges of the box), and the 1.5× interquartile range below and above the box as whiskers. Dots are presented as genes with normalized signals beyond these defined ranges. Statistics were calculated between the indicated samples in the plot (dTAG vs. DMSO at both 5 and 10 min serum time points).

Two-sided Wilcoxon signed-rank test. *P*-values are indicated. **d** Pol II ChIP-Seq, PRO-Seq, and HA ChIP-Seq browser track in multiple conditions at the *FOSB* locus. See legend for sample identification. **e** Rate of Change in Coverage (ROCC) calculation (Proxy Rate) for 4-fold IEGs (*n* = 69) and 16-fold IEGs (*n* = 16) computed through the serum stimulation time course. Data represents the absolute average ROCC values for the respective sample (see X-axis) using PRO-Seq data from 2 biological replicates. The *n* number represents the number of genes in each cluster plotted. The Tukey plots indicate the median (black center line), the first and third quartiles (edges of the box), and the 1.5× interquartile range below and above the box as whiskers. Dots are presented as genes with normalized signals beyond these defined ranges. Statistics were calculated between the indicated samples in the plot (dTAG vs. DMSO) with a two-sided Wilcoxon signed-rank test. *P*-values are indicated.

above basal until ~30 min (Supplementary Fig. 1f), analogous to the fact that active Pol II reaches the end of *FOS* by 5 min of serum stimulation (Supplementary Fig. 4d) while it takes 10 min to reach the end of *NR4A1* (Supplementary Fig. 4e). Thus, we divided the length of each IEG into quartiles (e.g., a 10-kb gene is split into four 2.5-kb regions, labeled as quartile 1–4 (Q1–Q4)) and quantified the Proxy Rate for each IEG quartile using the ROCC values from the "single time point" approach. Expectedly, this analysis further revealed heterogeneous IEGs transcription kinetic profiles (Supplementary Fig. 5d). While *FOS* had increases in the estimated rate upon KAP1 depletion in all quartiles in the 5 min serum stimulation time point, no major changes were observed at the 10 min serum stimulation time point. Contrary, other IEGs (e.g., *FOSB* and *NR4A1*) had the greatest increases in ROCC values upon KAP1 depletion in the promoter-proximal quartiles (Q1/Q2) at the 5 min serum stimulation time point and had the greatest increases in the promoter-distal quartiles (Q3/Q4) at the 10 min serum stimulation time point (Supplementary Fig. 5d). These observations are likely dependent on different kinetic behaviors of individual IEGs getting expressed at different timescales and with various magnitudes (e.g., Pol II reaches the end of *FOS* in its first transcription cycle prior to *FOSB* and *NR4A1*, which are longer genes).

Collectively, we posit that enhanced Pol II elongation kinetics during early serum stimulation upon KAP1 depletion (Fig. 4) counterintuitively leads to an overall decrease in IEGs RNA and protein expression (Fig. 1), consistent with Pol II occupancy data (Fig. 3). Notably, while the overall changes in nascent transcription and shifts in active Pol II occupancy are modest, they are biologically meaningful given the decreased IEGs expression at late serum stimulation time points.

### Acute KAP1 depletion does not affect Pol II pause release during early serum stimulation

While the ChIP-Seq and PRO-Seq datasets collectively revealed increases in Pol II occupancy and elongation kinetics upon KAP1 depletion during early serum stimulation, these data do not entirely exclude changes in pause release as the mechanism for augmented nascent transcription. Pause release occurs when Pol II transitions from the promoter-proximal paused state into the gene for active transcription, a step primarily driven by CDK9-mediated phosphorylation of the Pol II CTD and the SPT5 subunit of DSIF[17,18]. While CDK9 phosphorylates several SPT5 residues, phosphorylation of the C-terminal region 1 (CTR1), including at T806, is characterized as a marker of pause release[17,61,62].

If Pol II pause release was primarily responsible for the nascent transcript increases upon KAP1 depletion, it would be expected that either CDK9 kinase occupancy and/or activity were increased at the early serum stimulation time point (5 min). To assess this alternative model, CDK9 occupancy and activity were monitored by ChIP-Seq upon acute KAP1 depletion and serum stimulation. Kinase occupancy was directly examined with an antibody detecting total CDK9 and kinase activity was indirectly evaluated with an

antibody towards SPT5 phosphorylated at T806 (referred here to as pSPT5). While the occupancy of CDK9, SPT5, and pSPT5 expectedly increased upon 5 min of serum treatment in control (DMSO-treated) cells, consistent with signal-induced transcription, acute KAP1 depletion led to no major increases in factor occupancy at 16-fold IEGs in a metagene analysis (Supplementary Fig. 6a, b, c). Notably, browser tracks of one representative IEG (*FOS*) and one control gene (*FUS*) support the metagene observations (Supplementary Fig. 6d, e).

Together these data suggest that KAP1's primary role may not be to inhibit CDK9 occupancy nor decrease its activity to block Pol II pause release to restrict early transcription. These data are consistent with the fact that elongation kinetics (Fig. 4, Supplementary Fig. 5) may be the primary mechanism for the perceived increases in active transcription upon KAP1 depletion at early serum stimulation time points.

### Acute KAP1 depletion decreases the occupancy of transcription initiation and elongation factors during late serum stimulation

Given that the increased elongation at early serum stimulation time points upon KAP1 depletion diminished IEGs transcription and Pol II recruitment at late serum stimulation time points, we predicted a model whereby KAP1 depletion indirectly impacts the assembly of transcription machinery at gene promoters at late serum stimulation time points.

To address this part of the model, we performed ChIP-Seq of transcription elongation (SPT5, CDK9) and initiation (CDK7, MED1, TFIIB) factors upon late (30 min) serum stimulation (Fig. 5). In response to serum treatment, all factors were shown to occupy the expected genomic regions with SPT5 localizing to PP regions, GB regions, and 3′ ends while CDK9, CDK7, MED1, and TFIIB primarily found in PP regions (Fig. 5a–c, Supplementary Fig. 7a, b). Notably, metagene analysis and signal quantitation at PP regions showed that KAP1 depletion led to modestly reduced occupancy of every factor tested, albeit at different levels (Fig. 5a–f, Supplementary Fig. 7a–d). Gene browsers of two representative IEGs (*FOS* and *ATF3*) (Fig. 5g, h) and one non-IEG (*FUS*) (Supplementary Fig. 7e) underscore the findings at the metagene level.

Pol II CTD phosphorylation at both Ser5P and Ser2P, which are deposited by CDK7 and CDK9 kinases, respectively, and which are important for transcription initiation, pause release, and termination, were also decreased upon KAP1 depletion relative to DMSO control during late serum stimulation (Supplementary Fig. 7f, g), consistent with decreased recruitment of transcription initiation and elongation factors. Importantly, Pol II Ser5P and Ser2P data was normalized to total Pol II, suggesting that alterations to kinase recruitment at late serum stimulation time points have direct downstream consequences on Pol II CTD phosphorylation.

Collectively, these data indicate that changes in Pol II elongation kinetics, which are elicited by loss of KAP1 transcriptional control during early serum stimulation, potentially dampen the recruitment

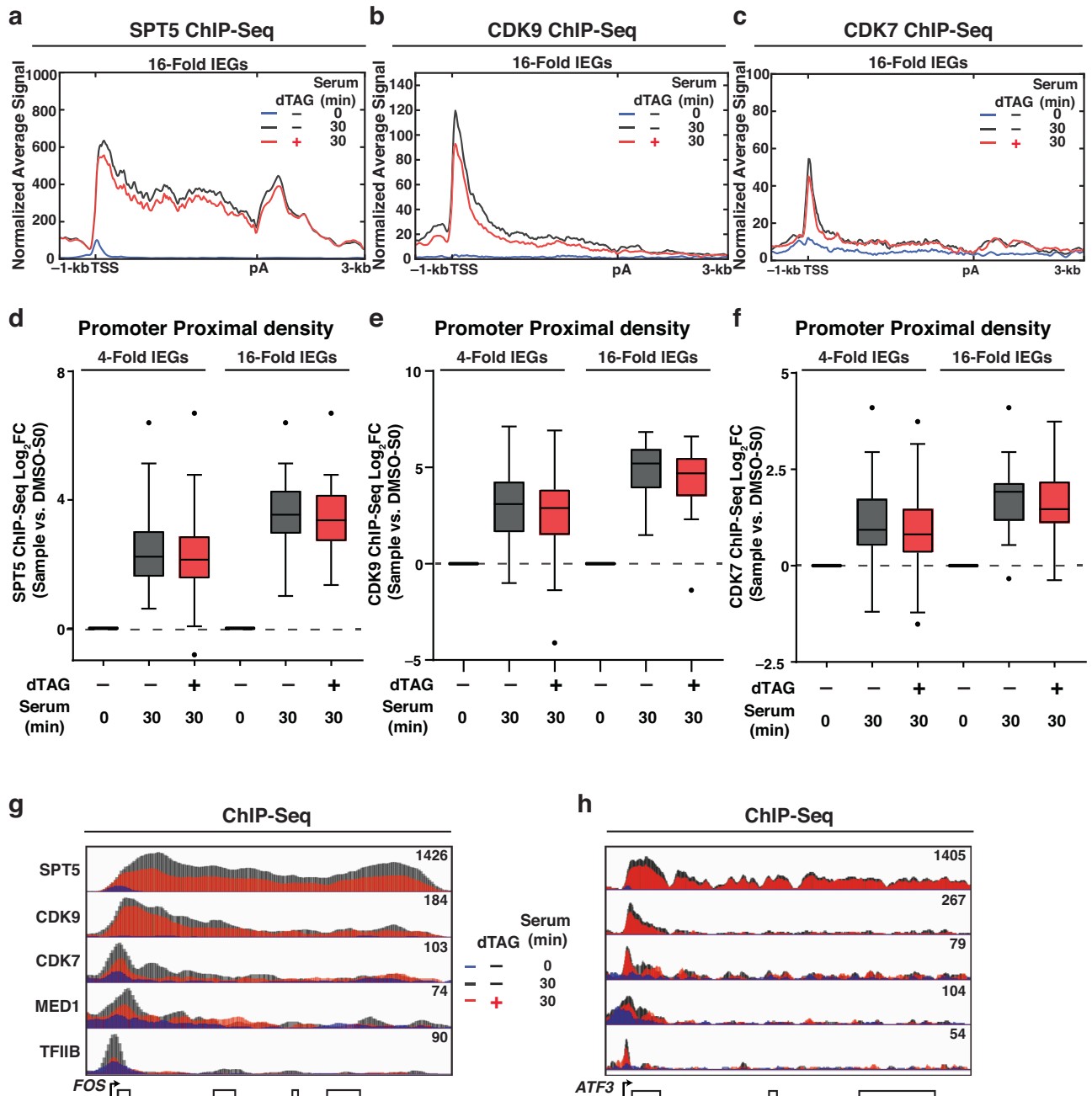

**Fig. 5 | Acute KAP1 depletion leads to decreased occupancy of regulators of transcription elongation and initiation at late serum stimulation. a–c** ChIP-Seq metagene profile of **a** SPT5, **b** CDK9, and **c** CDK7 at 16-fold IEGs in the three indicated conditions. See legend for sample identification. **d–f** ChIP-Seq quantitation of **d** SPT5, **e** CDK9, and **f** CDK7 in PP regions of 4-fold IEGs ($n = 69$) and 16-fold IEGs ($n = 16$). Data represents the Log2FC value for the respective sample (see X-axis) normalized to serum 0 min in DMSO-treated cells using ChIP-Seq signal from 1 biological replicate. The $n$ number represents the number of genes in each cluster plotted. The Tukey plots indicate the median (black center line), the first and third quartiles (edges of the box), and the 1.5× interquartile range below and above the box as whiskers. Dots are presented as genes with normalized signals beyond these defined ranges. **g, h** ChIP-Seq browser tracks of all factors in the three indicated conditions. See legend for sample identification at the **g** *FOS* locus and **h** *ATF3* locus. Source data are provided as a Source Data file.

of the transcription apparatus, including Pol II, at late serum stimulation, possibly explaining the overall decrease in IEGs expression levels.

## Discussion

By using an acute depletion system to directly examine KAP1 transcriptional roles, we provide evidence that KAP1 primarily contributes to the activation of signal-induced transcription via a counterintuitive mechanism of Pol II elongation control (Fig. 6a). Further, we provide

evidence that acute KAP1 depletion increases Pol II elongation kinetics during early stimulation time points, consequently derailing the normal progression through the transcription cycle during late stimulation, and ultimately dampening the magnitude of IEGs expression (Fig. 6b). Overall, our data reveal an additional layer of transcriptional control of signal-inducible genes requiring specialized factors like KAP1 to maintain Pol II elongation kinetics during early stimulation to guarantee the proper transition through multiple transcriptional cycles during late stimulation.

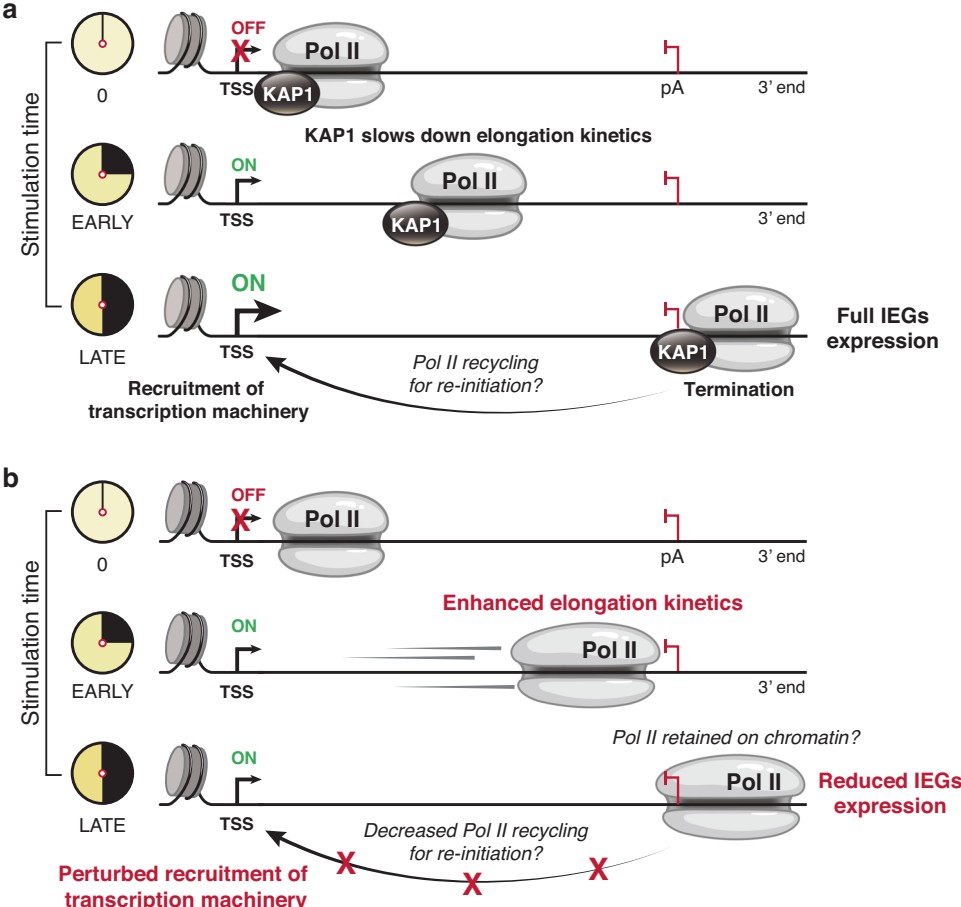

**Fig. 6 | KAP1 facilitates signal-induced transcription activation by negatively regulating Pol II elongation kinetics. a** Model of KAP1 facilitating signal-induced transcription by negatively regulating Pol II elongation kinetics during early stimulation to allow for the timely transition of elongation to termination and normal re-initiation events (including recruitment of the transcription apparatus to the promoter) during late stimulation. **b** KAP1 depletion increases Pol II elongation kinetics during early stimulation, consequently leading to defects in the recruitment of the transcription apparatus to the promoter during late stimulation thereby reducing IEGs expression.

Prior studies using chronic knockout or silencing methods have suggested that KAP1 promotes transcriptional pausing, and pause release or controls the switch from initiation into elongation to regulate basal and signal-induced transcription[44,45,47,63,64]. These studies have proposed that KAP1 physically interacts with several transcriptional regulators (e.g., TRIM24, EZH2, CDK9, Pol II, transcription factors) and that KAP1 depletion or silencing dampens their recruitment to gene promoters to facilitate transcription activation[40,44,45,65]. Our studies support the notion that at least some of these molecular events are potential indirect consequences of chronic KAP1 depletion, which dampens long-term transcription initiation and elongation because of abnormal increases in Pol II elongation kinetics during prior rounds of transcription, consistent with our findings that the transcription apparatus is largely deregulated during late serum stimulation time points (Fig. 5).

Our discoveries spur several interesting questions to define the molecular mechanisms. First, how does KAP1 regulate Pol II elongation kinetics? One potential mechanism is that KAP1 binds Pol II or Pol II-associated elongation rate machinery (e.g., SPT5) to decrease the elongation rate upon stimulation. KAP1 binds Pol II in cells and directly in vitro[45], highlighting that this is indeed a possibility and is in line with KAP1 traveling with Pol II upon stimulation (Fig. 2). Additionally, while SPT5 phosphorylation by CDK9 is a marker of pause release, and in our study did not change upon acute KAP1 depletion, SPT5 CTR1 phosphorylation has been extensively characterized to regulate elongation rate[26,61,66]. While we did not observe changes to the phosphorylation

status of T806 in the CTR1 domain, other SPT5 residues within either the CTR1 or CTR2 domains may be perturbed upon KAP1 depletion, which may confer KAP1-dependent changes in elongation rate.

An interesting non-mutually exclusive and SPT5-independent mechanism is that KAP1 negatively regulates elongation kinetics through the transient and dynamic scaffolding of heterochromatin machinery including Heterochromatin Protein 1 (HP1) family members alongside H3K9 methyltransferases to IEGs upon cell stimulation, reminiscent of how KAP1 silences retroelements[67,68]. This mechanism would propose that cells have evolved KAP1 to counterintuitively repurpose a repressive role for signal-induced transcription activation, revealing that KAP1's "split personality" (both repression and activation) could utilize similar repressive machinery but for divergent outputs. Supporting this view, recruitment of HP1 proteins and the deposition of histone silencing marks (H3K9me3) upon signal-induced transcription activation has been previously documented[69,70], and future studies will investigate if either a direct (Pol II-binding) or indirect (through silencing chromatin marks) mechanism influences KAP1's effect on Pol II elongation kinetics.

KAP1 could also regulate elongation indirectly as a participant in transcription-induced DNA damage and repair. It is well established that high rates of transcription induce localized DNA damage to resolve torsional strain induced by actively transcribing polymerases[71,72] and KAP1 has been suggested to be a part of the DNA damage and repair machinery at both heterochromatin and euchromatin[48]. In a provocative study, Bunch et al. have linked KAP1 to

the DNA damage response (DDR) during transcription elongation[47]. They reported that serum treatment in HEK293T cells induces γ-H2AX on chromatin, a marker of double-strand breaks (DSB), and site-specific KAP1 phosphorylation (pS824), which has been associated with DDR[48]. Despite they did not test how the loss of KAP1 affects IEGs expression upon serum treatment, as we have done using acute depletion, they utilized the evidence of co-occupancy of γ-H2AX and pS824-KAP1 to suggest that KAP1 regulates DNA damage at IEGs upon stimulation. It will thus be interesting to explore the contribution of DNA damage and site-specific KAP1 phosphorylation to transcription elongation control.

A second, important question to address in future research is: how and why do increases in Pol II elongation kinetics during early stimulation decrease downstream transcription activation during late stimulation? One model that we speculate is that increased elongation kinetics leads to Pol II pileups in the 3′ end, which may trigger a kinetic delay in Pol II termination and/or eviction from chromatin[73]. Decreased Pol II availability as a consequence of 3′ end retention may alter the proper kinetics of re-initiation, highlighting that early elongation control may have impacts on the next wave of transcription, including derailing the recruitment of initiation and elongation factors (Fig. 6b). This model is consistent with several previous observations documenting the impact of elongation on termination and termination on re-initiation. Disrupting elongation kinetics may alter 3′ pause sites and terminators, which has been shown to reduce transcription of the same gene[74]. Additionally, a plethora of data supports that promoters and terminators of genes are in juxtaposition[75–79] and that contacts between 5′ and 3′ ends are either dependent on transcription[80,81] or induced upon transcriptional signals[82]. Altogether, it is not unprecedented that affecting one transcription step (e.g., elongation kinetics) can lead to downstream transcription defects, and our work shows that decreasing the kinetics of IEGs transcription elongation is critical for their full activation across multiple transcription cycles.

Our study exposed yet another layer of precise elongation control that differentiates from previous mechanisms revealing that Pol II deceleration in the 3′ end is important for proper termination and avoidance of readthrough transcription[26,83,84]. In these studies, Pol II deceleration by protein phosphatase 1 (PP1) in the 3′ end was critical for the formation of the 3′ pause site and for enabling Xrn2 (the 5′−3′ RNA exonuclease) to catch up to Pol II for normal eviction after degradation of the exposed nascent RNA post-cleavage. In our dataset, we do not observe striking increases in readthrough Pol II upon KAP1 depletion, but rather an increase in "piled up" paused Pol II in the 3′ end (Fig. 3a), overall suggesting that 3′ pausing is established but that Pol II molecules reach the 3′ end at an earlier time when KAP1 is depleted, potentially owed to increased elongation rate in the gene body. Given that our phenotypes differ from observations with the PP1-Xrn2 mechanism, it appears KAP1 may not function with known regulators of Pol II deceleration in the 3′ end (e.g., PP1)[26]. However, an intriguing mechanism that is in line with the phenotypes that KAP1 loss elicits is potential regulation by R-loops, which are DNA:RNA hybrids formed at G-rich pause sites in gene bodies and 3′ ends[85,86]. Blocking the Senataxin (SETX) helicase that resolves R-loops[87] decreases Xrn2 occupancy at 3′ ends, suggesting that R-loop formation and resolution are important for transcription termination[88]. In their study, SETX silencing also increased paused Pol II in the 3′ ends with a concomitant decrease at 5′ ends, virtually phenocopying KAP1 depletion in our study. Thus, understanding the dynamics by which KAP1 potentially functions with the R-loop machinery (SETX/R-loops themselves) and/or termination machinery (Xrn2) directly or indirectly at signal-induced genes will be an interesting direction for future research.

Another question that remains to be addressed is why KAP1 specifically regulates transcription of IEGs, but not global transcription, like other factors do (e.g. Pre-initiation complex, SPT5, CDK9). While we tested how alterations to elongation kinetics at the earliest time points of stimulation impact the recruitment of global transcription regulators (e.g., SPT5/CDK9), we did not test if the recruitment of signal-specific machinery (e.g., pElk-1) is also perturbed, which could potentially partially explain the dependence of KAP1 on IEGs. Additionally, serum treatment induces localization of KAP1 to GB regions and 3′ ends of IEGs, suggesting that this specificity is at least in part mediated by chromatin events. However, how serum alters KAP1 occupancy on chromatin elements it interacts with (e.g., DNA, histones, Pol II itself) to activate IEGs transcription is an important area for future research.

Finally, our work does not explain why acute KAP1 depletion leads to modest upregulation of a small subset of IEGs in homeostatic conditions, and we also observed similar modest changes in the serum-starved condition (Supplementary Fig. 1c, d, h). Prior studies using RNAi have suggested that KAP1 represses genes in the basal condition by stabilizing paused Pol II prior to stimulation[63] and this could explain a repressor-activator switch mechanism where KAP1 is induced to promote activation (potentially by an elongation rate mechanism) upon stimulation. However, while we clearly observed Pol II occupancy changes upon serum stimulation, we did not find pause release upon acute KAP1 depletion in the serum-starved (basal) condition by either Pol II ChIP-Seq or PRO-Seq (Supplementary Figs. 3a and 4g), so it remains unclear why modest RNA inductions are observed prior to cell stimulation. One mechanism that may explain the upregulation is KAP1's role in controlling other processes (DNA repair or replication) that, if not timely resolved, may trigger stress responses known to induce IEGs expression[71,89]. Although exciting models to test, we focused this current study on evaluating KAP1's role in signal-induced transcription activation because of the clear observational differences when KAP1 was acutely depleted upon cell stimulation.

Collectively, our studies have helped clarify current conundrums revolving around the mechanism whereby KAP1 facilitates transcription activation. Overall, KAP1 plays a "repressive" role in maintaining Pol II elongation kinetics. Alterations in this regulatory mechanism at early stimulation time points elicit downstream consequences in the transcriptional cycle ultimately dampening signal-induced transcription. Our work importantly clarifies the roles of KAP1 in transcription and potentially links a long-standing KAP1 repressive function for transcription activation of genes important for cell fate choices.

## Limitations

We acknowledge that our studies have limitations that deserve to be discussed to help the reader illuminate gaps for future research.

1. KAP1 "direct" effects on Pol II transcription: Given the use of an acute KAP1 depletion system to minimize secondary effects, the observed signal-induced KAP1 recruitment to target genes and KAP1 traveling with elongating Pol II, as well as KAP1 direct interaction with Pol II[45], we consider the Pol II transcription defects upon KAP1 loss to be direct. However, the lack of KAP1-Pol II structural information makes it impossible to test the "direct" claim without the use of add-back experiments with either KAP1 wild-type or Pol II-binding-deficient KAP1 mutants. Given this intrinsic limitation, we have opted to tone down the wording in our study until structural and biochemical data become available to better assess the "direct" claim.

2. Analysis of Pol II elongation kinetics by Proxy Rate: Because of the way Proxy Rate measurements are made (due to the gene length limitations discussed), the actual value does not generate a single "leading edge" or distance that Pol II has traveled. Thus, we cannot extrapolate an approximate value for the increase in elongation rate when the KAP1 function is lost, but can only infer that the distance that Pol II has traveled when KAP1 is depleted is likely further into the gene bodies of IEGs.

3. Effects of acute KAP1 depletion on other transcription steps: The collected evidence supports a major effect of KAP1 depletion on

Pol II signal at gene bodies and at 3′ ends. The main question that arises is whether this effect is solely attributed to KAP1 direct regulation of elongation or whether other upstream steps also become altered. We did not observe major Pol II occupancy nor transcription alterations at promoter-proximal regions during early stimulation time points, consistent with the lack of defects at upstream transcription steps. Transcription initiation is highly dynamic and thus difficult to capture molecular phenotypes at that level. While our studies are limited by an inability to examine every single component of the transcription initiation apparatus, our data suggest that initiation is not subject to alterations. Pausing is a rate-limiting step and thus easier to pinpoint changes, but we have not seen any alterations in recruitment nor the activity of CDK9 at promoter-proximal regions. Finally, the contribution of premature termination to the outlined mechanism was not considered. Although we are not fully discounting potential effects during the conversion of paused Pol II into an elongation-competent form by reducing premature termination, we think this scenario is less likely because of the lack of Pol II signal perturbations at promoter-proximal regions. While we have provided evidence supporting alterations at the elongation level, future higher-resolution approaches might help distinguish and rule out effects at other levels.

## Methods

### Cell culture

HCT116 KAP1$^{dTAG}$ and HCT116 parental cells (to build the cell line) were cultured in Dulbecco's modified Eagle's medium (DMEM) (Gibco, catalog 11965118) supplemented with 7% Fetal Bovine Serum (FBS) (MilliporeSigma, catalog F4135) and 1% Penicillin/Streptomycin (P/S) (Gibco, catalog 15140163) at 37 °C with 5% CO$_2$. The original HCT116 parental cell line was purchased from ATCC (catalog CCL-247). Mycoplasma tests (SouthernBiotech, catalog 13100-01) were conducted every ~3–6 months.

### dTAG and serum treatments

dTAG-13 (Tocris, catalog 660520) was added at a final concentration of 500 nM for the indicated time points in each experiment with a corresponding DMSO control. For all serum inductions, cells were first seeded in full serum media (DMEM-7% FBS-1% P/S). Upon reaching 80% confluence, cells were washed with 1× DPBS (ThermoFisher Scientific, catalog 14190250), and then serum-free (DMEM-1% P/S) media was added to cells. DMSO/dTAG was added 40 h later followed by the addition of 18% serum-containing media (DMEM-18% FBS-1%P/S) for the time points indicated in each experiment.

### Cloning of dTAG constructs

For cloning of dTAG constructs as well as making the HCT116 KAP1$^{dTAG}$ cell line, published procedures were followed with minor modifications[90]. Briefly, the gRNA was designed from a combination of Benchling for off-target scores and the Washington University gRNA designer tool for structural scores (https://crisprdb.org/wu-crispr-website/index.html). The primers for both gRNA and arm cloning were designed based on this gRNA selection following the tutorial[90]. To clone the gRNA plasmid, custom-made DNA oligonucleotides (MilliporeSigma) were annealed with T4 ligase (NEB, catalog M0202S) and phosphorylated with T4 polynucleotide kinase (NEB, catalog M0201L) by incubating at 37 °C for 30 min, 95 °C for 5 min, and a ramp down to 25 °C at 5 °C/min. To prepare the sgRNA backbone vector, the empty universal cutting vector (pX330A-sgX-sgPITCh) was digested with BbsI-HF (NEB, catalog R0539S) for 1 h at 37 °C followed by a 15 min treatment with Quick CIP (NEB, catalog M0525S) and subsequent gel purification (QIAGEN, catalog 28606). Prepared oligonucleotides were then ligated into the digested backbone vector with T4 ligase at room temperature for 1 h followed by heat-inactivation and transformation

into home-made DH5α competent cells followed by plasmid preparation and sequence verification using the primers listed in Supplementary Table 2. To clone arm plasmids, the pCRIS-PITChv2-dTAG-Puro, and pCRIS-PITChv2-dTAG-BSD plasmids were digested with MluI-HF (NEB, R3198S) followed by gel purification to prepare the linearized vector. To prepare the arm cassette, the original arm plasmid was utilized as a template for PCR using the designed arm primers to insert the KAP1 C-terminal homology arms using Q5® Hot Start High-Fidelity DNA Polymerase (NEB, catalog M0493L). The ~1-kb product was purified and Gibson assembly was performed (50 °C, 30 min) using 1.5 μL of NEBuilder HiFi Master Mix (NEB, catalog E2621S) and the digested vector and PCR product. The Gibson product was diluted and transformed into homemade DH5α competent cells, and plasmid DNAs were prepared and verified by Sanger sequencing. All plasmids and primers used in this study are listed in Supplementary Tables 1 and 2, respectively.

### Creation of HCT116 KAP1$^{dTAG}$ cells

To generate HCT116 KAP1$^{dTAG}$ cells, we used a double selection approach in which both puromycin- and blasticidin-resistant dTAG cassette plasmids were transfected in tandem with a sgRNA plasmid. We selected cells with both antibiotics to target the two *KAP1/TRIM28* alleles and single-sorted for expansion followed by western blot to ensure no presence of the endogenous (lower molecular weight) KAP1 protein band. Briefly, HCT116 parental cells were seeded in 6-well plates (~500,000 cells/well) and transfected ~24 h post-seeding with three plasmids (BSD arm plasmid, Puro arm plasmid, and gRNA plasmid) targeting the C-terminus of KAP1 at a ratio of 1:1:1 (0.33 μg each). Before transfection, the media was changed into 1 mL of fresh DMEM. Two mixes were prepared: (1) a DNA mix consisting of 1 μg of total plasmid DNA (0.33 μg of each plasmid) into 100 μL of OPTIMEM (Gibco, catalog 11058021) and a Lipofectamine 2000 (ThermoFisher Scientific, catalog 52887) mix of 3 μL incubated in final 100 μL OPTI-MEM. The Lipofectamine 2000 mix was added dropwise to the DNA mix and incubated for 10 min at room temperature. After incubation, the 200 μL mix was added dropwise to cells, and the media was changed into 2 mL of DMEM media after 5 h. Two days post-transfection, selection media (1.5 μg/mL puromycin (ThermoFisher Scientific, catalog 227420500) and 10 μg/mL blasticidin (ThermoFisher Scientific, catalog BP264750) in complete DMEM) was added to the cells for 14 days to induce targeting of both *KAP1/TRIM28 alleles*, at which point stable colonies began to expand. A single-selection approach including 0.5 μg of only PURO arm plasmid and 0.5 μg of gRNA plasmid was included as control. Cells were re-split in a 6-well plate, grown for 4 days, and then expanded to a 10-cm tissue culture dish. Population lysates were collected and subjected to KAP1 western blot to verify targeting efficiency. Cells were single-cell sorted in 96-well plates at the UTSW Children's Research Institute Moody Foundation Flow Cytometry Core in complete DMEM (200 μL per well with no antibiotic). Cells grew in single-cell format for ~4–5 weeks prior to slowly expanding to 10-cm tissue culture dishes (24-well first, 6-well second, and then finally to 10-cm dishes). To verify *KAP1* targeting, anti-KAP1 western blots were performed.

### Western Blots

All western blots were run on 10–12% SDS-PAGE gels and transferred on nitrocellulose membranes (Bio-Rad, catalog 1620115) using the Bio-Rad Trans-Blot Turbo Transfer System, blocked for 1 h in 5% Milk (or BSA) + Tris-buffered saline-Tween-20 (TBST), probed with primary antibody (see Supplementary Table 3) in 5% Milk (or BSA) + TBST. Primary and secondary antibody concentration and time of incubation are indicated in Supplementary Table 3. Blots were exposed using either Clarity Western ECL (Bio-Rad, catalog 1705060) or SuperSignal™ West Femto Maximum Sensitivity Substrate (ThermoFisher Scientific, catalog 34095) or utilizing the starbright secondary channel.

## Cell confluence assay

For the cell growth assay, cells were treated with either 8 h DMSO or dTAG and then the IncuCyte S3 Live Cell Imaging System (Essen Bioscience) was used for cell proliferation assays. 2000 cells/well were seeded into 96-well plates and imaged every 6 h over 144 h (6 days). Phase contrast images were used to calculate cell confluence using the IncuCyte software and the cell confluence at each time point was normalized to time 0.

## RNA extraction and RT-qPCR

For RT-qPCR, cells were treated as described and RNA was extracted using the Zymo Quick-RNA MiniPrep Kit (Zymo Research, catalog R1055) following the kit instructions. RNA was quantified using the DeNovix DS-II FX+ Spectrophotometer and all RNAs were diluted to the same yield and re-quantified prior to cDNA synthesis reaction. To prepare cDNA, 2 μg of RNA was incubated with 1/200th of a unit of hexanucleotide primers (MilliporeSigma, catalog H0268-1UN) and 1 μL of 10 mM dNTP mix (NEB, catalog N0447L) for 5 min at 70 °C. Next, 2 μL of 10× M-MuLV Reverse Transcriptase buffer and 1 μL of M-MuLV Reverse Transcriptase (NEB, catalog M0253L) were added to each sample (final volume 20 μL) and incubated at 42 °C for 1 h. The reaction was inactivated at 70 °C for 10 min and samples were diluted by adding 80 μL with $H_2O$. For qPCR, 1 μL of diluted cDNA, 5 μL PowerUp™ SYBR™ Green Master Mix for qPCR (ThermoFisher Scientific, catalog A25741), 3 μL of $H_2O$, and 1 μL of 5 μM primer mix was used for each well in a 96-well plate. All primers used for RT-qPCR analysis are listed in Supplementary Table 2. Samples were amplified for 40 cycles using the Applied Biosystems QuantStudio™ 3 Real-Time PCR System. All RT-qPCR data was analyzed using the ΔΔCt method where each RNA for each sample was normalized to the DMSO-Serum 0 min (S0) sample and then each target gene was normalized to two control genes (*RPL19* and *GAPDH*) using the geometric mean of their expression.

## RNA-Seq

RNAs for RNA-Seq were prepared as described for RT-qPCR. For RNA-Seq library preparation, 1 μg of RNA was used as input with 2 μL of a 1:100 dilution of ERCC RNA Spike-In Mix (ThermoFisher Scientific, catalog 4456740) and the KAPA RNA Hyper+RiboErase HMR kit was used (Roche, catalog 8098131702) following the manufacturer's instructions for library preparation using KAPA beads (Roche, catalog KK8001). The following steps were performed: Oligo hybridization and rRNA depletion, KAPA bead cleanup, digestion with DNAse, and another KAPA bead cleanup as described in the kit protocol. RNA elution, fragmentation (8 min at 94 °C), and priming were performed followed by first-strand synthesis and second-strand synthesis/A-tailing. Adapter ligation (15 min at 20 °C) was performed with 7 μM NEBNext® Adaptor for Illumina® (NEB, kit catalog E6446L) followed by a 3 μL USER enzyme digest (15 min at 37 °C). This was immediately followed by two cleanups (first with 0.63× KAPA beads and second with 0.7× PEG/NaCl solution). After adapter ligation, samples were PCR amplified using Illumina indexed primers (NEB, E7335L) for 8 cycles. Following a final KAPA bead purification and elution with 20 μL of 10 mM Tris-HCl pH = 8.0, size distribution and quality of libraries was determined using the Agilent Tapestation DNA ScreenTape (Agilent, catalog 5067-5582). Libraries were then quantified using the Qubit dsDNA HS Assay (ThermoFisher Scientific, catalog Q32851) and sequenced at ~33E6 paired-end reads/sample with 50-bp length using a NextSeq 500 (Illumina) at the UT Southwestern McDermott Center Sequencing Core. Three biological replicates per treatment condition were submitted.

## TT-Seq

For TT-Seq, the protocol from Patrick Cramer's lab was followed with minor modifications[91]. Three 10-cm tissue culture dishes per condition were seeded such that cells reached 75% confluence 3 days post-

seeding. On the third day, cells were treated with DMSO or dTAG for 8 h, after which 500 μM 4-Thiouridine (MilliporeSigma, catalog T4509) was added to the media for 10 min. An extra dish was seeded per condition and ~12E6 cells were counted per plate and thus 36E6 cells/ sample were used to prepare RNA. Cells were washed with 5 mL of 1× PBS once and then immediately 1 mL of TRIzol™ Reagent (ThermoFisher Scientific, catalog 15596026) was added and followed by 10 min of rocking. Cells were scraped off the plate, transferred to a 15 mL tube, and then homogenized by pipetting ~10 times. Cells were placed at −80 °C until ready for RNA extraction. To isolate total RNA, 200 μL chloroform was added to the 1 mL of sample (3 tubes per replicate) and vortexed 15 s followed by centrifugation (13,000 × *g*, 15 min, 4 °C). The upper, aqueous layer (~490 uL) was added to a fresh tube with 1 μL Glycoblue and an equal volume of Isopropanol (~490 μL) followed by mixing, incubation at room temperature for 10 min, and then centrifugation (13,000 × *g*, 10 min, 4 °C). The pellet was washed twice with 1 mL 75% Ethanol followed by centrifugation (7500 × *g*, 5 min, 4 °C). After drying, the pellet was resuspended in 320 μL RNAse-free water per replicate, quantified by nanodrop, and diluted to 750 ng/μL followed by sonication using a Q800R3 Qsonica (50% amplitude with 3 cycles 30 s ON, 30 s OFF at 4 °C). The RNA was denatured at 65 °C for 10 min, placed on ice for 5 min, and diluted to 150 μg/700 μL to prepare for biotinylation. 10× Biotinylation buffer (100 mM Tris-HCl pH = 7.5, 10 mM EDTA pH = 8.0) was prepared along with 5× (1 mg/mL) EZ-Link™ HPDP-Biotin (MilliporeSigma, catalog 21341) in N,N-Dimethylformamide (Acros Organics, catalog 279600010). 300 μgs total RNA was used per replicate, and thus each condition had 2 × 1000 μL biotinylation reactions. A mix of Biotinylation Buffer (100 μL per reaction) and Biotin-HPDP was made (200 μL per reaction) and added to 700 μL of diluted RNA, the samples were mixed and then nutated for 2 h at room temperature while covered in foil. To purify RNA after biotinylation, 500 μL of each reaction was transferred to a new tube followed by the addition of an equal volume of chloroform (500 μL). Samples were vortexed, incubated for 3 min, and then centrifuged (1500 × *g*, 5 min, 4 °C), followed by transfer of the aqueous layer (~300 μL) to 2 new tubes per sample (~600 μL each) and then precipitation with 1/10th volume of 5 M NaCl (~60 μL), 1 μL glycoblue, and 1× isopropanol (~600 μL). Samples were vortexed, centrifuged (13,000 × *g*, 30 min, 4 °C), washed two times with 75% ethanol, and then resuspended in 500 μL of RPB buffer (10 mM Tris-HCl pH = 7.5, 1 mM EDTA pH = 8.0, 300 mM NaCl). To isolate biotinylated RNA, 200 uL of Dynabeads MyOne Streptavidin T1 (ThermoFisher Scientific, catalog 65601) was aliquoted per sample and washed 4 times with 1 mL beads wash buffer (10 mM Tris-HCl pH = 7.5, 1 mM EDTA pH = 8.0, 50 mM NaCl) and finally suspended in 1 mL beads wash buffer per sample plus 0.1% polyvinylpyrrolidone (ThermoFisher Scientific, catalog BP431-100). Beads were then nutated at room temperature for 10 min, washed one time with 1 mL beads wash buffer, and then suspended in RPB (200 μL per reaction). The biotinylated RNA product was denatured at 65 °C for 5 min, placed on ice for 2 min, and then 200 μL of blocked beads to each sample was added followed by a 30 min incubation with rotation at room temperature. After 30 min, beads were washed 5 times with 4sU wash buffer (10 mM Tris-HCl pH = 7.5, 1 mM EDTA pH = 8.0, 1 M NaCl, 0.1% Tween-20). Biotinylated, 4sU-RNA was eluted from beads twice by adding 75 μL of 0.1 M DTT to beads and nutating for 15 min at room temperature, yielding a final elution volume of ~150 μL. 4sU-RNA was purified and concentrated using the Zymo RNA Clean and Concentrator Kit (Zymo Research, catalog R1013) following kit instructions and RNA was quantified using the Qubit™ RNA High Sensitivity (HS) Assay (ThermoFisher Scientific, Q32852). Libraries were prepared similarly to RNA-Seq except with the minor modifications described here. First, 2 μL of a 1:4000 ERCC Spike-in mix was added to approximately ~50 ngs of nascent RNA as input to library preparation. Fragmentation was done for 6 min at 94 °C and 13 cycles of PCR were used for PCR amplification step.

Libraries were quality-checked with Tapestation and Qubit and then sequenced at ~100E6 paired-end reads/sample with 50-bp length using a NextSeq 500 (Illumina) at the UT Southwestern McDermott Center Sequencing Core. Two biological replicates per treatment condition were submitted.

## ChIP-Seq

All ChIP-Seq experiments were performed as previously described[45] with minor modifications. For each ChIP, a 1 × 15-cm tissue culture dish (yielding ~30E6 cells/plate) was utilized and seeded according to the experimental design in each figure (DMSO/dTAG ± serum at different time points), and 20E6 cells were utilized per ChIP. For all Pol II ChIPs (RPB3, Ser5P Pol II, Ser2P Pol II), cells were crosslinked with 0.5% methanol-free formaldehyde (ThermoFisher Scientific, 28908) by adding directly to the tissue culture dish in media at room temperature for 10 min with rocking and neutralization with 150 mM glycine for 5 min with rocking. For all other ChIP-Seq experiments (HA (KAP1), SPT5, pSPT5, CDK9, CDK7, TFIIB, MED1), 1% formaldehyde for 10 min was used. For all experiments, the media was removed and cells were washed twice with cold 1× PBS, cold PBS was then added to the plate and then cells were collected by scraping. Cells were centrifuged (1000 × g, 5 min, 4 °C), and pelleted, flash frozen in liquid nitrogen, and frozen at −80 °C until ready to use. To perform the ChIP, cells were resuspended in 4 mL/dish of Farnham Lysis Buffer (5 mM PIPES pH = 8.0, 85 mM KCl, 0.5% NP-40, 1 mM PMSF, 1× Protease Inhibitor (RPI, catalog P50900-1)), counted by hemocytometer, resuspended to 10E6 cells/mL, nutated for 30 min at 4 °C, and then centrifuged to isolate nuclei (1000 × g, 5 min, 4 °C). The supernatant was removed and nuclei resuspended in Szak's RIPA Buffer (50 mM Tris-HCl pH = 8.0, 1% NP-40, 150 mM NaCl, 0.5% Na-Deoxycholate, 0.1% SDS, 2.5 mM EDTA pH = 8.0, 1 mM PMSF, and 1× Protease Inhibitor) at a concentration of 25E6 nuclei/mL. The chromatin was sheared using a Q800R3 Qsonica (50% amplitude with 25 cycles 30 s ON, 30 s OFF at 4 °C) to a DNA molecular weight range of 200–400-bp. After sonication, chromatin was centrifuged (21,000 × g, 15 min, 4 °C) and the supernatant was taken as clarified chromatin. For ChIP-Seq experiments that contained Drosophila spike-in, 50 ng of spike-in chromatin (Active Motif, catalog 53083) was added to each chromatin sample. Sheared chromatin was pre-cleared by incubating with 25 μL of Szak's RIPA equilibrated Protein G Dynabeads (ThermoFisher Scientific, 10003D) for 1 h at 4 °C. To equilibrate beads for pre-clearing, a master mix of beads was incubated with RIPA buffer, nutated for 5 min, and placed on a magnet to remove supernatant three times. To bind antibody to beads, 100 μL of Protein G Dynabeads per sample were equilibrated with 1× PBS + 0.05% Tween-20 and resuspended to a final volume of 250 μL per ChIP. The corresponding antibody (see Supplementary Table 3 for antibody details) was then added to the 250 μL beads, and the bead-antibody mix was nutated for 1 h at 4 °C. For samples that contained Drosophila spike-in, 2.5 μg of antibody specific for Drosophila H2Av (Active Motif, catalog 61686) was added to each tube. Antibody bound beads were then blocked in Szak's RIPA Buffer + 5% BSA for 1 h at 4 °C with rotation. Pre-cleared sheared chromatin was then added to beads and incubated overnight at 4 °C with rotation. Beads from each sample were washed 2 times with 900 μL of Szak's RIPA Buffer, Low-Salt Buffer (0.1% SDS, 1% NP-40, 2 mM EDTA pH = 8.0, 20 mM Tris-HCl pH = 8.0, 150 mM NaCl), High-Salt Buffer (0.1% SDS, 1% NP-40, 2 mM EDTA pH = 8.0, 20 mM Tris-HCl pH = 8.0, 500 mM NaCl), LiCl buffer (250 mM LiCl, 1% NP-40, 1% sodium deoxycholate, 1 mM EDTA pH = 8.0, 20 mM Tris-HCl pH = 8.0), and TE Buffer (10 mM Tris-HCl pH = 8.0, 1 mM EDTA pH = 8.0). After the final wash, samples were pulse spun in a table-top centrifuge to get rid of residual buffer before placing it on the magnet. Samples were then eluted from beads in 100 μL of elution buffer (100 mM NaHCO3 pH = 8.0, 1% SDS) for 30 min at 65 °C while vortexing every ~10 min. Input samples (40 μL) were volumed up to 100 μL by adding 60 μL elution buffer. DNA was eluted by placing the beads on a magnet, elutions transferred to new tubes, and de-crosslinked for 4 h at 65 °C with 100 μL volume of de-crosslinking buffer (500 mM NaCl, 2 mM EDTA pH = 8.0, 20 mM Tris-HCl pH = 6.8, 0.5 mg/mL Proteinase K (Epicentre, catalog MPR-90938)). ChIP DNA was purified and concentrated with the Zymo ChIP DNA Clean & Concentrator (Zymo Research, catalog D5201). ChIP samples were quantified by Qubit, and libraries were prepared using the KAPA Hyper Prep Kit (KAPA Biosystems, catalog KK8502) following the manufacturer's instructions. Briefly, samples underwent End Repair & A-tailing followed by adapter ligation with 300 nM to 1.5 μM NEB adapter depending on initial yield. Following a post-ligation cleanup, PCR amplification was performed with anywhere between 8-14 total cycles depending on initial yield. A KAPA bead cleanup was performed (1×) followed by size selection. For size selection, the first cleanup utilized 35 μL KAPA beads where the final supernatant contains the DNA of interest. The second and final cleanup utilized 10 μL of beads where the beads contain the bound, desired DNA. Following elution with 20 μL of 10 mM Tris-HCl pH = 8.0, libraries were quality-checked with Tapestation and Qubit, and then sequenced at ~33E6 paired-end reads/sample with 50-bp length using a NextSeq 500 (Illumina) at the UT Southwestern McDermott Center Sequencing Core. Two biological replicates of each ChIP-Seq for HA, Pol II, and Ser2P Pol II were completed, while one replicate of SPT5, pSPT5, CDK9, CDK7, TFIIB, MED1, and Ser5P Pol II were performed in each condition.

## PRO-Seq

For PRO-Seq experiments, we followed the qPRO-Seq protocol[92] with minor modifications. Cells were seeded in 10-cm tissue culture dishes and treated as indicated and were ~90% confluent prior to collection. For each PRO-Seq, 4E6 cells were utilized. To prepare permeabilized cells for run-on, cells were first washed twice on the plate with 5 mL of ice-cold 1× PBS. Then, 2.5 mL of ice-cold Cell Permeabilization Buffer (CPB) (10 mM Tris-HCl pH = 8.0, 250 mM Sucrose, 10 mM KCl, 5 mM MgCl2, 1 mM EGTA pH = 8.0, 0.1% NP-40, 0.5 mM DTT, 0.05% Tween-20, 0.1% Triton X-100, 10% Glycerol, 1× protease inhibitor, and 2 μL SUPERase-In RNase inhibitor (ThermoFisher Scientific, catalog AM2696) per 10 mL) was immediately added to the tissue culture dish. Triton X-100 was added to CPB to increase permeabilization. The tissue culture dishes were placed on ice and then immediately scraped with a cell scraper. At this point, cells were collected in a 15 mL tube, placed on ice for 5 min, checked for permeabilization by trypan blue staining (>95%), and then centrifuged in a swinging bucket rotor (1000 × g, 4 min, 4 °C). Following this, cells were handled using cut tips. The supernatant was removed and samples were resuspended in 1 mL of Cell Wash Buffer (CWB) (10 mM Tris-HCl pH = 8.0, 250 mM Sucrose, 10 mM KCl, 5 mM MgCl2, 1 mM EGTA pH = 8.0, 0.5 mM DTT, 10% Glycerol, 1× protease inhibitor, and 2 μL SUPERase-In RNase inhibitor per 10 mL) and centrifuged (1000 × g, 4 min, 4 °C) for a total of 2 washes. After the second wash, samples were resuspended in 1 mL total Cell Freeze Buffer (CFB) (50 mM Tris-HCl pH = 8.0, 5 mM MgCl2, 0.5 mM DTT, 40% Glycerol, 1.1 mM EDTA pH = 8.0, and 2 μL SUPERase-In RNase inhibitor per 10 mL), counted by hemocytometer, and then spun down in 1.5 mL tubes in an angled rotor centrifuge (1000 × g, 5 min, 4 °C). Samples were resuspended in 52 μL of CFB for every 4E6 cells, flash frozen in liquid nitrogen, and then frozen at −80 °C until ready to use. For run-on assays, two biotinylated NTP's (UTP and CTP, PerkinElmer, catalog NEL543001EA and NEL542001EA) were used with unbiotinylated ATP and GTP (MilliporeSigma, catalog 11277057001) and 2× ROMM buffer (10 mM Tris-HCl pH = 8.0, 5 mM MgCl2, 1 mM DTT, 300 mM KCl, 40 μM Biotin-11-CTP, 40 μM Biotin-11-UTP, 40 μM ATP, 40 μM GTP, 1% Sarkosyl (MilliporeSigma, catalog L5125) and 1 μL SUPERase-In RNase inhibitor per reaction) was prepared exactly as recommended. 50 μL of pre-heated 2× ROMM buffer was added to 50 μL of cell suspension, pipetted with a cut 200 μL tip 20–25 times quickly, and incubated (with 700 RPM shaking) for 5 min, after which

250 µL of Trizol LS (ThermoFisher Scientific, catalog 10296028) was immediately added, mixed by pipetting, vortexed, and placed on ice. 65 µL chloroform was added to each sample, vortexed, incubated on ice for 3 min, and centrifuged (20,000 × g, 8 min, 4 °C). Approximately ~150 µL of the aqueous phase was collected in a new tube followed by addition of 1 µL Glycoblue (ThermoFisher Scientific, catalog AM9515) and 2.5× volumes of 100% ethanol, and samples were then vortexed for 5 s and centrifuged (20,000 × g, 20 min, 4 °C). The supernatant was removed, washed with 75% ethanol once with gentle inversion and pulse spun. RNA pellets were airdried followed by resuspension in 30 µL of RNAse-free water. RNA was denatured for 30 s at 65 °C, snap-cooled on ice, and then fragmented with 7.5 µL of cold 1 M NaOH on ice for 10 min. 75 µL of 0.5 M Tris-HCl pH = 6.8 was added and mixed by pipetting, and then samples were passed through a calibrated Micro Bio-Spin™ P-30 Gel Columns, Tris Buffer (RNase-free) (Bio-Rad, catalog 7326250) following the manufacturer's instructions. The volume of each sample was brought up to 200 µL with water, and 1 uL glycoblue, 8 µL NaCl, and 500 µL 100% ethanol were added, vortexed, and then centrifuged (20,000 × g, 20 min, 4 °C). Ethanol was removed and RNA pellets were stored in −80 °C overnight. The next day, 75% ethanol was used to wash the pellet, spun down, and then pellets were resuspended in 6 µL of RNAse-free water. 1 µL of 10 µM VRA3 oligo (see Supplementary Table 2) was added to the RNA and samples were denatured for 30 s at 65 °C and snap-cooled on ice. A buffer containing T4 RNA Ligase 1 (ssRNA ligase) (NEB, catalog M0204L) was added to samples for ligation for 1 h at 25 °C. 10 µL/sample of Dynabeads™ MyOne™ Streptavidin C1 Beads (ThermoFisher Scientific, catalog 65001) were then equilibrated by removing storage buffer by placing on a magnet, washed once in 1 mL bead preparation buffer (0.1 M NaOH and 50 mM NaCl) and twice with 1 mL binding buffer (10 mM Tris-HCl pH = 7.5, 300 mM NaCl, 0.1% Triton X-100, 1 mM EDTA pH = 8.0, and 2 µL SUPERase-In RNase inhibitor per 10 mL). Each wash was done by adding the indicated buffer, flipping the tube on the magnet twice, and then removing the buffer without letting the beads dry. Beads were resuspended in 25 µL binding buffer per sample and placed on ice until ready to use. After 3′ adapter ligation, 55 µL of binding buffer and then 25 µL of beads were added to each sample and rocked for 20 min at room temperature to bind biotinylated, nascent RNA to the beads. Beads were washed with 500 µL High-Salt Buffer (HSB, 50 mM Tris-HCl pH = 7.5, 0.5% Triton X-100, 2 M NaCl, 1 mM EDTA pH = 8.0 and 2 µL SUPERase-In RNase inhibitor per 10 mL), then 500 µL Low-Salt Buffer (LSB, 5 mM Tris-HCl pH = 7.5, 0.1% (v/v) Triton X-100, 1 mM EDTA pH = 8.0 and 2 µL SUPERase-In RNase inhibitor per 10 mL), and then resuspended in a buffer containing T4 polynucleotide kinase (NEB, catalog M0201L) and incubated for 30 min at 37 °C. Samples then underwent 5′ DeCapping using a buffer containing RppH enzyme (NEB, catalog M0356S) for 1 h at 37 °C, which was followed by 5′ adapter ligation on-beads (VRA5 oligo (see Supplementary Table 2) was added to each sample and then RNA-bound beads were denatured as described for 3′ adapter ligation) for 1 h at 25 °C. After 5′ ligation, the beads were washed with both HSB and LSB, resuspended in 300 µL of Trizol, vortexed, and incubated on ice for 3 min. 60 µL chloroform was added to each sample, vortexed, incubated on ice for 3 min, and then centrifuged (20,000 × g, 8 min, 4 °C). Approximately ~180 µL of the aqueous phase was collected in a new tube followed by an addition of 1 µL Glycoblue and 2.5× volumes of 100% ethanol. Samples were then vortexed and centrifuged (20,000 × g, 20 min, 4 °C). The supernatant was removed, washed with 75% ethanol once with gentle inversion and pulse spin, and RNA pellets were airdried followed by resuspension in 13.5 µL Reverse Transcriptase (RT) resuspension mix to begin cDNA synthesis, which consisted of 4 µL of 10 µM RPI oligo (see Supplementary Table 2) and dNTPs. Samples were denatured at 65 °C for 5 min, placed on ice, and resuspended in an RT master mix consisting of Maxima H Minus RT enzyme (ThermoFisher Scientific, catalog EP0752), and cDNA synthesis was performed on a thermal cycler (50 °C

for 30 min, 65 °C for 15 min, and 85 °C for 5 min). 2.5 µL of a 10 µM RPI-X (designated RPI indexing primer) was added to each cDNA synthesis reaction followed by 78.5 µL of PCR amplification master mix (consisting of Q5® High-Fidelity DNA Polymerase) and PCR was performed with a total of 14 cycles following the conditions (56 °C extension) in the original protocol. The samples were purified using 180 µL of KAPA pure beads, and eluted in 15 µL of 10 mM Tris-HCl pH = 8.0. Libraries were quality-checked with Tapestation and Qubit and then sequenced at ~66E6 paired-end reads/sample with 50-bp length using a NextSeq 500 (Illumina) at the UT Southwestern McDermott Center Sequencing Core. If samples had excess adapter dimer contamination, they were subsequently electrophoresed on a 2% agarose gel (80 V for 60 min) followed by gel excision in a cold room, purification by column (QIA-GEN, catalog 28606), and additional TapeStation and Qubit analyses before sequencing. Two biological replicates per treatment condition were submitted.

## RNA-Seq and TT-Seq data analysis
Detailed scripts for pre-processing and downstream analysis, tutorials, and files needed to run the analysis for this paper are under our lab GitLab page. Briefly, raw fastq data files were run through fastqc/0.11.8 and low-quality reads/adapter contaminations were removed using trimgalore/0.6.4. Reads were mapped to the hg38 human reference genome using star/2.7.3a. Spike-ins were mapped to a fasta file of ERCC sequences provided by the manufacturer. featureCounts (subread/1.6.3) was used to calculate counts across the entire gene of protein-coding genes using a .gtf file containing all protein-coding genes as well as ERCC spike-ins. EDASeq/2.32.0, RUVSeq/1.32.0, and EdgeR/3.40.1 commands were used for spike-in normalization and differential gene expression analysis of genes that contained at least 10 counts in 6 samples (RNA-Seq) and 50 counts in 4 samples (TT-Seq) was performed. Differential expression analysis files are provided as a table in Supplementary Data labeled 'DEG Analysis File'. IEGs were identified by $Log_2FC$ values between DMSO-Serum Starved (SS) and DMSO-Serum (S). $Log_2FC$ values for each cluster are described by the following: 2-fold IEGs ($Log_2FC$ > or = 1.0, $n$ = 236), 4-fold IEGs ($Log_2FC$ > or = 2.0, $n$ = 69), 8-fold IEGs ($Log_2FC$ > or = 3.0, $n$ = 34), 16-fold IEGs ($Log_2FC$ > or = 4.0, $n$ = 16). non-DE genes ($n$ = 10,361) were identified as genes that were less than 1.4-fold up or downregulated and were expressed by parameters described above. Volcano plots were made using EnhancedVolcano/1.16.0 from Bioconductor. Pathway analysis was performed using Enrichr[93].

## ChIP-Seq analysis
Detailed scripts, tutorials, and input files needed to run all analyses for ChIP-Seq analysis for this paper are under our lab GitLab page. Briefly, raw fastq data files were run through fastqc/0.11.8 and low-quality reads/adapter contaminations were removed using trimgalore/0.6.4. Reads were mapped to the hg38 human reference genome using bowtie2/2.4.2. Reads were also mapped to the dm6 Drosophila genome to extract reads originating from the spike-in chromatin for ChIPs that were normalized using spike-ins. Duplicates were marked and removed using picard/2.10.3 and then files were sorted and indexed using samtools/1.6 in preparation for bigWig generation. Normalized bigWig's were made from sorted bam files using the bamCoverage command using deepTools/2.3.5[94]. The following ChIP-Seqs were normalized using Drosophila spike-ins: HA, SPT5, pSPT5, CDK9, CDK7, TFIIB, and MED1. The following ChIP-Seqs were normalized to read depth (counts per millions or CPM using deepTools): Pol II, Ser5P Pol II, Ser2P Pol II. For spike-in normalization, scaling factors (Supplementary Table 5) were defined using deepTools command multiBamSummary and then inputted into bamCoverage for bigWig generation for normalization. Importantly, all spike-in normalized datasets were compared to read-depth normalized datasets to ensure quality and agreement. Ser5P Pol II and Ser2P Pol II ChIP-Seq data were normalized

to Pol II ChIP-Seq data using deepTools command bamCompare. Detailed alignment statistics for both human genome reads and spike-in coverage are included in Supplementary Table 5. The Integrated Genome Viewer (IGV) was used for visualization.

### PRO-Seq analysis and proxy rate analysis

Scripts for running PRO-Seq analysis and performing the Proxy Elongation Rate analysis using the ROCC method are under our lab GitLab page. The PRO-seq libraries were analyzed using the Proseq2.0 pipeline[95]. Briefly, raw fastq data files were run through fastqc/0.11.8 and low-quality reads/adapter contaminations were removed using Cutadapt/2.5. Reads were subsequently aligned to the human reference genome hg38 using BWA/0.7.5. The aligned bam files were converted into RPKM-normalized bigWig format using deepTools/2.3.5[94] and bedGraphToBigWig[96] program to visualize in IGV. For the Proxy Rate analysis, a set of ROCC's for each 50-bp bin in the set of IEGs is estimated for the 0, 5, and 10 min PRO-Seq data (DMSO/dTAG) utilizing bigWig files that contained the normalized read counts (coverage). ROCC benchmarking, performed as described for the serum dataset in this paper, was done using a previous estrogen-stimulated dataset in MCF7 cells[23]. A generalized linear model was applied to the overall intensities in the datasets to determine the slope of each condition at each 50-bp bin across the three time points (the ROCC value). Each 50-bp bin ROCC value for each gene is averaged to form a final Proxy Rate value. The generalized linear model was done by fixing the y-intercept to be 0 to decrease the influence of the serum 0 time point, which often contained low or no signal affecting the final rate calculations. To ensure that this did not drastically affect the analysis, a secondary Proxy Rate analysis was conducted by taking the normalized measure of the coverage at every 50-bp bin for the 0 min time point and subtracting this from both the 5 and 10 min points separately, and then dividing by 5 and 10, respectively. This analysis allows the serum 0 time point to exert maximal influence and yielded similar results. All analysis was completed using the R programming language, and custom scripts that utilized the 'rtracklayer' library[97] for interacting with the bigWig files. Benchmark validation of the ROCC method was done by comparison to the leading edge methodology followed by three common procedures, including Pearson's correlation (the standard measure assuming normality), Spearman's rank correlation (a non-parametric procedure that rank transforms the data and then performs regression), and Kendalls Tau (a non-parametric method which calculates the number of pairs that are in concordance, that is both x increases and y increases).

### Graphing and statistical analysis

All bar graphs, boxplots (Tukey plots), violin plots, and XY plots were created using GraphPad Prism v10. Statistical tests (two-sided unpaired Student's *t*-test and paired Wilcoxon Signed-Rank Test) defined in each figure legend were also performed in GraphPad Prism v10.

### Metagene analysis and quantifications

Scripts for metagene analysis and quantifications along with tutorials and our custom BED files are provided on the D'Orso lab GitLab page. RNA-Seq data was used to define and cluster IEGs. Metagenes were created using deepTools commands computeMatrix and plotProfile. Quantitation of signal was computed using computeMatrix. Promoter Proximal (PP) is defined as −100/+500-bp of TSS, Gene Body (GB) is defined as +500-bp of TSS to the pA site and 3′ end is pA site to 3000-bp downstream of each gene. Gencode annotation of Transcript Start Site (TSS) and Transcript End Site (TES) regions was used for preliminary analysis. However, after assessment of gene browser tracks of Pol II ChIP-Seq data, a custom BED file was created that more accurately represented the position of TSS and TES based on the criteria below, which is thoroughly explained in our GitLab. For TES,

many genes had a Gencode TES site designation, which was well before the putative pA site. Thus, we used NCBI Gene reference sequences to define pA sites with integration of Pol II ChIP-Seq and Ser2P Pol II ChIP-Seq data to define the pA site for analysis. Additionally, Gencode annotation revealed a few IEGs that had the incorrect start site usage (e.g. mostly due to alternative start sites not utilized in HCT116) and thus a TSS-Seq repository (https://dbtss.hgc.jp/#kero:chr2:96145427-96145452:-) was used to more closely define putative TSS sites. Given that this designation was primarily manual, Gencode TSS and TES were used for non-DE metagene profiles. For HA, Pol II, and Ser2P Pol II ChIP-Seq data, two biological replicates were analyzed independently until the bigWig stage and were merged after visual inspection of individual browser tracks in IGV and Pearson correlation analysis using deep-Tools (see Supplementary Table 4 for Pearson correlation coefficients). All PRO-Seq experiments contained two biological replicates that were analyzed independently until the bigWig stage and were merged after visual inspection and correlation analysis (see Supplementary Table 4 for Pearson correlation coefficients). One replicate of each antibody in each condition was performed for the following ChIP-Seqs: SPT5, pSPT5, CDK9, CDK7, TFIIB, MED1, and Ser5P Pol II.

### Reporting summary

Further information on research design is available in the Nature Portfolio Reporting Summary linked to this article.

## Data availability

Original western blot data generated in this study have been deposited in the Mendeley database (https://data.mendeley.com/datasets/ffsbjbkkvx/1). Raw NGS data generated in this study have been deposited at NCBI GEO under accession number GSE246218. Source data are provided with this paper.

## Code availability

Detailed scripts and commands used for all NGS analysis in this paper have been deposited in the D'Orso Lab public GitLab (https://git.biohpc.swmed.edu/ivandorsolab/hyder-et-al._-2024).

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

## Acknowledgements

We thank I. Cuevas for assistance with the IncuCyte, A. Shukla for conducting preliminary TT-Seq analysis, and A. Diehl for revising the model

in Fig. 6. We thank N. Conrad, M. Conacci-Sorrell, and L. Banaszynski for feedback throughout the project. The research reported in this publication was largely supported by NIAID (R01AI114362) and NCI (R03CA259672) grants to I.D. and an NCI grant (F99CA264296) to U.H. W.L.K. was supported by an NIDDK grant (R01DK058110) and funds from the Cecil H. and Ida Green Center for Reproductive Biology Sciences Endowment. We are indebted to the support of the UTSW Simmons Comprehensive Cancer Center (P30CA142543), UTSW Lung Cancer SPORE (P50 CA070907) and UTSW Kidney Cancer SPORE (P50CA196516).

## Author contributions

U.H. and I.D. conceived the study and designed the experiments. U.H. and A.C. performed the experiments. U.H. performed most of the computational analysis with assistance in PRO-Seq analysis from M.T. (who developed the proxy elongation rate analysis), T.N. (who helped analyze the PRO-seq data, including generating bigWig tracks), and W.L.K (who provided suggestions for PRO-seq and proxy elongation rate analyses and supervised M.T. and T.N.). U.H. wrote the primary draft and I.D. edited the manuscript. All authors reviewed the manuscript prior to submission. I.D. secured funding.

## Competing interests

The authors declare no competing interests.
