## [Peer Review File · Nature Communications]

REVIEWER COMMENTS

Reviewer #1 (Remarks to the Author):

The manuscript by Hyder et al describes KAP1/TRIM28 as a regulator of elongation that acts selectively on induced transcription. By comparing transcription induced by serum re-induction after starvation in normal HCT116 cells and cells with degran-depleted KAP1, authors propose that KAP1 deficiency leads to redistribution of Pol II that is apparently consistent with altered elongation kinetics. KAP1 is necessary for full activation and is distributed to gene bodies with activation. The findings would be of potential interest, although in its current form the claims may not be fully substantiated.

Major points.

The data presented do not seem to allow one to discriminate between the effect of KAP1 depletion on kinetics of elongation versus pause release. The data on the speed of elongation are weak. PRO-seq does not appear to show differences in elongation kinetics. For example, Fig 4A, apart from showing effects that are rather minor, more importantly do not seem to show changes in Pol II location, but appear more to show differences in Pol II amounts at the same locations. In Figure 4E, the very top panel does appear to show potential differences in kinetics, but this is the only visible evidence as far as I can see. Other genes, per replicate, could be shown in main or supplement figures alongside control expression-matched genes.

An argument is made that the immediate early genes induced in this serum starvation system are too short to measure elongation velocity as done in Hah et al., PMC3099127, and this may be the compounding factor. However, ROCC, the method that is used to estimate the elongation velocity here, and possibly the first method to do so on short genes, does not appear to be validated for this purpose. Visually from browser shots, there appears to be little if any differences in elongation, but rather, the defects are in pause release or in initiation.

ChIP-seq comparisons are not clearly described in terms of normalization. This is particularly important because normalization will change how the data look in terms of signal spillover to the gene body versus the overall increase of signal. Appropriate ChIP-seq spikein molecules would be different for each ChIP type and should be shown for each ChIP type (Pol II, HA, transcription factor, nucleosome, individually for each ChIP replicate. This is important for Fig 2 and Fig 5 interpretation. Fig 5 should have examples of individual loci or metaplots of affected and control groups as well rather than or in addition to box plots.

A point related to the above one is that, regardless of ChIP normalization, KAP1 ChIP-seq metaplots in Fig 2 and Sup Fig 2A should be extended outside of the genes to show non-specific background, a

nonchanging control gene example should be shown in addition to Sup 2B, and a metaplot for basal expression-matched and/or Pol II signal-matched control genes should be shown.

Is grouping genes by numerical magnitude of fold-change (2,4, 8, 16-fold) the best way to find measurable changes in Pol II and other factor dynamics? Would looking for genes with the highest absolute change be a better approach? For example, a gene with basal expression value of 1 and fold-activation 16 (+15 net expression change) would be picked here, but a gene with basal expression of 1000 and 1.5 fold change (+500 net expression change) would not be picked, even though the latter should show larger changes in factor occupancies.

Minor points.

Line 96. Estimate the % protein depletion at 8-hour datapoint in figure 1B.

Line 99-100. No difference in cell proliferation rates +/-KAP1 depletion might be connected to either incomplete protein depletion or to the way proliferation is measured. Can authors estimate the time course in terms of a number of cell divisions? This is more relevant if this is a slowly dividing cell line.

Line 99-100. More information may be needed on how cell confluency was measured, including possibly some images.

Lines 103-104. Could a browser shot be shown for an example gene for TT-seq tracks?

Lines 113-116. The argument for minimal changes in protein expression leading to choosing the IEGs appears convoluted. Perhaps this is just wording.

Sup figure 1D shows GO analysis of genes with FC>1.5. What about genes with 2,4,8,16 fold-change selected for further analyses? Are known IEGs being caught among these genes?

Lines 142-144. The effect of IEG depletion is modest.

Sup Fig S2A is redundant with main Fig 2A.

Sup Fig S2B tracks are not distinguishable from background unless extended outside in both directions. This comment is related to a major comment above.

Lines 177-178. Authors should show control genes that are expression- and/or Pol II occupancy-matched to IEGs in basal conditions. Same applies to lines 220-223 (Fig 4).

Figure 4D. Control basal expression-matched genes should be shown.

Figure 5. Example browser shots should be shown in main or supplementary figure.

Wording:

Line 121 starved from serum should be "...of serum"?

Line 123 “diminished the temporal expression” – the meaning of this wording is unclear.

Line 204 by increasing kinetics, did authors mean increasing elongation velocity or speed?

Lines 209-210 “actively synthesizing increased magnitudes of transcription”

Line 270 “increases Pol II elongation kinetics” – is velocity meant?

Reviewer #2 (Remarks to the Author):

In this study, Hyder et al. investigate effects of acute depletion of KAP1 (also known as TRIM28), a factor previously implicated in regulation of RNA polymerase II (Pol II) elongation whose precise function remains unclear. By introducing an inducible degron, or dTAG, in frame with KAP1, they achieve efficient degradation of KAP1 within ~8 hr of small-molecule addition in HCT116 colon cancer cells. The effects on global transcription, measured by transient transcriptome sequencing (TT-seq), are modest, but among genes that are upregulated is a subset of serum-inducible, immediate early genes (IEGs). The authors investigate the response to serum stimulation in the presence or absence of KAP1, and find that, while expression of IEGs was attenuated at both mRNA and protein levels by depletion of KAP1, Pol II elongation kinetics, measured by precision run-on transcription sequencing (PRO-seq), was accelerated; when KAP1 was degraded, Pol II transcribed farther into the bodies of IEGs by 5 and 10 min after serum addition. How this might lead to reduced steady-state IEG expression is suggested by the reduced occupancy of multiple components of the Pol II transcription machinery, including initiation and elongation factors and Pol II itself, at later time points (30 min). The authors posit a direct causal connection: faster elongation at early time points leading to increased accumulation or aberrant retention of elongation complexes in 3' gene regions, impairing Pol II recruitment to the promoter region for additional rounds of transcription. This is an attractive model, which might fit with previous studies indicating connections between termination and initiation, but the data presented do not rule out (and the authors do not discuss) alternative, more indirect mechanisms. How KAP1 is influencing Pol II elongation rate (negatively, according to the model) is addressed only speculatively in the Discussion. The core findings of the study are interesting and will be of considerable interest to the field, so I would recommend publication after the authors have tested at least one of the more likely mechanisms for effects on Pol II elongation speed or impaired initiation (see below) and toned down some of their stronger mechanistic conclusions.

My general concerns:

1. Throughout, the authors are a bit careless with terms that, depending on context, can mean differences in either time or space (e.g. “downstream,” “early/late”). Because both time (after serum stimulation) and distance (into gene bodies) are critical variables here, it is important to avoid such ambiguity. I have tried to flag instances where things get confusing in my specific comments below, but I

urge the authors to go over the manuscript carefully with an eye towards avoiding ambiguous or imprecise language.

2. As noted above, the authors reach for the simplest and most attractive mechanism to reconcile their “counterintuitive” results (increased elongation rate directly causing diminished initiation). An indirect effect, mediated by a signaling pathway that prematurely dampens IEG expression when the initial round of transcription is aberrant, say, by turning off a transcriptional activator, is also possible (and might also lead to reduced occupancy of Pol II machinery at the promoter). The authors should at least discuss such possibilities.

3. In the Discussion, the consideration of mechanisms by which KAP1 might influence elongation rate curiously omits the one known (and potentially testable) mechanism for regulating Pol II elongation speed: phosphorylation of the SPT5 subunit of DSIF by CDK9 (ref. 26, as well as doi: 10.1038/s41467-018-03006-4 and doi: 10.1038/s41586-018-0214-z). Also, although probably beyond the scope of this study, one way to test the favored model for how KAP1 depletion is negatively affecting IEG expression would be to ask if this phenotype can be rescued by expressing a “slow” mutant Pol II (see ref. 68).

Specific concerns and comments:

4. Lines 17-8: As an example of point 1 above, to many in the transcription field, an “early elongation defect” means a defect in elongation that occurs soon after release from the promoter-proximal pause, but here it refers to a defect in elongation on IEGs that was detected soon after cells were stimulated with serum.

5. Lines 17-20: More substantively, this passage infers causation where there is only evidence for correlation between genes where elongation kinetics are perturbed at early time points but expression is dampened at later times.

6. Line 28: “Briefly” should probably be changed to “Shortly” or “Soon.”

7. Line 40: SPT5 is a subunit of DSIF; referring to it as “SPT5/DSIF” is confusing (the same nomenclature is used earlier in the paragraph to describe a complex between two different proteins, CDK9/CycT1).

8. Lines 99-101: A minor point, but to conclude that the lack of effect of dTAG treatment on cell proliferation over 6 days is meaningful, the authors should provide immunoblot data to ensure that KAP1 proteins levels remain low over the extended time course.

9. Line 128: Also minor, but “we treated cells +/- 30 min serum +/- dTAG” is lab notebook shorthand and should be expanded and better explained.

10. Lines 151-6: The metagene profile of KAP1 ChIP-seq signals in Fig. 2a is indeed reminiscent of Pol II and other factors thought to “travel” with the elongation complex, but it would help to show a direct comparison/superimposition (for example, with the profile in Fig. 3b), perhaps in a supplementary figure.

11. Lines 185-90: Another minor point, but in describing the data in Fig. 3, comparing the Pol II occupancy data from “4-fold” and “16-fold” IEGs, the authors confusingly state the values as a range, e.g. “7-14% for 4-fold and 16-fold, respectively” and so forth. It would be better to say “7% and 14%.”

12. Lines 190-1: This may be more of an aesthetic judgment, but the genome browser tracks in Supp. Fig. 3e make a stronger visual case for altered elongation kinetics than do the metagene or box plots, and should perhaps be moved to the main figure.
13. Lines 224-6: In this sentence, describing data from 5- and 10-min time points, “late” is used to refer to a positional effect (“in the GB”) and, in the next breath, “early” is used to refer to a 15-min data point.
14. Lines 241-52: This passage, describing first the limitations of PRO-seq data obtained during IEG induction for calculating elongation rate and then a computational strategy for extracting an “elongation rate proxy” from the data, could be revised to increase clarity. First, the gene-length requirement (2 kb per minute of treatment) is not explained (I presume it refers to the estimated elongation rate of Pol II *in vivo*). Second, what is the relationship between the calculated “elongation rate proxy” and the actual elongation rate? Can the authors extrapolate an approximate value for the increase in elongation rate when KAP1 is degraded? Is this effect actually larger on “16-fold” IEGs than on “4-fold” IEGs, as Fig. 4d seems to suggest?
15. Fig. 5c-f and Supplementary Fig. 5c-e: Could the authors provide more detail on how the ChIP-seq data were normalized? Specifically in the case of pS5 and pS2 of the Pol II CTD, were these signals normalized to total Pol II (which is generally more useful than “raw” signals)?
16. Lines 266-8: This sentence doesn’t make sense as written, first mentioning “reduced occupancy” and then “stronger increases.” Do the authors mean “decreases?”
17. Lines 270-2: This concluding sentence of the results section needs to be toned down, in my opinion, e.g. by qualifying “dampens” with “might” and “explaining” with “possibly.”
18. Lines 282-5: The authors invoke delayed termination or release of Pol II from 3’-ends of IEGs as a possible explanation for a “reinitiation” defect. Have they analyzed their PRO-seq or Pol II ChIP-seq data for evidence of transcriptional read-through or increased Pol II occupancy beyond the normal termination zone, which would support such a mechanism?
19. Lines 308-9: The authors should say explicitly how their “phenotypes differ from these previous observations.” Without that context, I can’t assess the argument that follows.
20. Lines 310-5: Failure to resolve R-loops is invoked as a possible explanation for the KAP1-deficient phenotype of increased Pol II occupancy at IEG 3’ ends. Have the authors tried to ChIP for RNA:DNA hybrids?
21. Line 363: Another, potentially confusing use of “early Pol II elongation kinetics” could be fixed by deleting “early” and inserting “during prior rounds of transcription” after “kinetics.”
22. Lines 369-70: It is not clear which “other factors” this sentence refers to.
23. Lines 393-4: Again, I would avoid using “early transcriptional checkpoint” (especially when followed by “downstream consequences”)—terminology that has previously (and more commonly) been applied to checkpoint-like controls that are imposed within individual transcription cycles (see ref. 19 and the aforementioned doi: 10.1038/s41467-018-03006-4 for examples).

Point-by-point responses to the reviewers' critiques

Reviewer 1

The manuscript by Hyder et al describes KAP1/TRIM28 as a regulator of elongation that acts selectively on induced transcription. By comparing transcription induced by serum re-induction after starvation in normal HCT116 cells and cells with degran-depleted KAP1, authors propose that KAP1 deficiency leads to redistribution of Pol II that is apparently consistent with altered elongation kinetics. KAP1 is necessary for full activation and is distributed to gene bodies with activation. The findings would be of potential interest, although in its current form the claims may not be fully substantiated.

We thank the reviewer for suggesting that the findings in this study are of potential interest. Below we address all the reviewers' points to provide additional evidence strengthening the proposed model, more precisely discuss alternatives, and improve the presentation and clarity of the story. Collectively, we hope the reviewer agrees that we have addressed all the points below and provide compelling evidence supporting the model that KAP1 negatively regulates Pol II elongation kinetics to facilitate signal-induced transcription.

Major points

Point 1. The data presented do not seem to allow one to discriminate between the effect of KAP1 depletion on kinetics of elongation versus pause release. The data on the speed of elongation are weak. PRO-seq does not appear to show differences in elongation kinetics. For example, Fig 4A, apart from showing effects that are rather minor, more importantly do not seem to show changes in Pol II location but appear more to show differences in Pol II amounts at the same locations. In Fig 4E, the very top panel does appear to show potential differences in kinetics, but this is the only visible evidence as far as I can see. Other genes, per replicate, could be shown in main or supplement figures alongside control expression-matched genes.

We thank the reviewer for raising these important points, which have guided us to improve our story with new experiments, data analysis, and clarifications throughout the manuscript. Below we address each subpoint one-by-one.

1. Minor nature of the transcriptional phenotype upon acute KAP1 depletion.

We completely agree with the reviewer that the effects observed upon acute KAP1 depletion are “minor”, but we are “directly” examining the contribution of a single factor to transcriptional outputs, which allows us to reveal the underlying molecular mechanism without any confounding indirect effects. We have now made a few additions to the revised manuscript to reflect this minor phenotype (**pgs. 4-5, lines 134-139** to describe changes in RT-qPCR data and **pg. 7, lines 254-260** to describe changes in Pol II ChIP-Seq data). Additionally, our PRO-Seq data in **Fig. 4** reveals an almost 2-fold median increase in nascent transcription at the 5 min serum time point upon acute KAP1 depletion when compared to DMSO control (see metagene in **Fig. 4A** and quantitation of nascent transcription in **Fig. 4C**). Modest changes in gene expression alterations can lead to important functional outcomes, and at least in this case, the gene expression changes conferred by acute KAP1 loss do confer changes in protein expression (**Fig. 1H**) and may help explain diseased states accrued upon chronic KAP1 loss. Notably, the modest magnitude of the transcriptional phenotype upon acute KAP1 depletion is in line to the recently observed transcriptional phenotypes upon acute depletion of other transcriptional regulators, such as members of the PAF1 complex (PMID: 34146481) and NELF (PMID: 32155413). Collectively, acute factor depletion reveals nuances in biology that can often be overrepresented by chronic knockout approaches, and that minor transcriptional phenotypes can also be biologically meaningful.

2. PRO-seq does not appear to show differences in elongation kinetics, and changes in Pol II location vs differences in Pol II amounts at the same locations.

We agree with the reviewer that indeed there are changes to Pol II amounts between dTAG-treated and DMSO-treated conditions upon serum stimulation at the same locations. This is shown throughout the manuscript, but is exemplified in the metagene profiles for Pol II ChIP-Seq data in **Fig. 3A-B**, Pol II ChIP-Seq browser tracks in **Fig. 3F-G**, as well as PRO-Seq data in metagene profiles in **Fig. 4A-B** and browser tracks in **Fig. 4D and Supplementary Fig. 4D-E**. Collectively, these data show that at early serum stimulation time points (15 min for Pol II ChIP-Seq, and 5 and 10 min for PRO-Seq), we observe increased read density in the gene bodies and 3' ends of IEGs in the dTAG-treated condition when compared to DMSO vehicle control.

In addition to increases in Pol II amounts at the same location, the data show modest but clear changes of Pol II occupancy extending towards the 3' end of IEGs (depending on the time point tested). In particular, the 5 min PRO-Seq data shows that acute KAP1 depletion caused Pol II to move further into the gene bodies of IEGs at the earliest time point of serum induction (5 min, Fig. 4A). By the late time point of serum stimulation (10 min), we observed an increased density of Pol II in the 3' ends of IEGs (10 min, **Fig. 4B**). Given that inducible transcription allows the precise tracking of Pol II as it transcribes the template DNA, this data implies that acute KAP1 depletion causes Pol II to reach the end of IEGs at an earlier time point than when KAP1 is expressed.

In the classic form of elongation rate metrics, prior groups have measured elongation rate changes by tracking the “leading edge” of Pol II after treating with CDK9 inhibitors to arrest Pol II in the promoter-proximal region (and then releasing the drug) (PMID: 19820712 and PMID: 30503775) or after signal induction (PMID: 23523369). In either case, how far Pol II moves into the gene body (the leading edge) prior to reaching the end of the gene is then divided by time of release of CDK9 inhibition or signal activation to calculate an elongation rate (kb traveled/min). Based on this rationale that a Pol II molecule that travels further after signal induction has a faster rate, we noticed that our 5 min serum stimulation PRO-Seq data had a similar profile. As explained in the revised manuscript, because IEGs in the serum response network are overall short (average length ~17-kb), monitoring Pol II release after CDK9 inhibition or measuring the “leading edge” are difficult as Pol II reaches the 3' ends of some genes in ~5 min, and of most genes in ~10 min (making kinetic calculations nearly impossible). Our new ROCC approach (described in detail to address Reviewer 1 Point 2 below) mitigates these issues and has allowed us to pinpoint differences in elongation kinetics. Because of the gene length limitation, ROCC does not calculate Pol II elongation rate directly (distance traveled/time), but provides an accurate estimate, termed here “Proxy Rate”. The details of the method are described in **pgs. 9-11, lines 348-420**. Despite the acknowledged limitations of the approach (also described in **pgs. 9-11, lines 348-420**), ROCC allows us to make otherwise impossible quantitation's of Pol II elongation kinetics that are consistent with other experiments presented throughout the manuscript. Together, our new data and data analysis provide a strong support to the elongation kinetics claim and the ROCC method will be a valuable tool for the field.

3. Discriminate between the effect of acute KAP1 depletion on kinetics of elongation vs pause release.

The data presented in **Fig. 4** indicate that Pol II traveled further into the gene bodies of IEGs at the early serum stimulation time points when KAP1 is acutely depleted, without major changes to nascent transcription at promoter-proximal regions. Together these two observations support the proposed model that KAP1 negatively regulates elongation kinetics. However, these results do not entirely exclude changes in pause release as, or as part of, the mechanism for augmented nascent transcription during early serum stimulation. Pause release occurs when stalled Pol II transitions from the promoter-proximal paused state into the gene body for active transcription, which is primarily driven by the CDK9 kinase phosphorylating the Pol II CTD as well as the SPT5 subunit of DSIF (PMID: 32859893 and PMID: 12052871). To address whether increased pause

release (rather than or in addition to increased elongation kinetics) can explain the observed transcriptional phenotypes, we now provide additional experimentation of pause release markers as well as refined analysis on the PRO-Seq data.

One mechanism whereby KAP1 could negatively regulate pause release would be to inhibit CDK9 recruitment to the promoters of IEGs. If this model was correct, acute KAP1 depletion would lead to increased CDK9 occupancy at the early serum stimulation time point (5 min). However, profiling of CDK9 at this time point revealed that CDK9 occupancy does not increase upon acute KAP1 depletion, showing that KAP1's primary role may not be to inhibit CDK9 occupancy to block pause release (**Supplementary Fig. 6A**).

Another possible mechanism would be to regulate CDK9 activity, rather than recruitment. Measuring the phosphorylation status of SPT5 (pSPT5) is an indirect, accepted practice in the field that informs on the level of kinase activity. While many SPT5 residues are substrates of CDK9, phosphorylation of the SPT5 C-terminal region 1 (CTR1) domain is characterized as a marker of pause release, including site specific phosphorylation at residue Thr806 (labeled as pSPT5) (PMID: 32859893, PMID: 16427012, and PMID: 10757782). To assess this further, we monitored total and pSPT5 by ChIP-Seq. While ChIP-Seq of total SPT5 and pSPT5 revealed that occupancy of these factors expectedly increased upon 5 min of serum treatment in DMSO vehicle control, acute KAP1 depletion led to no major increases in occupancy, suggesting that KAP1's primary role may not be to decrease CDK9 activity to block pause release to restrict early transcription (**Supplementary Fig. 6A-C**). These data are overall consistent with the fact that regulation of Pol II elongation kinetics (**Fig. 4, Supplementary Fig. 4-5**) is the primary mechanism for increases in transcription upon acute KAP1 depletion observed at the early serum stimulation time points.

Another supporting evidence is to instead examine Pol II itself to ascertain pause release, which is classically defined by Pol II molecules traveling from the promoter-proximal site to the gene body. If KAP1 primarily regulated pause release to induce transcription, it would be expected that promoter-proximal Pol II levels in the early serum stimulation time point (5 min) would potentially decrease with a concomitant increase in gene body signal. Although Pol II signal in the gene body regions of IEGs increased (~2-fold) in the 16-Fold IEG cluster at the early serum stimulation time point (5 min) upon acute KAP1 depletion (**Supplementary Fig. 4B**), consistent with increased transcription, metagene analysis showed no major Pol II decreases in the promoter-proximal region (**Fig. 4A**). Additionally, browser tracks of three representative IEGs (*FOSB* shown in **Fig. 4D**, *FOS* shown in **Supplementary Fig. 4D**, and *NR4A1* shown in **Supplementary Fig. 4E**), as well as PRO-Seq quantitations in promoter-proximal and gene body regions (**Supplementary Fig. 4A, B**) underscore these observations that the strongest differences in signal observed upon acute KAP1 depletion are increases in signal in gene body regions with minimal changes to promoter-proximal regions. Overall, these data, along with monitoring the occupancy of pause release markers, support that pause release does not appear to be the major cause for the increases in nascent transcription during early serum stimulation upon KAP1 acute depletion. Despite the many regulatory steps in the transcriptional cycle and their interrelationships, multiple evidence in our manuscript, carefully explained in this rebuttal letter, indicate that elongation, rather than pause release, is at least a major driver of the defined transcriptional phenotypes.

4. Fig. 4E the very top panel does appear to show potential differences in kinetics, but this is the only visible evidence as far as I can see. Other genes, per replicate, could be shown in main or supplement figures alongside control expression-matched genes.

We thank the reviewer for acknowledging that at least one example of data in the parental manuscript did potentially show differences in elongation kinetics. To further bolster the evidence provided, we now have placed browser tracks of many genes in the main manuscript, including

three representative IEGs and one control gene (**Fig. 4E** and **Supplementary Fig. 4D-F**). Below we also include two representative IEGs (*FOSB* and *NR4A1*) replicate-wise as the reviewer requested. We hope these additional browser tracks in the story and data presented below will compel the reviewer to agree with the original statement that the original browser track presentation does visually show differences in transcription elongation.

Point 2. An argument is made that the immediate early genes induced in this serum starvation system are too short to measure elongation velocity as done in Hah et al., PMC3099127, and this may be the compounding factor. However, ROCC, the method that is used to estimate the elongation velocity here, and possibly the first method to do so on short genes, does not appear to be validated for this purpose. Visually from browser shots, there appears to be little if any differences in elongation, but rather, the defects are in pause release or in initiation.

We thank the reviewer for this comment, which is fully addressed below.

1. Benchmark validation of the ROCC method as a proxy of elongation rate.

The reviewer was entirely correct that the approach for measuring ROCC had not been benchmark validated to serve as a proxy for elongation rate calculations. We have now included critical data validating the approach as a proxy for elongation rate in **Supplementary Fig. 5A** with discussion in the manuscript (**pgs. 9-10, lines 348-369**). We also now include an entirely dedicated section to this new approach on **pgs. 9-11, lines 327-420**.

Given the gene-length constraints of the short genes in the serum response network (counter to the longer estrogen-inducible genes defined in Hah et al., PMC3099127), we devised a method called the Rate of Change in Coverage (ROCC), which relies on nascent RNA transcription signal intensity rather than nascent RNA transcription wavefront. The ROCC is estimated at each bin (50-bp) by examining the coverage at that position at each time point and regressing the coverage on time directly. The high-resolution estimates (one per each bin in the gene) are then combined to form a gene-wide estimate, which is not an exact calculation of elongation rate, but serves as an estimate of elongation rates (referred to as Proxy Rate calculation).

To provide benchmark validation that the ROCC method is consistent with elongation rates previously estimated using the leading edge, which calls the nascent RNA wavefront using a Hidden Markov Model, a published nascent transcription dataset for estrogen-inducible genes was examined to estimate elongation rates according to the leading edge procedure (PMID: 23523369) and the newly devised ROCC method. Comparison of the two datasets showed that there is a statistically significant correlation between the average ROCCs and the original rates that were computed according to the leading edge method (**Supplementary Fig. 5A**). While the two methods are indeed correlated, it is noteworthy that the leading edge and ROCC approaches examine different characteristics of the same dataset. The leading edge calls the most likely position of the front of the transcribing wave, while ROCC utilizes nascent transcription information to estimate a rate across the time course for each binned position. Thus, we emphasize that ROCC is a 'Proxy Rate' that infers the elongation rate, allowing us to bypass the gene-length constraints of serum-inducible IEGs. See figures (**Fig. 4E** and **Supplementary Fig. 5B-D**) and text discussion (**pgs. 9-11, lines 327-420**) for a detailed account of how we utilized the ROCC and Proxy Rate analysis to determine how acute KAP1 depletion perturbs elongation kinetics at the earliest time points of serum induction.

2. Differences in elongation (as claimed by the authors) vs pause release or initiation (as claimed by the reviewer).

We actually do not know what visualization and/or data interpretation the reviewer did to conclude that pause release or initiation are driving the transcriptional phenotypes upon acute KAP1 depletion. The primary PRO-Seq data and its analysis (metagenes in **Fig. 4A-B**, quantitation's in **Fig. 4C**, genome browser tracks in **Fig. 4D** and **Supplementary Fig. 4D-E**, and Proxy Rate analysis in **Fig. 4E** and **Supplementary Fig. 5B-D**) clearly reveal that nascent transcription increases upon acute KAP1 depletion in the early serum stimulation time point (5 min). These data does not support the model that KAP1 primarily role is to control pause release and/or initiation.

Pause release: Why the data does not support the reviewer's proposed model of pause release is thoroughly described with new additional experimentation and data analysis in response to **Point 1** above.

Initiation: Serum stimulation induces multiple rounds of transcription by collectively promoting Pol II promoter recruitment (initiation), pause release, elongation, termination and subsequent rounds of re-initiation. In our datasets, initiation at the early serum stimulation time points expectedly occurs at IEGs, as Pol II promoter-proximal levels in the Pol II ChIP-Seq (**Fig. 3A**) and PRO-Seq (**Fig. 4A, B**) data increase (~1.5–3-fold) above the serum-starved condition (in DMSO vehicle control). However, there is no apparent increase in signal in either dataset in the dTAG-treated over the DMSO-treated cells, clearly ruling out an initiation effect. At an aggregate level, most genes do not display increased Pol II levels upon dTAG treatment in their promoter-proximal regions (**Fig. 3A** and **Fig. 4A, B**), highlighting that KAP1 does not directly regulate Pol II recruitment at IEGs. The increases in gene body signal upon acute KAP1 depletion are much stronger than the increases in promoter-proximal signal, in line with an elongation mechanism, not initiation, being the primary driver of increased transcription upon acute KAP1 depletion during early serum stimulation time points.

To provide further evidence that initiation does not increase at the early serum stimulation time points upon acute KAP1 depletion, the occupancy of MED1, a subunit of the Mediator complex that participates in Pol II recruitment, was now measured at 5 min of serum treatment in dTAG- and DMSO-treated cells. Contrary to the reviewer's proposed initiation model, acute KAP1 depletion does not increase MED1 recruitment to IEG promoter-proximal regions, again ruling out the reviewer's proposed initiation model. This new data is in line with no major increases in promoter-proximal Pol II levels by ChIP-Seq and PRO-Seq, and thus strongly support that transcription initiation is not the driver of early phenotypes observed upon KAP1 loss.

Finally, we would like to highlight that, rather than initiation, transcription re-initiation is decreased upon acute KAP1 depletion in the late serum stimulation time points (30 min), as evidenced by a drop in Pol II recruitment upon acute KAP1 depletion (**Fig. 3B**). However, this phenotype is observed late in the process of signal-induced transcription and is thus a secondary phenotype as a consequence of the increased Pol II elongation defects that occur early in the signal-induced transcription timescale (before 15 min of induction). Since the secondary phenotype is stronger, the reviewer may have been confused, as it can easily capture the reader's attention but there is no indication that KAP1 is involved in this process and is simply a long-term, potentially indirect, transcriptional consequence. We made sure that this is clearly elaborated throughout the revised manuscript to improve readability.

Point 3. ChIP-seq comparisons are not clearly described in terms of normalization. This is particularly important because normalization will change how the data look in terms of signal spillover to the gene body versus the overall increase of signal. Appropriate ChIP-seq spike in molecules would be different for each ChIP type and should be shown for each ChIP type (Pol II, HA, transcription factor, nucleosome, individually for each ChIP replicate. This is important for Fig 2 and Fig 5 interpretation. Fig 5 should have examples of individual loci or metaplots of affected and control groups as well rather than or in addition to box plots.

We thank the reviewer for this comment regarding data normalization.

1. ChIP-Seq comparisons and normalization.

We thank the reviewer for these comments regarding normalization, and we completely agree that differences in normalization will drastically change how the data looks. In the parental manuscript, we wrote the following in the methods section: “*Duplicates were marked and removed using picard/2.10.3 and then files were sorted and indexed using samtools/1.6 in preparation for Bigwig generation. Normalized BigWigs were made from sorted bam files using the bamCoverage command using deeptools/2.3.5 command bamCoverage. HA ChIP-Seqs (Fig. 2) were normalized using spike-in Drosophila reads while the remainder of the ChIPs (Fig. 3 and Fig. 5) were normalized to read depth (counts per millions or CPM using Deeptools). For spike-in normalization, scaling factors were defined using Deeptools command multiBamSummary and then inputted into bamCoverage for Bigwig generation for normalization. The Integrated Genome Viewer (IGV) was used for visualization.*”

We apologize if there was any potential lack of clarity and detailed description of ChIP-Seq normalization. Below, we describe each ChIP-Seq data (**Fig. 2, 3, and 5**), how it was analyzed in the parental manuscript, and describe new data analysis utilizing both read depth and spike-in normalization, as well as new experimentation for **Fig. 5**, which were lacking spike-ins included in the initial experiment. We have also written a thorough explanation of the methods in the Supplementary Information section, and updated our GitLab page documentation, which includes details on how pre-processing and normalization is conducted (see data availability) to increase rigor and reproducibility. There are three figures with ChIP-Seq in the paper: **Fig. 2** (HA ChIP-Seqs to monitor KAP1 occupancy +/- serum treatment), **Fig. 3** (Pol II ChIP-Seqs to monitor Pol II dynamics +/- serum +/- dTAG treatment), and **Fig. 5** (ChIP-Seqs of factors to monitor dynamics +/- serum +/- dTAG treatment). Below we detail each figure and normalization strategies used.

Fig. 2: For **Fig. 2**, the data was originally normalized to Drosophila spike-ins, and thus we think the reviewer would agree that this is an appropriate method of normalization. To further address the reviewer’s comment, we have listed out the total number of aligned and deduplicated human and Drosophila reads in **Supplementary Table 5**, which includes scaling factors generated from Drosophila reads that were used to generate normalized bigWig files used in all the analysis. We hope the reviewer will accept that spike-ins provide a robust approach to normalize ChIP-Seq data, especially when trying to increase rigor by performing a factor’s ChIP in a dTAG context.

Fig. 3: For **Fig. 3**, the data was originally read-depth normalized (as stated in the methods section), however we did add spike-ins during the ChIP experiment. In the past our lab has had issues with spike-in scaling factors causing abnormal genome-wide changes to Pol II levels that could be attributed to a normalization artifact, so we initially computed the analysis in the original submission using read depth normalization. Thus, to address this reviewer’s point, we have re-performed the analysis using spike-in alignment and scale factor generation prior to bigWig normalization using those scale factors. We hope the reviewer will appreciate the data below (metagene profiles of 16-Fold IEGs and non-DE genes at serum 0 min, 15 min, and 30 min +/- dTAG treatment) and per-replicate browser track data highlighting that spike-in normalization does not lead to any major changes in conclusions generated from the data as metagene profiles

and browser tracks show that spike-in normalization does not affect Pol II phenotypes identified from the read depth normalized data.

Serum 15 min: dTAG treatment at this time point shows the same increases in Pol II density in the gene body and 3' ends of 16-Fold IEGs when compared to DMSO vehicle control.

Serum 30 min: dTAG treatment at this time point shows the same Pol II density decreases at 16-Fold IEGs throughout the gene (promoter and gene body).

For both the serum 15 min and 30 min time points, we observed no major changes at the non-inducible cluster of genes (non-DE genes, n=10,361), highlighting that spike-in normalization does not cause any major genome-wide alterations to Pol II signal. The reviewer can compare Pol II signal densities of the two analyses (spike-in and read depth-normalized) below.

Serum 0 min: The only major qualitative difference we noticed between read depth and spike-in normalization methods was in the Serum 0 time point where we observed a modest increase in Pol II signal in the promoter-proximal regions of IEGs in dTAG condition compared to control DMSO when using spike-in normalization, while read depth normalization saw very minor signal changes. To determine if this was specific to IEGs, we looked at non-DE genes for all conditions and observed a similar trend with increased Pol II density at the promoters of non-DE genes in the serum 0 time point, potentially suggesting that this may be a normalization artifact rather than a biological result (Data in figure below).

To assess whether this is a true global increase in promoter-proximal Pol II levels or a potential normalization artifact, we used a quantitative orthogonal approach (ChIP-qPCR) to monitor Pol II occupancy at the promoters of three representative IEGs (*ATF3*, *FOSB*, and *FOS*) and a negative control gene (*GAPDH*) +/- dTAG treatment, and found no significant changes, strongly confirming that no Pol II occupancy changes exist at promoter-proximal regions between dTAG and DMSO treated cells (Data in figure below).

Given the fact that this data perfectly agree with read depth analysis metrics, we hope that the reviewer will appreciate that spike-in addition did not drastically change the core findings that acute KAP1 depletion alters Pol II density upon serum stimulation at both time points tested (15 and 30 min). Because the spike-in did introduce potential minor normalization issues with the serum 0 min time point, we decided to keep the read depth normalized data in the main manuscript.

If the reviewer would like to visualize the data, ChIP-Seq tracks of Pol II data, per replicate, are shown below for one representative IEG (*FOSB*) and one control gene (*SFPQ*).

Fig. 5: For **Fig. 5**, the data was originally read-depth normalized (as stated in the Methods section), however we did not add spike-ins during the ChIP experiment. To ensure that we properly address the reviewer's comment, we repeated the ChIP-Seq with spike-ins, and included the spike-in normalized data in the manuscript. Additionally, any new experiments to address this reviewer's comments (such as **Supplementary Fig. 6**) were also performed with spike-ins. Data analysis on these new ChIP-Seq assays found that spike-ins do not drastically change the results, and thus we have plotted the repeated experiments using spike-in normalization in **Fig. 5** and **Supplementary Figs. 6** and **7**. We also included browser tracks of the following IEGs: *FOS* (**Fig. 5G**, **Supplementary Fig. 6D**), *ATF3* (**Fig. 5H**) and the control gene *FUS* (**Supplementary Fig. 6E**, **Supplementary Fig. 7E**). In case the reviewer would like to see another control gene example, an additional control gene browser track (*NXF1*) for all eight ChIP-Seqs is plotted below.

Finally, we list out the alignment statistics and total number of spike-in molecules (reads) per ChIP-Seq experiment as requested in **Supplementary Table 5**. While all details are in the ChIP-Seq analysis section of the methods as well as in the D'Orso lab GitLab page, few brief details about how we arrived at those alignment statistics are here: *The alignment and deduplication statistics are extracted from Picard. The scaling factors are generated by the multiBamSummary program from deepTools, which generates scale factors from deduplicated and processed spike-in aligned bam files that are then used to scale and normalize the human-aligned bam files to generate spike-in normalized bigWigs.*

Taken together, we hope the reviewer overall agrees that we have provided extensive clarification on how ChIP-Seq data was normalized throughout the manuscript, and that normalization using multiple methods across varying experiments (read-depth normalization vs spike-in normalization) did not lead to any changes in the final conclusions of this study.

Point 4. A point related to the above one is that, regardless of ChIP normalization, KAP1 ChIP-seq metaplots in Fig 2 and Sup Fig 2A should be extended outside of the genes to show non-specific background, a nonchanging control gene example should be shown in addition to Sup 2B, and a metaplot for basal expression-matched and/or Pol II signal-matched control genes should be shown.

We thank the reviewer for raising this important point and for highlighting an oversight on our part in presenting and utilizing our HA ChIP-Seq data. Addressing this point has revealed new insights and has allowed us to clarify KAP1 density at both IEGs and control genes. We have now performed the following analysis included in the revised manuscript:

1. Revised extended metaplots.

A KAP1 (HA) ChIP-Seq metaplot that is extended outside of the genes to show non-specific background (50-kb prior to the TSS and 50-kb after the pA site) is now plotted in **Supplementary Fig. 2A**. This metaplot average data highlights that KAP1 density observed at the promoter and gene bodies of IEGs is indeed above background signal. The reviewer (in a minor comment below) mentions that the original **Supplementary Fig. 2A** was redundant, which showed the additional (180 min) serum stimulation time point. We agree with this reviewer, and due to the fact that addition of a fourth line decreases visibility of the figure, we have taken this plot out of the manuscript. However, to properly address this reviewer's point, we created an extended metaplot of the former **Supplementary Fig. 2A**, which can be seen below alongside the original plot highlighting KAP1 occupancy at the 16-fold IEGs cluster in untreated (serum 0), treated (serum 30 and 180 min) and in a dTAG condition for specificity (serum 30 min). Overall, this extended metaplot analysis, along with the new **Supplementary Fig. 2A**, clearly shows signal above background in the HA ChIP-Seq experiment and has helped improve the revised manuscript.

2. Nonchanging control gene.

Per the reviewers request, a non-changing control gene browser track is now shown below:

3. Metaplot for basal expression-matched genes.

The reviewer's point prompted us to look at KAP1 occupancy deeper at non-inducible genes. We wanted to only assess KAP1 signal at genes that contained KAP1 occupancy to examine occupancy genome-wide, as well as at IEGs (as we did in the parental manuscript). To address this carefully, we used MACS2 to call peaks of our KAP1 ChIP-Seq data in the serum 0 min time point condition, in which $n=12,665$ genes were identified to have a KAP1 peak at their promoters. We have plotted metaprofiles of KAP1 occupancy in all four conditions (DMSO-Serum 0 min, DMSO-Serum 30 min, DMSO-Serum 180 min, and dTAG-Serum 30 min) at all KAP1 peaks. Metagene profiles for KAP1 occupancy for the three conditions (to increase visibility) are included in **Fig. 2E**. To increase rigor, (1) we also extended the signal of the plots ± 50 -kb (All KAP1 peaks) with and without the 180 min serum stimulation time point, and (2) plotted the KAP1 metagene profiles for the top 200 KAP1 peaks (identified from MACS2 signal) below:

Overall, these data display that KAP1 promoter-proximal occupancy decreases at non-inducible genes upon serum treatment, suggesting that KAP1 may undergo a redistribution from the promoters of non-inducible genes to IEGs upon serum treatment. While we are unsure about the mechanism for this redistribution, it highlights the specificity that KAP1 occupies IEG gene bodies in a signal-dependent manner, as we have shown throughout the rest of the manuscript, and is in line with KAP1 specifically facilitating IEGs transcription elongation upon serum stimulation. While on average the KAP1 signal density decreases at non-IEG promoters, some genes show no major changes (*ZMYM6*), suggesting a heterogenous redistribution between

KAP1 molecules at IEGs and non-IEGs. A gene browser track of a representative IEG (*FOS*) in all four treatment conditions is shown in **Fig. 2B**; a gene browser track of a non-IEG (*SFPQ*) is shown in **Fig. 2D**; and throughout the manuscript we show browser tracks of additional IEGs (e.g., *FOSB*, *ATF3*, *NR4A1*) placed under Pol II profiling experiments (ChIP-Seq or PRO-Seq) that highlight the occupancy of KAP1 upon serum treatment and provide further potential evidence of KAP1 distribution to gene bodies as Pol II is transcribing. Additionally, we quantified the signal at the top 200 KAP1 peaks, which are plotted in **Supplementary Fig. 2C**. Replicate-wise browser tracks are also included for the reviewer below:

To increase rigor in our ChIP-Seq data, and to further convince the reviewer of the specificity of KAP1 binding at IEGs, we include below HA (KAP1) ChIP-qPCR data under the same four treatment conditions as the ChIP-Seq using primers that anneal to the *FOS* 3' end, validating at least the increase in signal at the 30 min time point at this one genomic region, concomitant with decrease signal at the 180 min serum time point (showing reversal of KAP1 binding) and when dTAG is added to the cells (revealing signal specificity).

Point 5. Is grouping genes by numerical magnitude of fold-change (2, 4, 8, 16-fold) the best way to find measurable changes in Pol II and other factor dynamics? Would looking for genes with the highest absolute change be a better approach? For example, a gene with basal expression value of 1 and fold-activation 16 (+15 net expression change) would be picked here, but a gene with basal expression of 1000 and 1.5 fold change (+500 net expression change) would not be picked, even though the latter should show larger changes in factor occupancies.

We thank the reviewer for this comment, which is fully addressed below.

1. Clustering methods: We agree that different clustering methods can generate differences in measurable gene expression changes. Thus, to address this comment, we turned to our RNA-Seq data (feature counts) and determined the average expression change between the Serum 0 min and Serum 30 min time points across the three RNA-Seq replicates of All IEGs induced >4-Fold in our dataset. Below is a histogram of All IEGs (n=69) for this figure plotted based on their net expression change.

The net expression change range of the 69 genes was 108 counts to 44,032 counts and we clustered them into 3 clusters (>8000 (n=12), >4500 (n=19), and >2000 (n=33)) based on this data. We also classified All IEGs (n=69) as an additional cluster for our analysis and compared this data to the previously defined 16-Fold IEGs used in the parental manuscript based on magnitude of fold-change. We then plotted meta-profiles of Pol II ChIP-Seq data (serum 0 min, 15 min, and 30 min time points with and without dTAG treatment). The profiles are included in the next page for the reviewer to assess.

Overall, we do not see major changes between the designation based on fold-change vs overall expression change. Additionally, as the reviewer implied, the larger the expression change, the larger the difference in Pol II occupancy intensities, similar to the data that was observed when comparing 16-Fold (n=16) and All IEGs (n=69) differences. Given that the difference between clustering conditions was minimal, we decided to leave this data in the reviewer responses, which will also be public if this manuscript is published.

Pol II ChIP-Seq (Serum 15 min)

Pol II ChIP-Seq (Serum 30 min)

Pol II ChIP-Seq (Serum 0 min)

Minor points.

Line 96. Estimate the % protein depletion at 8-hour datapoint in figure 1B.

We have quantified the 8 hr dTAG time point which shows an 88% decrease in protein expression. This is now added to the text (pg. 4, lines 98-101). For completion, the other relevant time points and their degradation are detailed in the Table below:

dTAG time (hr)	Percent acute KAP1 depletion
4	60%
6	76%
8	88%
12	90%
24	91%

Line 99-100. No difference in cell proliferation rates +/-KAP1 depletion might be connected to either incomplete protein depletion or to the way proliferation is measured. Can authors estimate the time course in terms of a number of cell divisions? This is more relevant if this is a slowly dividing cell line.

We agree with the reviewer's comment that the lack of difference in cell proliferation rates +/- acute KAP1 depletion might be connected to: (1) either incomplete protein depletion or (2) to the way cell proliferation was measured. Due to the kinetics of the dTAG system never achieving complete degradation (as seen in the quantitation in the last reviewer comment as well in Fig. 1B), this is a minor limitation of the system. Importantly, the lack of complete degradation did not affect our ability to observe modest gene expression changes.

To address the second point about the way cell proliferation is assessed, we provide a detailed account of the assay in the Supplementary methods section labeled as 'Cell confluence assay' (pg. 14 of Supplementary Information).

To address the comment that cell growth could be impacted by the measurement chosen, we repeated the cell growth assay below by seeding cells in 6-well plates and then counting cells for 3-days post-dTAG treatment. The results agree with the original IncuCyte measuring system that dTAG-induced KAP1-degradation does not affect cell growth in the evaluated time frame.

Regarding cell divisions, HCT116 cells are not a slowly dividing cell line and have a doubling time of ~18 hours. Thus, during the 144 hours of cell confluence measurements (for the IncuCyte experiment), there are ~8 divisions and for the cell counting assay above, there are ~3-4 divisions. Across this number of divisions, there does not seem to be a major cell growth or density change as the data in the above experiment indicates.

Line 99-100. More information may be needed on how cell confluency was measured, including possibly some images.

Please see the response to item in the above comment labeled as 'Line 99-100' for a response to this point. Unfortunately, the images from the IncuCyte were removed after this experiment was conducted, and thus we only have access to the raw confluency data. We hope that presenting the cell counting will suffice as an orthogonal approach to show that acute KAP1 depletion does not lead to major growth defects in this cell line, barring the limitations of the system discussed above regarding near complete depletion.

Lines 103-104. Could a browser shot be shown for an example gene for TT-seq tracks?

Yes. Gene browser tracks for four genes (two IEGs and two non-IEGs) are shown below. The two IEGs shown were called as >1.5-fold differentially expressed upon dTAG treatment in homeostatic conditions.

Lines 113-116. The argument for minimal changes in protein expression leading to choosing the IEGs appears convoluted. Perhaps this is just wording.

We agree that the wording was unclear. Thus, we fixed this by writing the following on **pg. 4, lines 120-128**: *“These IEGs are either lowly or not expressed in basal conditions, but can be induced up to ~100-1000-fold upon cell stimulation to produce proteins required for several cell fate choices, suggesting that modest gene upregulation (<1.5-fold) upon acute KAP1 depletion may have minimal, if any, consequences on protein expression and cellular phenotypes. Given the prominent role of growth-factor signaling to cancer development, we decided to examine if and how KAP1 modulates the transcriptional response to cell stimulation (when IEGs are truly expressed), using the serum response network as a surrogate, which has been implicated in multiple tumorigenic phenotypes.”*

Sup figure 1D shows GO analysis of genes with FC>1.5. What about genes with 2,4,8,16 fold-change selected for further analyses? Are known IEGs being caught among these genes?

The GO plot (**Supplementary Fig. 1E**) and TT-Seq volcano plot (**Supplementary Fig. 1D**) are analysis conducted for the TT-Seq, which was performed to monitor nascent transcription changes in HCT116 cells with and without acute KAP1 depletion in homeostatic conditions (not serum-treated). The 2, 4, 8, and 16-Fold upregulated IEGs were identified from the RNA-Seq data treated with and without serum. The answer to the question posed by the reviewer is that yes, some genes identified in the RNA-Seq data to identify IEGs were indeed upregulated upon acute

KAP1 depletion in homeostatic conditions by TT-Seq. The fact that these IEGs were included in the original TT-Seq dataset prompted us to investigate KAP1-dependent serum responses (**pg.4, lines 120-128**), where we surprisingly found that loss of KAP1 leads to reduced transcription of IEGs upon serum treatment based on published datasets identifying serum-inducible genes (PMID: 20098423). A Venn diagram of 4-Fold IEGs (n=69 total) and genes >1.5-fold upregulated (n=91 total) upon acute KAP1 depletion in homeostatic conditions (TT-Seq) is below. Twenty genes belong to both categories.

Lines 142-144. The effect of IEG depletion is modest.

We **completely agree** and added modest in the text multiple times (see response to Reviewer 1 Point 1 above). We also want to highlight that many groups, which have used acute protein degradation rather than chronic knockout have also observed modest transcriptional phenotypes (see examples in response to Reviewer 1 Point 1). This appears to be the norm when using acute depletion systems for tuners of transcriptional regulation as opposed to critical upstream regulators such as the CDK9 kinase.

Sup Fig S2A is redundant with main Fig 2A.

We agree with this point. We have thus removed this Figure from the manuscript. Please see the response to the major comment to **Point 4**.

Sup Fig S2B tracks are not distinguishable from background unless extended outside in both directions. This comment is related to a major comment above.

We agree with this point and have fixed it. Please see the detailed response in the major **Point 4** above with new data analysis and browser tracks.

Lines 177-178. Authors should show control genes that are expression- and/or Pol II occupancy-matched to IEGs in basal conditions. Same applies to lines 220-223 (Fig 4).

We thank the reviewer for this comment to drastically improve our data presentation. We have now added a variety of control gene browser tracks for IEGs throughout the manuscript (**Fig. 2F, Supplementary Fig. 3F, Supplementary Fig. 4F, Supplementary Fig. 6E, Supplementary Fig. 7E**). We hope that inclusion of browser tracks for various control genes across multiple experiments (HA ChIP-Seq, Pol II ChIP-Seq, PRO-Seq, ChIP-Seq of all tested transcriptional regulators) including those defined in a few other points in this rebuttal letter (Reviewer 1 Points 3-4) provide ample occupancy-matched tracks to better define specificity for the defined phenotypes at IEGs.

Figure 4D. Control basal expression-matched genes should be shown.

This is a great comment and suggestion. Regarding **Fig. 4D**, it is not meaningful to plot the Proxy Rate for non-IEGs as these genes do not undergo major increases in gene body signal as IEGs do, thus the Proxy Rate analysis is **only an informative calculation for IEGs** and is **not** an informative calculation to measure elongation rate of all genes, which is certainly a limitation of the Proxy Rate approach. To address this concern and to investigate our PRO-Seq data more thoroughly for basal expression matched-genes, we have done the following:

1. In the initial submission, we plotted the PRO-Seq density profiles at basal expression-matched genes (labeled as non-DE genes) with and without KAP1 across the three serum time points (Serum 0 min, Serum 5 min, Serum 10 min) now presented in **Supplementary Fig. 4H-J**. This data shows that there are minimal, if any, genome-wide perturbations to active Pol II occupancy in the basal or serum-treated state and that IEG's are selectively perturbed upon acute KAP1 depletion.
2. We have now added PRO-Seq plots (under Pol II ChIP-Seq data to show that these are Pol II occupancy-matched) of a control gene to the main manuscript in **Supplementary Fig. 4F**.
3. Below, we have also now plotted the nascent transcription levels of a random set of genes ($n=69$) selected from the non-DE list of uninducible genes. Importantly, dTAG treatment does not significantly perturb the nascent transcription of basal expression-matched genes, highlighting the specificity of KAP1-dependent regulation on IEGs.

Figure 5. Example browser shots should be shown in main or supplementary figure.

We thank the reviewer for this suggestion to improve the visualization of our data. We have now added browser tracks for multiple genes, including two representative IEGs: *FOS* (**Fig. 5G**) and *ATF3* (**Fig. 5H**), and one control gene (*FUS*) (**Supplementary Fig. 7E**). An additional control gene (*NXF1*) is plotted above in the response to Reviewer 1 Point 3.

Wording:

Line 121 starved from serum should be "...of serum"?

We thank the reviewer for the suggestion. This sentence has now been revised to the following (see **pg. 4, lines 131-134**): "To evaluate if KAP1 regulates signal-induced transcription (specifically the serum response network), cells were initially serum starved, treated with dTAG (8 hr) to deplete KAP1 (or DMSO vehicle control), and later replenished with serum-containing media to induce IEG expression for various treatment time points (**Fig. 1C**)."

Line 123 “diminished the temporal expression” – the meaning of this wording is unclear.

We thank the reviewer for the suggestion. We have now changed the wording to the following (Pgs. 4-5, lines 134-139): “Acute KAP1 depletion modestly decreased the expression of candidate IEGs (e.g., FOS, ATF3, NR4A1, and EGR1) (ranging from ~8-35% depending on the gene and time point), with no major changes to a control gene (U6), suggesting that KAP1 is required for full transcription activation of this subset of IEGs upon serum stimulation (Fig. 1D; Supplementary Fig. 1F).”

Line 204 by increasing kinetics, did authors mean increasing elongation velocity or speed?

Yes, the data overall indicate that Pol II elongation during early serum stimulation time points is faster upon acute KAP1 depletion. As explained in detail in the responses to Points 1 and 2 above, with the IEGs length limitations we were unable to accurately measure elongation rates, but have provided an estimate using the Proxy Rate method. Thus, we used the term kinetics to define that acute KAP1 depletion increases the movement of Pol II molecules through the gene body in a kinetic manner as signal-inducible genes are turned on. While velocity and speed appear synonymous, we personally choose “kinetics”, rather than “rate”, “velocity”, or “speed”, due to the limitations of the Proxy Rate method.

Lines 209-210 “actively synthesizing increased magnitudes of transcription”

We have fixed this sentence to read: “Given that ChIP-Seq detects a combination of different kinds of Pol II molecules (e.g., paused, active, backtracked), it is difficult to assess if the increases in Pol II density at the GB regions and 3' ends in the absence of KAP1 are indicative of a faster moving, actively transcribing Pol II versus a slower moving Pol II that is being retained on chromatin for longer times.”

Line 270 “increases Pol II elongation kinetics” – is velocity meant?

Please see the response to the above comment labeled as **Line 204** as the response is similar.

Reviewer 2

In this study, Hyder et al. investigate effects of acute depletion of KAP1 (also known as TRIM28), a factor previously implicated in regulation of RNA polymerase II (Pol II) elongation whose precise function remains unclear. By introducing an inducible degron, or dTAG, in frame with KAP1, they achieve efficient degradation of KAP1 within ~8 hr of small-molecule addition in HCT116 colon cancer cells. The effects on global transcription, measured by transient transcriptome sequencing (TT-seq), are modest, but among genes that are upregulated is a subset of serum-inducible, immediate early genes (IEGs). The authors investigate the response to serum stimulation in the presence or absence of KAP1, and find that, while expression of IEGs was attenuated at both mRNA and protein levels by depletion of KAP1, Pol II elongation kinetics, measured by precision run-on transcription sequencing (PRO-seq), was accelerated; when KAP1 was degraded, Pol II transcribed farther into the bodies of IEGs by 5 and 10 min after serum addition. How this might lead to reduced steady-state IEG expression is suggested by the reduced occupancy of multiple components of the Pol II transcription machinery, including initiation and elongation factors and Pol II itself, at later time points (30 min). The authors posit a direct causal connection: faster elongation at early time points leading to increased accumulation or aberrant retention of elongation complexes in 3' gene regions, impairing Pol II recruitment to the promoter region for additional rounds of transcription. This is an attractive model, which might fit with previous studies indicating connections between termination and initiation, but the data presented do not rule out (and the authors do not discuss) alternative, more indirect mechanisms. How KAP1 is influencing Pol II elongation rate (negatively, according to the model) is addressed only speculatively in the Discussion. The core findings of the study are interesting and will be of considerable interest to the field, so I would recommend publication after the authors have tested at least one of the more likely mechanisms for effects on Pol II elongation speed or impaired initiation (see below) and toned down some of their stronger mechanistic conclusions.

We thank the reviewer for stating that the core findings of our study are interesting and potentially of considerable interest to the field. Below we fully address all major concerns and specific points to improve the manuscript and figures accordingly.

My general concerns

Point 1. Throughout, the authors are a bit careless with terms that, depending on context, can mean differences in either time or space (e.g. “downstream,” “early/late”). Because both time (after serum stimulation) and distance (into gene bodies) are critical variables here, it is important to avoid such ambiguity. I have tried to flag instances where things get confusing in my specific comments below, but I urge the authors to go over the manuscript carefully with an eye towards avoiding ambiguous or imprecise language.

We thank the reviewer for this important comment which has prompted us to drastically improve our communication. Not only did we correct the points below in the ‘specific concerns and comments’ sections, but we also extensively reviewed the manuscript and made several changes to improve readability. We replaced “early” and “late” when referring to the time point of serum to “early stimulation time point (15 min)” vs “late stimulation time point (30 min)” by clearly describing the time of treatment used. This avoids confusions with related terms if not properly described such “early vs late elongation” and “early vs late in gene bodies”. We think this reduces ambiguity and helps improve readability.

We specifically made the changes in the following additional places, along with many others in the revised manuscript:

1. **pg. 6, lines 214-218:** Additionally, signal quantitation in GB regions revealed that while 180 min of serum stimulation began to redistribute KAP1 away from GB regions of 16-Fold IEGs, 4-Fold IEGs still had increased KAP1 density at their GB regions (**Fig. 2D**),

potentially due to this lowly expressed IEGs cluster having longer kinetics of activation at the 30 min serum stimulation time point.

2. **pg. 7, lines 247-250:** Given that signal-induced transcription is dynamic, Pol II ChIP-Seq was performed before (0 min) and after two serum stimulation time points (15 and 30 min) with dTAG treatment or vehicle DMSO control.
3. **pg. 8, lines 279-282:** Collectively, increased Pol II occupancy upon KAP1 depletion at an early stimulation time point (15 min) led to decreased Pol II occupancy at a late stimulation time point (30 min) ultimately dampening IEGs expression, consistent with other metrics such as transcriptome and protein expression analyses (**Fig. 1**).
4. **pg. 8, lines 284-285:** We changed the header section to add 'time points': Acute KAP1 depletion increases Pol II elongation kinetics during early serum stimulation time points.
5. **pg. 8, lines 302-305:** Strikingly, active Pol II had reached further into the GB regions of IEGs at 5 min serum stimulation upon KAP1 depletion (**Fig. 4A**, see zoomed in metaprofile of the GB region), and its density increased towards the end of the GB regions and 3' ends at 10 min serum stimulation (**Fig. 4B**).
6. **pgs. 11-12, lines 454-457:** Given that the increased elongation at early serum stimulation time points upon acute KAP1 depletion diminished IEGs transcription and Pol II recruitment at late serum stimulation time points, we predicted a model whereby KAP1 depletion indirectly impacts assembly of transcription machinery at gene promoters at late serum stimulation time points.

Point 2. As noted above, the authors reach for the simplest and most attractive mechanism to reconcile their “counterintuitive” results (increased elongation rate directly causing diminished initiation). An indirect effect, mediated by a signaling pathway that prematurely dampens IEG expression when the initial round of transcription is aberrant, say, by turning off a transcriptional activator, is also possible (and might also lead to reduced occupancy of Pol II machinery at the promoter). The authors should at least discuss such possibilities.

We thank the reviewer for these comments. We fully agree with the reviewer that any notable alterations to signaling pathway activation, such as turning on or off of a transcriptional activator, will largely impact transcription activation. We also want to clarify that we fully agree that signal-regulated transcription activators may be turned off when the initial round of transcription is aberrant (as we observe with the general transcription machinery upon KAP1 depletion at early time points of serum stimulation) and that this phenomenon may be contributing to the reduced occupancy of Pol II machinery at the promoter-proximal regions of IEGs. This is in line with evidence that the occupancy of many general transcription factors, including CDK9, SPT5, CDK7, MED1, and TFIIB are reduced at the 30 min serum stimulation time point upon KAP1 depletion, suggesting that, in general, acute KAP1 depletion largely reduces occupancy of the Pol II machinery at the promoter-proximal regions of IEGs at the 30 min serum stimulation time point.

To address the likelihood of an indirect effect driving this secondary transcriptional phenotype, we examined whether critical signaling events, such as site-specific phosphorylation of the master regulator Elk-1, were aberrantly altered upon acute KAP1 depletion. Elk-1 is a component of the ternary complex that binds the serum response element and mediates gene activity in response to serum and growth factors (PMID: 8386592, PMID: 7834740). Elk-1 is directly phosphorylated by MAP kinase pathways at a cluster of S/T motifs, particularly Ser383, which is critical for transcriptional activation. While serum stimulation promotes parallel signaling events, Elk-1 phosphorylation (pElk-1) is one of the most critical steps for serum-induced transcription activation (PMID: 8386592, PMID: 7834740, PMID: 20098423). Thus, to test the possible contribution of signaling events to the long-term transcriptional phenotypes (decreased IEGs transcription upon acute KAP1 depletion), MAP kinase mediated site-specific phosphorylation (Ser383) of the master transcriptional regulator Elk-1 was monitored throughout a serum stimulation time course in cells with and without acute KAP1 depletion. Remarkably, we found no changes in the kinetics of serum-induced site-specific pElk-1 during the 5-30 min time course in cells with and without acute KAP1 depletion. KAP1 and c-Fos western blots are included as controls to show KAP1 depletion and the expected reduced c-Fos protein expression upon KAP1 depletion. The new data is in **Supplementary Fig. 1G** and discussed in the text (**pg. 5, lines 161-178**). Together, the new data suggest that acute KAP1 depletion does not alter critical cell signaling events at the bulk level that initially turn on and later sustain IEGs transcription during serum stimulation.

Because the above data provides bulk level information, we attempted to monitor pElk-1 interaction with IEGs upon 15 min serum stimulation (time point selected from the bulk western data in **Supplementary Fig. 1G**) using ChIP-Seq with an antibody previously validated by ChIP-qPCR (Fig. 4A of PMID: 20098423). However, we recovered extremely low levels of DNA that when sequenced did not allow us to retrieve any useful information regarding pElk-1–promoter interactions. The browser tracks below show very low signal-noise ratio at the promoters of two representative IEGs (*FOS* and *FOSB*) relative to nearby background noise. Overall, we are unable to assess if defects to transcription observed at early time points of serum (increased elongation) also affect the occupancy of signal-specific transcription activators on chromatin. However, we posit that this mechanism is possible, and not something that disrupts our model that KAP1 negatively regulates elongation kinetics to activate signal-induced transcription. As indicated by the reviewer, we have added this possibility to the Discussion section (**pg. 15, lines 583-589**).

Point 3. In the Discussion, the consideration of mechanisms by which KAP1 might influence elongation rate curiously omits the one known (and potentially testable) mechanism for regulating Pol II elongation speed: phosphorylation of the SPT5 subunit of DSIF by CDK9 (ref. 26, as well as doi: 10.1038/s41467-018-03006-4 and doi: 10.1038/s41586-018-0214-z). Also, although probably beyond the scope of this study, one way to test the favored model for how KAP1 depletion is negatively affecting IEG expression would be to ask if this phenotype can be rescued by expressing a “slow” mutant Pol II (see ref. 68).

We thank the reviewer for these important comments, and we apologize for our oversight in missing dissection of SPT5 phosphorylation as a potential model for KAP1-induced increases in transcription elongation in our original discussion. We fully agree with the reviewer that KAP1 could influence (directly or indirectly) elongation through SPT5 phosphorylation by CDK9. Since our collective evidence supports a model in which Pol II elongation increases during the early serum stimulation time points, we have now performed ChIP for both total SPT5 and site-specific phosphorylated SPT5 (Thr806 in the CTR1 domain, referred to as pSPT5) at the early serum time point (5 min) to test if increased Pol II elongation rate upon acute KAP1 loss is either correlated with or caused by SPT5 phosphorylation. While ChIP-Seq of total SPT5 and pSPT5 revealed that their occupancy increased upon 5 min of serum treatment in DMSO vehicle control, acute KAP1 depletion led to no major increases in occupancy, suggesting that KAP1’s function in regulating elongation rate may be independent of SPT5 phosphorylation at this one residue. This data is now presented in **Supplementary Fig. 6B-E** and is described in the text (**pg. 11, lines 427-445**) and in the Discussion in the text (**pg. 13, lines 504-515**). While we didn’t observe changes to Thr806 phosphorylation levels, we do not discount that the phosphorylation levels of other SPT5 residues in the CTR1 or the less-characterized CTR2 domain (PMID: 36206739, PMID: 32859893, and PMID: 10757782) may be perturbed upon acute KAP1 depletion. Future studies beyond the scope of this study, should include extensive and thorough investigation of SPT5 site-specific phosphorylation events at early serum stimulation time points. Given that precisely defining the mechanisms will take years to solidify a model, as indicated by the reviewer, we choose to tone down the mechanistic conclusions. Finally, we fully agree with the reviewer that expressing slow Pol II mutants, such as Pol II funnel domain mutant R749H, can be used to test the favored model, but these future studies can be complicated if KAP1 is maintaining elongation kinetics by mechanisms other than restricting Pol II directly.

Specific concerns and comments:

4. **Lines 17-8:** As an example of point 1 above, to many in the transcription field, an “early elongation defect” means a defect in elongation that occurs soon after release from the promoter-proximal pause, but here it refers to a defect in elongation on IEGs that was detected soon after cells were stimulated with serum.

We have now clarified the abstract to resolve the reviewers important point. We hope the new clarification allows us to better clarify that we are reflecting to defects observed soon after cells are stimulated with serum.

5. **Lines 17-20:** More substantively, this passage infers causation where there is only evidence for correlation between genes where elongation kinetics are perturbed at early time points but expression is dampened at later times.

We agree with the reviewer and thus have now revised the abstract appropriately.

6. **Line 28:** “Briefly” should probably be changed to “Shortly” or Soon.”

We have changed “briefly” to “Shortly” in **pg.2, line 30**.

7. **Line 40:** SPT5 is a subunit of DSIF; referring to it as “SPT5/DSIF” is confusing (the same nomenclature is used earlier in the paragraph to describe a complex between two different proteins, CDK9/CycT1).

We have now corrected both references in this point on **pg. 2, lines 37-42**: *“Pause release is facilitated by the positive transcription elongation factor (P-TEFb) kinase (composed of CDK9/CycT1), which is recruited by several complexes in different contexts and phosphorylates both pausing factors (DSIF and NELF) and the Pol II carboxy-terminal domain (CTD) at the Ser2 position. After pause release, transcription proceeds with many factors binding Pol II to regulate elongation rate, including SPT5 (a subunit of DSIF), SPT6 and PAF1C.”*

8. **Lines 99-101:** A minor point, but to conclude that the lack of effect of dTAG treatment on cell proliferation over 6 days is meaningful, the authors should provide immunoblot data to ensure that KAP1 proteins levels remain low over the extended time course.

We agree with the reviewer. We performed an extended immunoblot dTAG time course (1-2-3-6 days) and observed that indeed KAP1 protein levels are sustained at low levels over this time course. This is now included in **Supplementary Fig. 1C**.

9. **Line 128:** Also minor, but “we treated cells +/- 30 min serum +/- dTAG” is lab notebook shorthand and should be expanded and better explained.

We agree with the reviewer. We have improved the wording with the following in **pg. 5, lines 140-142**: *“To assess this function globally, serum-starved cells were pre-treated with dTAG or vehicle DMSO control and later challenged with serum (30 min) to induce IEGs or remained untreated.”*

10. **Lines 151-6:** The metagene profile of KAP1 ChIP-seq signals in Fig. 2a is indeed reminiscent of Pol II and other factors thought to “travel” with the elongation complex, but it would help to show a direct comparison/superimposition (for example, with the profile in Fig. 3b), perhaps in a supplementary figure.

We thank the reviewer for this comment. We have now superimposed the KAP1 ChIP-Seq signal below Pol II data metagene profile in **Supplementary Fig. 3B**. We have also superimposed HA ChIP-Seq browser tracks with PRO-Seq and/or Pol II ChIP-Seq for the following representative IEGs: *NR4A1 (Fig. 3F)*, *ATF3 (Fig. 3G)*, and *FOSB (Fig. 4D)*.

11. **Lines 185-90:** Another minor point, but in describing the data in Fig. 3, comparing the Pol II occupancy data from “4-fold” and “16-fold” IEGs, the authors confusingly state the values as a range, e.g. “7-14% for 4-fold and 16-fold, respectively” and so forth. It would be better to say “7% and 14%.”

We thank the reviewer for this point to clarify our work. We have improved the writing to the following in **pg. 7, lines 254-260**: “Unexpectedly, KAP1 depletion leads to divergent Pol II occupancy phenotypes at IEGs at the two evaluated serum stimulation time points (15 and 30 min) (Fig. 3A, B). At 30 min stimulation, Pol II levels across IEGs modestly decreased upon KAP1 depletion in PP regions (7% for 4-Fold IEGs and 14% for 16-Fold IEGs), GB regions (7% for 4-Fold IEGs and 18% for 16-Fold IEGs), and 3' ends (1% for 4-Fold IEGs and 8% for 16-Fold IEGs) (Fig. 3B, C, D, E, superimposed metagene with HA ChIP-Seq in Supplementary Fig. 3B), in line with decreased IEGs expression by RNA-Seq at this time point (Fig. 1).” We hope this clarifies the points trying to be made and accurately separates the percent changes at the various IEGs clusters.

12. **Lines 190-1:** This may be more of an aesthetic judgment, but the genome browser tracks in Supp. Fig. 3E make a stronger visual case for altered elongation kinetics than do the metagene or box plots, and should perhaps be moved to the main figure.

We thank the reviewer for noticing this important aesthetic judgment that has allowed us to increase the presentation of our data. We have now moved the gene browser tracks into the main figures as **Fig. 3F and G**.

13. **Lines 224-6:** In this sentence, describing data from 5- and 10-min time points, “late” is used to refer to a positional effect (“in the GB”) and, in the next breath, “early” is used to refer to a 15-min data point.

We thank the reviewer for this glaring oversight in text. We have fixed it with the sentence in **pg. 8, lines 302-305**: “Strikingly, active Pol II had reached further into the GB regions of IEGs at 5 min serum stimulation upon acute KAP1 depletion (Fig. 4A, see zoomed in metaprofile of the GB region), and its density increased towards the end of the GB regions and 3' ends at 10 min serum stimulation (Fig. 4B).”

14. **Lines 241-52:** This passage, describing first the limitations of PRO-seq data obtained during IEG induction for calculating elongation rate and then a computational strategy for extracting an “elongation rate proxy” from the data, could be revised to increase clarity. First, the gene-length requirement (2 kb per minute of treatment) is not explained (I presume it refers to the estimated elongation rate of Pol II *in vivo*). Second, what is the relationship between the calculated “elongation rate proxy” and the actual elongation rate? Can the authors extrapolate an approximate value for the increase in elongation rate when KAP1 is degraded? Is this effect actually larger on “16-fold” IEGs than on “4-fold” IEGs, as Fig. 4d seems to suggest?

We thank the reviewer for this comment and apologize for our oversight in clarity at defining and validating the ROCC (Proxy Rate) approach. We have thoroughly revised the manuscript to include a detailed Proxy Rate section (**pgs. 9-11, lines 327-420**) to increase clarity about how the Proxy Rate is quantified, provide validation of the approach against a characterized dataset, and clarify its limitations.

1. Gene-length requirement: The gene length estimate in the initial manuscript was indeed referring to the estimated elongation rate of Pol II *in vivo* (in cells) based on published datasets that have estimated elongation rate. Because this estimate is different across different gene sets, particularly inducible genes (PMID: 23523369), we decided to leave the 2-kb per minute statement out of the revised manuscript (and we thank the reviewer for bringing light to this issue). The important point for the gene length requirement is that short genes and the current standard best approach for estimating these rates (amounting

to using a procedure to call either the position of the “leading edge”, and then regressing a line through these positions on time) does not allow for estimation of an elongation rate when genes are so short that the “leading edge” has advanced beyond the end of the gene prior to any observation occurring. Thus, short IEGs present a problem at precisely measuring elongation rate. We hope the discussion of the gene-length requirement on **pg. 9, lines 329-347** will help bring clarity to this point.

2. Relationship between the calculated “elongation rate proxy” and the actual elongation rate: The Proxy Rate measurement is merely an inference of elongation rate and is defined by quantifying coverage differences between DMSO- and dTAG-treated cells across the 5 and 10 min serum stimulation time points. Because of this, the actual Proxy Rate value does not generate a single “leading edge” or distance that Pol II has traveled (analogous to DRB washout experiments), and thus cannot calculate an actual rate (kb/min). Thus, we cannot extrapolate an approximate value for the increase in elongation rate when KAP1 function is lost, but can only infer that the distance Pol II has traveled when KAP1 is depleted is likely further into the gene. This is now discussed (**pgs. 9-10, lines 348-369**).
3. Effect on 16-Fold versus 4-Fold genes: The median rate for 4-Fold genes are as follows:

Gene cluster	4-Fold IEGs		16-Fold IEGs	
Condition	DMSO	dTAG	DMSO	dTAG
Median Rate	0.8796	0.9959	3.718	5.284

Thus, while we are unable to calculate rates for these identified gene subsets, there is a ~13% median increase at 4-Fold IEGs when KAP1 is depleted while there is a ~42% median increase for 16-Fold IEGs, suggesting that indeed the phenotype is stronger for the most inducible cluster of IEGs.

15. Fig. 5c-f and Supplementary Fig. 5c-e: Could the authors provide more detail on how the CHIP-seq data were normalized? Specifically in the case of pS5 and pS2 of the Pol II CTD, were these signals normalized to total Pol II (which is generally more useful than “raw” signals)?

Regarding CHIP-Seq data normalization, we have included a description in the methods of the manuscript under “**CHIP-Seq analysis**” as well as a detailed response to Reviewer 1 Point 3 above. Regarding specifically the pS5 and pS2 Pol II CTD ChIPs, the initial submission contained pS2/pS5 ChIP-Seqs that were read-depth normalized “raw” signals.

We thank the reviewer for this comment that normalized signals are more meaningful and have now normalized our pS2/pS5 Pol II data to total Pol II below. We observed that, upon normalization, signal densities for pS2 and pS5 were still modestly lower in dTAG than DMSO, suggesting some potential additive effects to Pol II CTD phosphorylation. Normalized ChIP-Seq plots are now presented in **Supplementary Fig. 7F-G**. Metagenes plots so the reviewer can compare are below:

16. **Lines 266-8:** This sentence doesn't make sense as written, first mentioning "reduced occupancy" and then "stronger increases." Do the authors mean "decreases?"

We apologize for this oversight. Yes, the reviewer is right that this was a mistake and "decreases" was meant. The sentence has now been revised to improve readability. See pg. 12, lines 463-465: "Notably, metagene analysis and signal quantitation at PP regions shown that KAP1 depletion led to modestly reduced occupancy of every factor tested, albeit at different levels (Fig. 5A, B, C, D, E, F, Supplementary Fig. 7A, B, C, D)."

17. **Lines 270-2:** This concluding sentence of the results section needs to be toned down, in my opinion, e.g. by qualifying "dampens" with "might" and "explaining" with "possibly."

We have now qualified the statement in pg. 12, lines 476-479: "Collectively, these data suggest that alterations of Pol II elongation kinetics, which are directly elicited by loss of KAP1 transcriptional control during early serum stimulation, potentially dampens the recruitment of the transcription apparatus, including Pol II, at late serum stimulation, possibly explaining the overall decrease in IEGs expression levels."

18. **Lines 282-5:** The authors invoke delayed termination or release of Pol II from 3'-ends of IEGs as a possible explanation for a "reinitiation" defect. Have they analyzed their PRO-seq or Pol II ChIP-seq data for evidence of transcriptional read-through or increased Pol II occupancy beyond the normal termination zone, which would support such a mechanism?

We thank the reviewer for this comment. Prior to submission, we had not quantified our data to test this hypothesis. To test it with our existing data, we utilized the PRO-Seq data as it quantifies engaged Pol II molecules and calculated the "Readthrough Index" (hereafter referred to as RI) (characterized in PMID: 36309014) in which signal density 2000-bp after the pA site is divided by signal density before the pA site. The reason for this normalization is to discount any effects observed at the elongation stage that may impact termination. If acute KAP1 depletion led to increased readthrough specifically over the already increased elongation impact, the expectation would be that the RI would be greater in the dTAG treatment compared to DMSO. Quantification of the RI, however, shows that for most of the clusters and time points (5 and 10 min), there is no significant increase in RI despite the increased in 3' end signal. However, for the 16-fold IEGs, we did observe a significant increase in RI at the serum 10 min time point. While this could be pointing to a termination defect, we have to acknowledge that this increase in RI could be a consequence of increased elongation in gene bodies at earlier time points (seen at 5 min). Additionally, at later serum stimulation time points transcription decreases upon acute KAP1 depletion, which poses a kinetic problem where potentially the overall decreases in Pol II have

caught up in the gene body but not in the 3' end. For now, it is difficult to abstract any meaning from this result, but future investigations will need to particularly examine how, and if any, direct transcription termination defects are occurring upon KAP1 depletion (potentially using Xrn2 mutants defined in PMID: 26474067).

19. **Lines 308-9:** The authors should say explicitly how their “phenotypes differ from these previous observations.” Without that context, I can’t assess the argument that follows.

We thank the reviewer for bringing insight into this lack of clarity. When we say previous observations, we are mainly referring to the PP1-XRN2 mechanism from PMID: 31677974 (Fig. 5G of this paper) in comparison to our data in **Fig. 3A** and **Fig. 4A-B**. To add context, we have altered the writing of the paragraph in **pgs. 14-15, lines 561-573**, which we are also copying below: “*Our study exposed yet another layer of precise elongation control that differentiates from previous mechanisms revealing that Pol II deceleration in the 3' end is important for proper termination and avoidance of readthrough transcription. In these studies, Pol II deceleration by protein phosphatase 1 (PP1) in the 3' end was critical for the formation of the 3' pause site and for enabling Xrn2 (the 5'-3' RNA exonuclease) to catch up to Pol II for normal eviction after degradation of the exposed nascent RNA post-cleavage. In our dataset, we do not observe striking increases in readthrough Pol II upon KAP1 depletion, but rather an increased in “piled up” paused Pol II in the 3' end (Fig. 3A), overall suggesting that 3' pausing is established but that Pol II molecules reach the 3' end at an earlier time when KAP1 is depleted, potentially owed to increased elongation rate in the gene body. Given that our phenotypes differ from observations with the PP1-Xrn2 mechanism, it appears KAP1 may not function with known regulators of Pol II deceleration in the 3' end (e.g., PP1).*” We hope this addition now clarifies how our data differs from the PP1 story, but at the same time argue that further research beyond the scope of this study will be needed to define the molecular details.

20. **Lines 310-5:** Failure to resolve R-loops is invoked as a possible explanation for the KAP1-deficient phenotype of increased Pol II occupancy at IEG 3' ends. Have the authors tried to ChIP for RNA:DNA hybrids?

We completely agree with the reviewer that failure to resolve R-loops could very well be an explanation for the observed transcriptional defects at IEGs when KAP1 is depleted. We have not yet measured RNA:DNA hybrids for this manuscript, and view this as an important and testable model for the future that could explain an indirect control in R-loop fidelity affecting elongation kinetics and/or termination.

21. **Line 363:** Another, potentially confusing use of “early Pol II elongation kinetics” could be fixed by deleting “early” and inserting “during prior rounds of transcription” after “kinetics.”

We wholeheartedly agree with the reviewer here. We have corrected the sentence exactly as the reviewer described in **pg. 13, lines 499-503**: *“Our studies support the notion that at least some of these molecular events are potential indirect consequences of chronic KAP1 depletion, which dampens long-term transcription initiation and elongation because of abnormal increases in Pol II elongation kinetics during prior rounds of transcription, consistent with our findings that the transcription apparatus is largely deregulated during late serum stimulation time points (Fig. 5).”*

22. Lines 369-70: It is not clear which “other factors” this sentence refers to.

We apologize for this oversight. We meant general transcription factors of the pre-initiation complex, as well as more “global” regulators of elongation (e.g., CDK9 and SPT5), along with factors such as PAF1 and Integrator. We have added the following clarification to the sentence in **pg. 15, lines 583-585**: *“Another question that remains to be addressed is why KAP1 specifically regulates transcription of IEGs, but not global transcription, like other factors do (e.g. Pre-initiation complex, SPT5, CDK9).”*

23. Lines 393-4: Again, I would avoid using “early transcriptional checkpoint” (especially when followed by “downstream consequences”)—terminology that has previously (and more commonly) been applied to checkpoint-like controls that are imposed within individual transcription cycles (see ref. 19 and the aforementioned doi: 10.1038/s41467-018-03006-4 for examples).

We thank the reviewer for this suggestion. We have clarified this by writing the following in **pg. 15, lines 611-615**: *“Overall, KAP1 plays a “repressive” role to maintain Pol II elongation kinetics. Alterations in this regulatory mechanism at early stimulation time points elicit downstream consequences in the transcriptional cycle ultimately dampening signal-induced transcription.”*

REVIEWERS' COMMENTS

Reviewer #1 (Remarks to the Author):

The revised R1 manuscript by Hyder et al nicely addresses experimental questions and this reviewer has no concerns with the data. The authors demonstrate convincingly that KAP1 travels along the elongating Pol II into gene bodies and that its depletion leads to subtle yet significant perturbation of transcription elongation that is most salient at Immediate Early Genes and has a temporal pattern which suggests that the involvement of KAP1 in elongation is transient and, therefore, may be related to regulating the timing of transcriptional responses to stimuli. These parts of the manuscript are now excellent.

Mechanistic interpretations offered by the authors are less certain even after revision.

Major points.

One remaining point has to do with what might be perceived as authors' over-reliance on calling the effect of KAP1 on elongation as direct. That KAP 1 travels with Pol II is a good piece of information, but it does not indicate direct effects including direct interaction with the elongating Pol II. An argument that other (known) signaling events are not altered after KAP1 depletion is good, but also is not direct. This point is already rightfully acknowledged in discussion so it should be left up to interpretation in results. This point is important, but may be potentially addressable by toning down the wording in the manuscript and/or explaining what authors mean by "direct" effect.

A second point is about ascribing the effect of KAP1 depletion to elongation. The arguments presented in new Figure 3 show a modest increase in transcription signal within gene bodies. One can argue that the same effect would be observed when another, potentially limiting upstream step, such as initiation or release, is sped up. The resolution of current approaches might not be able to distinguish these conclusively. One should also not discount potential effects on possible redirection of paused Pol II between elongation and premature termination. Making conclusive statements requires addressing all these mechanisms. The ChIP with GTFs is not conclusive since changes in promoter occupancy per time unit may lead to unpredictable changes in ChIP signal of some components. Authors currently do not present evidence reasonably excluding the effect of KAP1 on other steps of transcription. This includes the proposed methods to devise elongation kinetics by proxy that remains not well validated at this point. A modest positive correlation comparing outcomes using previously published elongation wave measurements make it somewhat possible, but not beyond reasonable doubt just yet.

One way to address these points is to tone down interpretation of data within Results section and offer interpretation as part of discussion, acknowledging limitations.

Minor points.

Line 14 and elsewhere in the text. What do authors mean by “high-resolution” approaches? Currently the data are interpreted at a gene body level max, as any high-resolution information (such as in line 353) is not used as such without averaging. Deleting this wording does not diminish anything. The same may apply to “in real time” (line 81): nascent sequencing is not necessarily real-time, but is a destructive technique that either is a pulse over time (TT-seq) or requires permeabilization of cells for a time at non-physiological temperature (PRO-seq), either way hardly real time (like live microscopy, for example). Wording should be amended or clarified.

Line 99 – delete “near complete”, as 90% depletion is not necessarily near complete and should be left for the reader to choose to interpret as such or not.

Do lines 125-128 better belong to just before 131 in the next section?

Line 243 and the rest of the section. The word “divergent” may be confusing as applies to transcription, as it by now reserved for bidirectional transcription, but is being used here for something else. Could authors find another way to describe changes in kinetics?

Authors still use “elongation kinetics” wording in a way that may be confusing, as the word has a broader meaning than speed. If authors mean elongation speed or elongation velocity, this substitute should be made as appropriate.

Line 159 – should “were” be “was”?

Line 206 – “signifying KAP1” should be “signifying that KAP1”

Lines 300-314. The effect on elongation does not seem to rise to the level of “strikingly”. This is because the timing at which this change in Pol II distribution is observed (5 minutes) is much, much slower than what it would have taken one Pol II to elongate through the gene. The effects observed are, therefore, most probably arise from a sum of many molecules in a quazi-equilibrium and the shape of this meta distribution does not necessarily directly reflect the speed of elongation.

Reviewer #2 (Remarks to the Author):

This is a revised version of a manuscript I reviewed previously. The authors have addressed all of my concerns and, in my opinion, the paper is now suitable for publication in Nature Communications. (I did catch one incorrect figure callout, in line 466, to a non-existent Fig. 4G,H, which should Fig. 5G,H.)

Point-by-point responses to the reviewers' critiques (Resubmission Round 2)

Reviewer 1

The revised R1 manuscript by Hyder et al. nicely addresses experimental questions and this reviewer has no concerns with the data. The authors demonstrate convincingly that KAP1 travels along the elongating Pol II into gene bodies and that its depletion leads to subtle yet significant perturbation of transcription elongation that is most salient at Immediate Early Genes and has a temporal pattern which suggests that the involvement of KAP1 in elongation is transient and, therefore, may be related to regulating the timing of transcriptional responses to stimuli. These parts of the manuscript are now excellent. Mechanistic interpretations offered by the authors are less certain even after revision.

We sincerely thank the reviewer for suggesting that the revised manuscript has addressed all experimental questions posed in the initial manuscript submission. We also thank the reviewer for shedding light on a few other concerns regarding interpretations that we now have addressed in this second resubmission, which includes an entirely new section labeled as "Limitations of the study" within the Discussion (pg. 16 of the revised manuscript) highlighting every point brought up by the reviewer. Finally, we agree with the reviewer that the mechanistic details by which KAP1 slows Pol II elongation remain unclear and will take years of careful biochemical, genetics and genomic experiments to come to light.

Major points.

Point 1. One remaining point has to do with what might be perceived as authors' over-reliance on calling the effect of KAP1 on elongation as direct. That KAP1 travels with Pol II is a good piece of information, but it does not indicate direct effects including direct interaction with the elongating Pol II. An argument that other (known) signaling events are not altered after KAP1 depletion is good, but also is not direct. This point is already rightfully acknowledged in discussion so it should be left up to interpretation in results. This point is important, but may be potentially addressable by toning down the wording in the manuscript and/or explaining what authors mean by "direct" effect.

We partially agree with the reviewer. We utilized the term "direct" when referring to the salient point that our study uses **acute KAP1 depletion** to record immediate transcription phenotypes, and that the observed molecular alterations are not due to secondary effects observed upon long-term KAP1 depletion. This is discussed in the manuscript: "Given KAP1 regulates many critical processes (Ref. 50, Randolph and Hyder et al. Front Cell Infect Microbiology 2022, PMID: 35281453), precisely defining KAP1's roles in signal-induced transcription requires advanced genetic approaches to directly define the underlying molecular mechanisms and help mitigate confounding indirect effects accrued upon long-term factor elimination." Therefore, importantly, the combination of the acute genetic system coupled with signal-induced KAP1 recruitment to IEGs directly showing that KAP1 travels with Pol II upon stimulation and our previously reported "direct" KAP1 and Pol II interaction (Ref. 45, Bacon et al., Mol Cell 2020, PMID: 32402252), together suggest a direct KAP1 involvement on Pol II transcriptional regulation. Given the reviewer's critique, we have discussed this point in the "Limitations of the study" section within the Discussion and toned down the "direct" claim throughout the manuscript as follows:

Pgs. 5-6, Lines 177-182: "Overall, these data indicate that KAP1 is required for full activation of IEGs upon serum stimulation, and that acute KAP1 loss does not alter key cell signaling events of the serum response network that initially turn on and later sustain IEGs transcription during serum stimulation, together suggesting KAP1 is a **direct** regulator of IEGs transcription. Below, we investigate the functional consequences of IEGs transcriptional regulation upon acute KAP1 depletion prior to and after cell stimulation with serum."

Pg. 6, Lines 186-190: To assess if KAP1 is recruited to IEGs to **directly** control their transcription, we performed Chromatin-ImmunoPrecipitation (ChIP)-Seq using an HA antibody

directed against the C-terminal dTAG epitope (Fig. 1A) before (0 min) and after two serum stimulation time points (30 and 180 min), in addition to a dTAG sample treated with 30 min of serum to ensure KAP1 ChIP-Seq signal specificity.

Additionally, we emphasize in the Discussion section that despite having convincingly demonstrated that KAP1 is on chromatin in a signal-dependent manner, we currently do not know how KAP1 regulates Pol II elongation (discussed in detail in **pgs. 13-14, lines 502-540**) and whether this is via direct or indirect means. We highlight several potential mechanisms by which KAP1 could control elongation, including direct interactions with Pol II and elongation rate factors (SPT5) (**lines 501-513**), the deposition of transient heterochromatin to slow Pol II in IEGs gene bodies (**lines 514-526**), or indirectly by operating with the DNA damage and repair machinery (**lines 527-540**). Collectively, we thank the reviewer for this comment, which has allowed us to improve the presentation of our work.

Point 2. A second point is about ascribing the effect of KAP1 depletion to elongation. The arguments presented in new Figure 3 show a modest increase in transcription signal within gene bodies. One can argue that the same effect would be observed when another, potentially limiting upstream step, such as initiation or release, is sped up. The resolution of current approaches might not be able to distinguish these conclusively. One should also not discount potential effects on possible redirection of paused Pol II between elongation and premature termination. Making conclusive statements requires addressing all these mechanisms. The ChIP with GTFs is not conclusive since changes in promoter occupancy per time unit may lead to unpredictable changes in ChIP signal of some components. Authors currently do not present evidence reasonably excluding the effect of KAP1 on other steps of transcription. This includes the proposed methods to devise elongation kinetics by proxy that remains not well validated at this point. A modest positive correlation comparing outcomes using previously published elongation wave measurements make it somewhat possible, but not beyond reasonable doubt just yet.

We thank the reviewer for this important point suggesting that we have not ruled out every other transcription step beyond elongation. It is noteworthy that the major transcriptional defect upon acute KAP1 depletion is found in gene bodies and 3' ends, but not at promoter-proximal regions, which are typically subject to phenotypic differences in Pol II levels upon genetic perturbation of transcriptional regulators. Together these provide supporting evidence ruling out initiation and pause release as contributing factors. We hope the reviewer will at least partially agree that by performing extensive experimentation in the revision period to assess pause release factors (CDK9 and SPT5) and CDK9 kinase activity (as Reviewer 2 well appreciated), showed that at least the major rate-limiting step of pausing–pause release does not appear to be driving the increased transcription observed upon acute KAP1 depletion during early serum stimulation. This work is presented in **Supplementary Fig. 6** and described in the text (**pg. 11, lines 420-448**). Below, we address every other comment the reviewer has mentioned in this point and made the corresponding modifications to the text, including a completely new “Limitations of the study” (**pg. 16, lines 616-651**) section to discuss potential limitations and future work.

The arguments presented in new Figure 3 show a modest increase in transcription signal within gene bodies.

We fully agree with the reviewer that the ChIP-Seq data in **Fig. 3** at the early serum stimulation time point (15 min) shows a modest increase in Pol II density in the gene bodies and 3' ends of IEGs upon acute KAP1 depletion, but no major changes to Pol II density in their promoter-proximal regions. We emphasized (with this reviewer's suggestion) in the revision that these phenotypes are indeed modest. Nonetheless, this modest phenotype upon acute depletion of a transcriptional regulator is the **true phenotype**, without any confounding secondary effects, thus increasing its value to define “direct” molecular mechanisms.

One can argue that the same effect would be observed when another, potentially limiting upstream step, such as initiation or release, is sped up. The resolution of current approaches might not be able to distinguish these conclusively.

We agree with the reviewer that augmented initiation could contribute to the increase in Pol II gene body levels observed by Pol II ChIP-Seq (**Fig. 3A**) and nascent RNA transcription observed by PRO-Seq (**Fig. 4A**). Although we have partially ruled out this possibility, we have now added a few sentences to reflect this in the new “Limitations of the study” paragraph in the Discussion section. We also agree that we have not measured the rate of “initiating” Pol II, which is more difficult to quantitate given its dynamic behavior. Nonetheless, we used a practical approach in the field (Mimoso and Adelman, Mol Cell 2023, PMID: 36965480) to indirectly monitor initiation, which is quantifying promoter-proximal Pol II levels using various experimental assays. The rate of increase in promoter-proximal Pol II levels is at least a proxy for initiation in this way, and by most accounts (ChIP-Seq in **Fig. 3A** and PRO-Seq in **Fig. 4A**) it shows that promoter-proximal Pol II levels do not increase at the early time points of serum stimulation at the meta (**Fig. 3A** and **Fig. 4A**) or gene-wise (**Fig. 3C, Fig. 3F, Fig. 3G, Fig. 4D, Supplementary Fig. 4A, Supplementary Fig. 4D, Supplementary Fig. 4E**) levels upon acute KAP1 depletion. We hope that this comment, along with the clear notation in the new Limitations section fully addresses this query.

One should also not discount potential effects on possible redirection of paused Pol II between elongation and premature termination. Making conclusive statements requires addressing all these mechanisms. The ChIP with GTFs is not conclusive since changes in promoter occupancy per time unit may lead to unpredictable changes in ChIP signal of some components.

We fully agree with this point and thus have now incorporated a statement regarding premature termination in the “Limitations of this study” section. We also agree that GTF ChIP-Seq’s presented may lead to unpredictable changes in ChIP signal, and again want to remind the reviewer that the changes in ChIP signal of GTFs at the late serum stimulation time point (30 min) (**Fig. 5; Supplementary Fig. 7**) is an indirect consequence of the transcription perturbations observed at the earlier serum stimulation time point (5 min). The suggestion that this data is an indirect consequence is discussed in **pg. 11, lines 452-455**. Importantly, the lack of changes at the level of promoter-proximal Pol II at the early serum stimulation time point, as well as no increases in the occupancy of transcription initiation factors such as MED1, which was measured during the revision period, indicates that initiation does not appear to be driving the major phenotype in gene bodies, together supporting the elongation model.

Authors currently do not present evidence reasonably excluding the effect of KAP1 on other steps of transcription. This includes the proposed methods to devise elongation kinetics by proxy that remains not well validated at this point. A modest positive correlation comparing outcomes using previously published elongation wave measurements make it somewhat possible, but not beyond reasonable doubt just yet.

We thank the reviewer for this comment. We respectfully disagree that we have at least partially addressed that pause release (described above and shown in **Supplementary Fig. 6**) may not be a major driver of the mechanism, as well as using accessible approaches to quantify levels of promoter-proximal Pol II to gain a proxy for initiation (described above and in the new “Limitations of the study” section within the Discussion). We also thank the reviewer for the comment regarding the correlation analysis, which at least partially validates the Proxy Rate approach. Given the technical limitations of short IEGs, it is impossible to conduct elongation rate measurements in a standard fashion; and thus, the only option available was to devise a statistical method that was benchmark validated against existing approaches. We hope the reviewer will understand the limitations and acknowledge the efforts on our parts to validate the model to help move the field forward.

One way to address these points is to tone down interpretation of data within Results section and offer interpretation as part of discussion, acknowledging limitations.

This is a good suggestion. We have now toned down the data interpretation within the Results section when considered appropriate and included a new set of paragraphs as part of the Discussion to acknowledge current limitations and future work. Taken together, we sincerely thank the reviewer for allowing us to improve the presentation of our work and provide clarity by increasing the transparency of the limitations of our study.

Minor points

Line 14 and elsewhere in the text. What do authors mean by “high-resolution” approaches? Currently the data are interpreted at a gene body level max, as any high-resolution information (such as in line 353) is not used as such without averaging. Deleting this wording does not diminish anything. The same may apply to “in real time” (line 81): nascent sequencing is not necessarily real-time, but is a destructive technique that either is a pulse over time (TT-seq) or requires permeabilization of cells for a time at non-physiological temperature (PRO-seq), either way hardly real time (like live microscopy, for example). Wording should be amended or clarified.

We thank the reviewer for this point. We have now removed “high-resolution” and “in real time” from all parts of the text. Specific locations are listed below.

Pg. 1, lines 13-16: We have removed ‘high-resolution’ here: “By combining an acute depletion system with several genomics approaches to interrogate synchronized, temporal transcription, we reveal that KAP1/TRIM28 is a first responder that fulfills the temporal and heightened transcriptional demand of IEGs.”

Pg. 3, lines 79-82: We have removed both ‘high-resolution’ and ‘in real time’ here: “To circumvent this caveat, we devised a dTAG-inducible acute KAP1 depletion system to monitor synchronized, signal-induced transcription during a stimulation time course using several genomic approaches.”

Line 99 – delete “near complete”, as 90% depletion is not necessarily near complete and should be left for the reader to choose to interpret as such or not.

We thank the reviewer for this comment and agree. We have corrected this statement to now read the following on **pg. 4, lines 98-101**: “Monitoring of protein degradation kinetics in this system revealed that expression of KAP1 is reduced by ~88% by 8 hr of dTAG treatment, and thus this timepoint was used for all subsequent experiments (**Fig. 1B; Supplementary Fig. 1A**).”

Do lines 125-128 better belong to just before 131 in the next section?

We thank the reviewer for this suggestion. After placing the text in both spots to assess clarity, we decided that the original location of the text makes the transition between the two sections more understandable and allows for a smooth transition to the next section. We hope the reviewer will understand this assessment that does not significantly change the overall text.

Line 243 and the rest of the section. The word “divergent” may be confusing as applies to transcription, as it by now reserved for bidirectional transcription, but is being used here for something else. Could authors find another way to describe changes in kinetics?

We agree that ‘divergent’ may not be the best word here but wanted to convey that the Pol II occupancy differences in the two time points were not identical. The revised title is: “Acute KAP1 depletion leads to **disparate** Pol II occupancy alterations during early and late serum stimulation”. (see **pg. 7, lines 241-242**).

Authors still use “elongation kinetics” wording in a way that may be confusing, as the word has a broader meaning than speed. If authors mean elongation speed or elongation velocity, this substitute should be made as appropriate.

After much consideration, we purposefully used the term “elongation kinetics” in the original and revised submissions. Since the largest changes upon acute KAP1 depletion at the earliest time points of serum induction were in the gene bodies of IEGs, we reasoned that elongation (and not pause release) was the primary mechanism. There are two ways to facilitate elongation, including regulating Pol II processivity or elongation rate/speed/velocity (described in detail by Muniz et al. EMBO J 2021, PMID: 34254686). Processivity is the ability for Pol II molecules to travel through the end of the gene, which we observed is occurring in both dTAG- and DMSO-treated cells (see the PRO-Seq data after 10 min of serum stimulation, **Fig. 4B**), suggesting that processivity is not affected upon acute KAP1 depletion. Elongation rate is a quantitative term used to define the number of nucleotides synthesized over time (kb/min). When assessing our own data, metagene profiling of active Pol II molecules suggested a modest phenotype where Pol II reached further into the ends of IEGs in dTAG-treated cells relative to DMSO-treated cells. This data suggests that elongation rate or speed may be a primary mechanism for increased transcription observed at the early time points of serum stimulation upon acute KAP1 depletion. However, we are unable to use the terms rate, velocity, or speed because of the technical limitations of short genes making it impossible to directly obtain rates. This is described in detail in **pg. 9, lines 327-353** and in the new “Limitations of the study” section within the Discussion. After much careful thought, and discussions with peers in the field, we think elongation kinetics, which is a broad term used to describe a faster movement of Pol II molecules in gene bodies not associated with a particular quantitation, is the appropriate word to use for reasons described above.

Line 159 – should “were” be “was”?

We thank the reviewer for this comment to improve the text. We have now corrected this on **pg. 5, line 157**.

Line 206 – “signifying KAP1” should be “signifying that KAP1”

We thank the reviewer for this comment to improve the text. We have now corrected this on **pg. 6, line 204**.

Lines 300-314. The effect on elongation does not seem to rise to the level of “strikingly”. This is because the timing at which this change in Pol II distribution is observed (5 minutes) is much, much slower than what it would have taken one Pol II to elongate through the gene. The effects observed are, therefore, most probably arise from a sum of many molecules in a quasi-equilibrium and the shape of this meta distribution does not necessarily directly reflect the speed of elongation.

We thank the reviewer for this comment but we partially disagree with the assessment. A 2-fold transcriptional phenotype upon acute depletion of a single transcriptional regulator (not a major regulator) is indeed remarkable. The problem is that the field was until now accustomed to larger phenotypes due to either the inactivation of major transcriptional regulators (for example an upstream kinase) or to the accumulation of indirect effects that feedback on the transcription process, thus obscuring the true phenotypes. However, given the reviewer’s critique, we have now changed the ‘Strikingly’ opener of the sentence to ‘Notably’ on **pg. 8, line 300**.

We also fully agree with the reviewer that the timing at which the change in Pol II distribution is observed (5 min) is much slower than what it would have taken one Pol II molecule to elongate through the gene. This is emphasized as the meta distribution shows that after 5 min of serum stimulation, Pol II **does not** reach the ends of IEGs and is still concentrated in the 5’ part of the gene (for both DMSO and dTAG conditions). In fact, this time point was chosen precisely for this

reason to monitor elongation, as elongation rate is normally measured by monitoring the wavefront of Pol II ('leading edge') after either Pol II arrest in the paused state (using inhibitors of pause release) or stimulation to turn on the gene (analogous to our case). Measuring rate in this way requires the monitoring of active Pol II molecules prior to the time it would take for Pol II to elongate through the gene. We hope that the reviewer agrees that this time point was a conscious one.

We also fully agree that the observed effects (increased engaged Pol II molecules in the gene body upon acute KAP1 depletion) are a sum of many molecules, and not a single-molecule measurement traveling across IEGs after serum stimulation. Despite the fact that this is a population-based assay, we posit that it is at least a proxy for identifying how individual loci may behave at the single-cell level, and currently population-based assays (such as ChIP-Seq, PRO-Seq, RNA-Seq, ATAC-Seq, etc.) are still being used to mechanistically characterize Pol II transcription behaviors. Given the impracticality of performing single-cell experiments (which are currently not feasible with run-on based assays such as PRO-Seq), we hope the reviewer will accept this as a technical limitation that does not impact the major conclusions of our study. Additionally, the approach of using the distance that Pol II travels in a particular cell condition using population-based assays (such as GRO-Seq) to specifically monitor elongation rate is a standard approach in the field (e.g., Jonkers et al., eLife 2014, PMID: 24843027 and Danko et al., Mol Cell 2013, PMID: 23523369). We simply co-opted this approach to calculate a Proxy for elongation rate.

Reviewer 2

This is a revised version of a manuscript I reviewed previously. The authors have addressed all of my concerns and, in my opinion, the paper is now suitable for publication in Nature Communications. (I did catch one incorrect figure callout, in line 466, to a non-existent Fig. 4G,H, which should be Fig. 5G,H.)

We thank the reviewer for this comment and for suggesting that the manuscript is suitable for publication. We also thank the reviewer for making note of this incorrect figure callout, which we have now corrected on the same line (**pg. 12, line 464**).